

# Riemann surfaces for KPZ with periodic boundaries

**Sylvain Prolhac**[*]

Laboratoire de Physique Théorique, IRSAMC, UPS, Université de Toulouse, France

[*] sylvain.prolhac@irsamc.ups-tlse.fr

## Abstract

The Riemann surface for polylogarithms of half-integer index, which has the topology of an infinite dimensional hypercube, is studied in relation to one-dimensional KPZ universality in finite volume. Known exact results for fluctuations of the KPZ height with periodic boundaries are expressed in terms of meromorphic functions on this Riemann surface, summed over all the sheets of a covering map to an infinite cylinder. Connections to stationary large deviations, particle-hole excitations and KdV solitons are discussed.

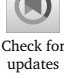

# 1   Introduction

KPZ universality in 1+1 dimension [1–9] describes large scale fluctuations appearing in a variety of systems such as growing interfaces [10], disordered conductors [11], one-dimensional classical [12–14] and quantum [15–17] fluids, or traffic flow [18]. The height field $h_\lambda(x,t)$ characterizing KPZ universality depends on position $x \in \mathbb{R}$, time $t \geq 0$, and on a parameter $\lambda > 0$ quantifying the strength of non-linear effects and the non-equilibrium character of the dynamics. The fluctuations of $h_\lambda(x,t)$ are believed to be universal in the sense that for a given geometry (infinite system, presence of various kinds of boundaries) and a given initial condition, the probability distribution of the appropriate height field $h_\lambda(x,t)$ is independent of the specific setting in KPZ universality and of the precise microscopic model studied at large scales. A prominent model, which has given its name to the universality class, is the KPZ equation [19], defined as the properly renormalized [20–22] non-linear stochastic partial differential equation $\partial_t h_\lambda = \frac{1}{2}\partial_x^2 h_\lambda - \lambda(\partial_x h_\lambda)^2 + \eta$ with $\eta$ a unit space-time Gaussian white noise, and which is related by the Cole-Hopf transform $Z_\lambda(x,t) = e^{-2\lambda h_\lambda(x,t)}$ to the stochastic heat equation with multiplicative noise $\partial_t Z_\lambda = \frac{1}{2}\partial_x^2 Z_\lambda - 2\lambda Z_\lambda \eta$ and Ito prescription in the time variable.

   Of particular interest is the limiting object $h(x,t) = \lim_{\lambda\to\infty}(h_\lambda(x,t/\lambda) - \lambda^2 t/3)$ into the regime where non-linear effects dominate, and for which a number of exact results have been obtained in the past 20 years [23–34] for the infinite system geometry $x \in \mathbb{R}$. Most notably, connections to random matrix theory have been identified: for given time $t \to \infty$ and position $x \in \mathbb{R}$, the probability distribution of $h(x,t)$ for specific initial conditions are equal to Tracy-Widom distributions [35], known for describing fluctuations of extremal eigenvalues in random matrix theory.

   We are interested in this paper in KPZ universality in finite volume, specifically with periodic boundary conditions $x \equiv x+1$, in the strongly non-linear regime $\lambda \to \infty$. There, the standard deviation of $h(x,t)$ grows as $t^{1/3}$ at short times like in the infinite system $x \in \mathbb{R}$, before eventually saturating after the statistics of fluctuations has relaxed to a stationary distribution where $x \mapsto h(x,t)$ is Brownian. Large deviations in the stationary state away from typical Gaussian fluctuations are known explicitly [36,37], and long time corrections to large deviations have been obtained explicitly [38] for a few specific initial conditions. Interestingly, the exact expressions in [36–38] involve polylogarithms with half-integer index. The analytical and topological structure of these special functions is at the heart of the present paper.

   The complete evolution in time of KPZ fluctuations with periodic boundaries, crossing over between the short time limit, where the correlation length is much smaller than the system size and the fluctuations of the infinite system are recovered, and the long time limit where stationary large deviations appear, has been studied recently. Exact expressions have been obtained for the one-point [39–41] distribution $P(h(x,t) < u)$ of the height field for specific

(sharp wedge, stationary and flat) initial conditions, as well as for the general, multiple-time joint distribution $P(h(x_1, t_1) < u_1, \ldots, h(x_n, t_n) < u_n)$ for sharp wedge initial condition [42]. All these exact expressions have a somewhat complicated structure involving combinations of square roots and half-integer polylogarithms.

A goal of the present paper is to show that the full crossover regime for KPZ fluctuations in finite volume have rather simple expressions using objects from algebraic geometry directly connected to stationary large deviations. More precisely, considering the (infinite genus) Riemann surface $\check{\mathcal{R}}$ on which polylogarithms with half-integer index are defined globally, the exact expression for the probability $\mathbb{P}(h(x, t) < u)$ with flat initial condition is rewritten as the integral around an infinitely long cylinder $\mathcal{C}$ of a holomorphic differential on $\check{\mathcal{R}}$ summed over all the sheets of a ramified covering from $\check{\mathcal{R}}$ to $\mathcal{C}$, see equation (1). Similar expressions are obtained for sharp wedge and stationary initial conditions, with an additional summation over finite subsets $\Delta$ of $\mathbb{Z} + 1/2$, and $\check{\mathcal{R}}$ replaced by related Riemann surfaces $\mathcal{R}^\Delta$, see equations (6), (11). Multiple integrals around the cylinder as well as meromorphic functions on pairs of Riemann surfaces $\mathcal{R}^\Delta \times \mathcal{R}^\Gamma$ are additionally needed for the multiple-time joint distribution with sharp wedge initial condition, see equation (16).

The paper is organized as follows. In section 2, our main results expressing KPZ fluctuations with periodic boundaries in terms of infinite genus Riemann surfaces $\check{\mathcal{R}}$, $\mathcal{R}^\Delta$ and ramified coverings from them to the infinite cylinder $\mathcal{C}$ are given. Interpretations in terms of particle-hole excitations and KdV solitons are pointed out at the end of the section. In section 3, we recall some classical aspects of the theory of Riemann surfaces and ramified coverings used in the rest of the paper, and define the Riemann surfaces $\check{\mathcal{R}}$ and $\mathcal{R}^\Delta$. In section 4, we study several meromorphic functions defined on these Riemann surfaces and needed for KPZ fluctuations. Finally, we explain in section 5 how our main results from section 2 are related to earlier exact formulas [39–42]. Some technical calculations are presented in appendix.

## 2 KPZ fluctuations and Riemann surfaces

In this section, we give exact expressions for KPZ fluctuations with periodic boundary conditions equivalent to those obtained in [39–42], but written in a more unified way by interpreting various terms as natural objects living on Riemann surfaces evaluated on distinct sheets. Connections to stationary large deviations and interpretations in terms of particle-hole excitations and KdV solitons are discussed in some detail toward the end of the section.

### 2.1 Riemann surfaces $\check{\mathcal{R}}$ and $\mathcal{R}^\Delta$

The exact formulas for KPZ fluctuations given below involve Riemann surfaces $\check{\mathcal{R}}$ and $\mathcal{R}^\Delta$, quotient under groups of holomorphic automorphisms of a Riemann surface $\mathcal{R}$ which has the topology of an infinite dimensional hypercube and which is a natural domain of definition for some infinite sums of square roots with branch points $2i\pi a$, $a \in \mathbb{Z} + 1/2$. An introduction to several topics related to Riemann surfaces used in this paper is given in section 3, starting with a finite genus analogue $\mathcal{R}_N$ of $\mathcal{R}$ before giving precise definitions of $\check{\mathcal{R}}$ and $\mathcal{R}^\Delta$.

The Riemann surface $\mathcal{R}$ has a kind of translation invariance inherited from that of the branch points $2i\pi a$, and which is eliminated by definition in $\check{\mathcal{R}}$. The Riemann surfaces $\mathcal{R}^\Delta$ are indexed by finite sets $\Delta$ of half-integers, which we write as $\Delta \sqsubset \mathbb{Z} + 1/2$. The elements $a \in \Delta$ index branch points $2i\pi a$ that have been removed from the infinite sum of square roots defined on $\mathcal{R}$. The translation invariance of $\mathcal{R}$ implies that $\mathcal{R}^\Delta \sim \mathcal{R}^{\Delta+1}$ are isomorphic Riemann surfaces.

The Riemann surface $\mathcal{R}$ can be partitioned into sheets $\mathbb{C}_P$, $P \sqsubset \mathbb{Z} + 1/2$, copies of the

$$
\begin{array}{ccccc}
 & & \check{\mathcal{R}} & & \\
 & \overset{\check{\Pi}}{\nearrow} & & \overset{\check{\rho}}{\searrow} & \\
\mathcal{R} & \overset{\Pi}{\longrightarrow} & & \mathcal{C} & \underset{\lambda}{\longrightarrow} \quad \mathbb{C}^* \\
 & \underset{\Pi^{\Delta}}{\searrow} & & \underset{\rho^{\Delta}}{\nearrow} & \\
 & & \mathcal{R}^{\Delta} & & 
\end{array}
$$

Figure 1: Summary of several useful covering maps between the Riemann surfaces considered in this paper.

complex plane glued together along branch cuts of the square roots, and we write $[v, P]$, $v \in \mathbb{C}$, $P \sqsubset \mathbb{Z} + 1/2$ for a point of $\mathcal{R}$ with branch cuts chosen as in figure 18 right. The Riemann surfaces $\check{\mathcal{R}}$ and $\mathcal{R}^{\Delta}$ are identified as fundamental domains for corresponding group actions on $\mathcal{R}$, see respectively sections 3.8.2 and 3.8.3, and may be partitioned by portions of the sheets $\mathbb{C}_P$. For $\check{\mathcal{R}}$, one can choose the union of infinite strips $\mathcal{S}_P^0 = \{[v, P], -\pi < \mathrm{Im}\, v \le \pi\}$, $P \sqsubset \mathbb{Z} + 1/2$, see figure 13 left. For $\mathcal{R}^{\Delta}$, a possible fundamental domain is the union of all $\mathbb{C}_P$, $P \cap \Delta = \emptyset$, see figure 16. Additionally, the collection of non-isomorphic $\mathcal{R}^{\Delta}$, $\Delta \equiv \Delta + 1$, may also be partitioned into the infinite strips $\mathcal{S}_P^0$ from all $\mathcal{R}^{\Delta}$ without the restriction $\Delta \equiv \Delta + 1$, see figure 17.

The definitions of $\check{\mathcal{R}}$ and $\mathcal{R}^{\Delta}$ from $\mathcal{R}$ provide covering maps $\check{\Pi}$ and $\Pi^{\Delta}$ from $\mathcal{R}$ to $\check{\mathcal{R}}$ and $\mathcal{R}^{\Delta}$, see figure 1. Additionally, there exists natural covering maps $\check{\rho}$ and $\rho^{\Delta}$ from $\check{\mathcal{R}}$ and $\mathcal{R}^{\Delta}$ to the infinite cylinder $\mathcal{C} = \{v \in \mathbb{C}, v \equiv v + 2\mathrm{i}\pi\}$, with ramification points $[2\mathrm{i}\pi a, P]$, $a \in \mathbb{Z} + 1/2$ (and additionally $a \notin \Delta$ for $\rho^{\Delta}$). One-point statistics of the KPZ height field with periodic boundaries are expressed below for various initial conditions as an integral over a loop around the cylinder. The integrand involves holomorphic differentials traced over the covering maps $\check{\rho}$ or $\rho^{\Delta}$, i.e. summed over all the sheets of the Riemann surfaces covering the cylinder. The functions and meromorphic differentials needed are studied in detail in section 4.

## 2.2 Flat initial condition

We consider in this section the one-point distribution $\mathbb{P}_{\mathrm{flat}}(h(x, t) > u)$ of KPZ fluctuations with periodic boundary conditions, $h(x, t) = h(x + 1, t)$, and flat initial condition $h(x, 0) = 0$. We claim that the properly renormalized random field $h(x, t)$ has the cumulative density function[1]

$$
\boxed{\mathbb{P}_{\mathrm{flat}}(h(x, t) > u) = \int_{\gamma} (\mathrm{tr}_{\check{\rho}}\, Z_{t,u}^{\mathrm{flat}})(v)}\,, \tag{1}
$$

with $\gamma$ a loop around the infinite cylinder $\mathcal{C}$ with winding number 1. The holomorphic differential $Z_{t,u}^{\mathrm{flat}}$ on the Riemann surface $\check{\mathcal{R}}$, defined away from ramification points of $\check{\rho}$ as

$$
Z_{t,u}^{\mathrm{flat}}([v, P]) = \exp\Big( \int_{[-\infty, \emptyset]}^{[v, P]} S_{t,u}^{\mathrm{flat}} \Big) \frac{\mathrm{d}v}{2\mathrm{i}\pi}\,, \tag{2}
$$

---

[1] The definition of $h(x, t)$ used in this paper corresponds to a growth of the height function for TASEP in the positive direction, which after proper rescaling gives a growth of the KPZ height function in the negative direction, $h(x, t) \to -\infty$ when $t \to \infty$, corresponding to a negative coefficient $-\lambda \to -\infty$ in front of the non-linear term in the KPZ equation. The same convention for the sign of $u$ is used in [39]. The opposite convention is used in [40], with the notation $x = -u$ there.

is built from an integral of the meromorphic differential $S_{t,u}^{\text{flat}}$ on $\check{\mathcal{R}}$ given by

$$S_{t,u}^{\text{flat}}(p) = \left( t\chi'(p) - u\chi''(p) - \frac{1/4}{1+e^{-v}} + \frac{\chi''(p)^2}{2} \right) dv \tag{3}$$

at $p = [v, P] \in \check{\mathcal{R}}$ away from ramification points of $\check{\rho}$. Here, $\chi$, $\chi'$, $\chi''$ are meromorphic functions on $\check{\mathcal{R}}$, obtained by analytic continuations of the polylogarithm $\chi_\emptyset(v) = -\text{Li}_{5/2}(-e^v)/\sqrt{2\pi}$ and its derivatives, see sections 4.4 and 4.5, and equations (66), (64), (57), (50) for precise definitions. We also refer to section 4 for explicit formulas for analytic continuations and proofs that $Z_{t,u}^{\text{flat}}$ is indeed holomorphic on $\check{\mathcal{R}}$ and independent from the path of integration in (2).

The trace of a meromorphic differential with respect to a covering map is defined in (34). For the covering map $\check{\rho} : [v, P] \mapsto v$ from the Riemann surface $\check{\mathcal{R}}$ to the infinite cylinder $\mathcal{C}$, the trace $\text{tr}_{\check{\rho}}$ consists in summing over all the infinite strips $\mathcal{S}_P^0 = \{[v, P], -\pi < \text{Im } v \le \pi\}$ partitioning $\check{\mathcal{R}}$, see sections 3.8.1 and 3.8.2 for a precise definition of $\check{\mathcal{R}}$, and especially the left side of figure 13 for a graphical representation of how $\check{\mathcal{R}}$ is partitioned into infinite strips. The integral in (1) is independent of the loop $\gamma$, since $\text{tr}_{\check{\rho}} Z_{t,u}^{\text{flat}}$ is holomorphic on the cylinder $\mathcal{C}$ by the properties of the trace. With the change of variable $z = e^v$, defining a covering map $\lambda$ from $\mathcal{C}$ to the punctured plane $\mathbb{C}^* = \mathbb{C} \setminus \{0\}$, the expression (1) can alternatively be written as the integral over a loop encircling $0$ in $\mathbb{C}^*$, and $\mathbb{P}_{\text{flat}}(h(x, t) > u)$ is then the residue at the essential singularity $z = 0$ of $\text{tr}_{\lambda \circ \check{\rho}} Z_{t,u}^{\text{flat}}$.

The trace in (1) can be evaluated explicitly by considering a partition into infinite strips $\mathcal{S}_P^0$, $P \sqsubset \mathbb{Z} + 1/2$ of the Riemann surface $\check{\mathcal{R}}$, see figure 13 left, so that $(\text{tr}_{\check{\rho}} Z_{t,u}^{\text{flat}})(v) = \sum_{P \sqsubset \mathbb{Z}+1/2} Z_{t,u}^{\text{flat}}([v, P])$. In terms of the functions $\chi_P$ and $I_0$ defined in (64), (68) one has

$$\mathbb{P}_{\text{flat}}(h(x, t) > u) = \sum_{P \sqsubset \mathbb{Z}+1/2} \frac{(-1)^{|P|} V_P^2}{4^{|P|}} \int_{c-i\pi}^{c+i\pi} \frac{dv}{2i\pi} e^{t\chi_P(v) - u\chi_P'(v) + I_0(v) + \frac{1}{2}\int_{-\infty}^v dv\, \chi_P''(v)^2}, \tag{4}$$

with $c \in \mathbb{R}^*$. The summation is over all finite subsets $P$ of $\mathbb{Z} + 1/2$, and $V_P$ is the Vandermonde determinant

$$V_P = \prod_{\substack{a,b \in P \\ a > b}} \left( \frac{2i\pi a}{4} - \frac{2i\pi b}{4} \right). \tag{5}$$

The function $\chi_P$ is the restriction of $\chi$ to the sheet $\mathbb{C}_P$, $I_0$ is given by $e^{I_0(v)} = (1 + e^v)^{-1/4}$ if $c < 0$ or $e^{I_0(v)} = e^{-v/4}(1 + e^{-v})^{-1/4}$ if $c > 0$, and $\int_{-\infty}^v dv\, \chi_P''(v)^2 = \lim_{\Lambda \to \infty} -|P|^2 \log \Lambda + \int_{-\Lambda}^v dv\, \chi_P''(v)^2$. The extra factor $(-1)^{|P|} V_P^2/4^{|P|}$ in (4) compared to (1) comes from the analytic continuation of $\int_{-\infty}^v dv\, \chi_\emptyset''(v)^2$ from $\mathbb{C}_\emptyset$ to $\mathbb{C}_P$, see sections 4.7 and 4.8.

The expression (1) is justified in section 5.1 by showing that (4) is equivalent to exact results obtained previously in [39, 40] from large scale asymptotics for the totally asymmetric simple exclusion process (TASEP), a discrete interface growth model in KPZ universality. This shows in particular that the corresponding expressions from [39] and [40] agree, which had not been properly derived before, and simply represent distinct choices for a fundamental domain $\check{\mathcal{R}}$ in $\mathcal{R}$.

The probability $\mathbb{P}_{\text{flat}}(h(x, t) > u)$ is interpreted in section 2.6.3 as a $N$-soliton $\tau$ function for the KdV equation, $N \to \infty$, averaged over the common velocity $v$ of the solitons, which is also identified as a moduli parameter for specific singular hyperelliptic Riemann surfaces.

## 2.3 Sharp wedge initial condition

We consider in this section one-point statistics of KPZ fluctuations with sharp wedge [2] initial condition $h(x,0) = -\frac{|x-1/2|}{0^+}$, where $h(x,t)$ is defined after appropriate regularization. More generally, it is expected from large deviation results [38] that any initial condition of the form $h(x,0) = h_0(x)/\epsilon$ where $h_0$ is continuous on the circle $x \equiv x + 1$ with a global minimum 0 reached at $x = 1/2$ only, is equivalent in the limit $\epsilon \to 0^+$ to sharp wedge initial condition.

We show in section 5.2 that known exact formulas [39, 40] for the cumulative density function of the KPZ height are equivalent to

$$\mathbb{P}_{\text{sw}}(h(x,t) > u) = \sum_{\substack{\Delta \sqsubset \mathbb{Z}+1/2 \\ \Delta \equiv \Delta+1}} \Xi_x^\Delta \int_\gamma \left(\text{tr}_{\check{\rho}^\Delta} Z_{t,u}^{\Delta,\text{sw}}\right)(\nu), \tag{6}$$

with $\gamma$ as in (1), $\check{\rho}^\Delta = \rho^\Delta$, $\Delta \neq \emptyset$ the covering map defined in section 3.8.3 from $\mathcal{R}^\Delta$ to the infinite cylinder $\mathcal{C}$ and $\check{\rho}^\emptyset = \check{\rho}$ the covering map defined in section 3.8.1 from $\check{\mathcal{R}}$ to $\mathcal{C}$, where $\check{\mathcal{R}} = \mathcal{R}/\check{\mathfrak{g}}$ is the quotient of $\mathcal{R}^\emptyset = \mathcal{R}$ by a group $\check{\mathfrak{g}}$ of translation automorphisms that exist only for $\Delta = \emptyset$. The trace with respect to $\check{\rho}^\Delta$ gives a holomorphic differential on the cylinder $\mathcal{C}$, periodic in $\nu$ with period $2i\pi$. The sum over non-isomorphic Riemann surfaces $\mathcal{R}^\Delta$ is weighted by

$$\Xi_x^\Delta = (i/4)^{|\Delta|} \sum_{\substack{A \subset \Delta \\ |A| = |\Delta \setminus A|}} e^{2i\pi x\left(\sum\limits_{a\in A} a - \sum\limits_{a\in\Delta\setminus A} a\right)} V_A^2 V_{\Delta\setminus A}^2, \tag{7}$$

with $V_A$ the Vandermonde determinant (5) and $2\pi(\sum_{a\in A} a - \sum_{a\in\Delta\setminus A} a)$ the momentum coupled to the coordinate $x$ along the interface, which appears only through $\Xi_x^\Delta$ in (6). Only sets $\Delta$ with cardinal $|\Delta|$ even contribute. The holomorphic differential

$$Z_{t,u}^{\Delta,\text{sw}}([\nu,P]) = \exp\left(\fint_{[-\infty,\emptyset]}^{[\nu,P]} S_{t,u}^{\Delta,\text{sw}}\right) \frac{d\nu}{2i\pi} \tag{8}$$

is built from an integral with appropriate regularization at $[-\infty,\emptyset]$ of the meromorphic differential $S_{t,u}^{\Delta,\text{sw}}$ on $\mathcal{R}^\Delta$ given by

$$S_{t,u}^{\Delta,\text{sw}}(p) = \left(t\chi'^\Delta(p) - u\chi''^\Delta(p) + \chi''^\Delta(p)^2\right)d\nu \tag{9}$$

at $p = [\nu,P] \in \mathcal{R}^\Delta$ away from ramification points of $\rho^\Delta$. The functions $\chi^\Delta$, $\chi'^\Delta$, $\chi''^\Delta$, analogues of $\chi$, $\chi'$, $\chi''$ from the previous section with ramification points $[2i\pi a,P]$, $a \in \Delta$ removed, are defined in (86), (91).

The trace in (6) can be evaluated more explicitly by considering appropriate partitions into sheets of the Riemann surfaces $\check{\mathcal{R}}$ and $\mathcal{R}^\Delta$. For the term $\Delta = \emptyset$, one has $(\text{tr}_{\check{\rho}} Z_{t,u}^{\emptyset,\text{sw}})(\nu) = \sum_{P \sqsubset \mathbb{Z}+1/2} Z_{t,u}^{\emptyset,\text{sw}}([\nu,P])$ like for flat initial condition, see figure 13 left. For $\Delta \neq \emptyset$, one has to sum instead over all strips $\mathcal{S}_P^m$, $P \cap \Delta = \emptyset$, $m \in \mathbb{Z}$, see figure 16, leading to an integral between $c - i\infty$ and $c + i\infty$ of $Z_{t,u}^{\emptyset,\text{sw}}([\nu,P])$, summed over all $P$, $P \cap \Delta = \emptyset$. The symmetry of the extension to $\overline{\mathcal{R}}$ of $Z_{t,u}^{\emptyset,\text{sw}}$ under the holomorphic automorphism $\overline{\mathcal{T}}$ defined in (44), see section 3.8.4, allows to sum instead over all $\Delta$ and not just equivalence classes $\Delta \equiv \Delta + 1$, and integrate only over the strip $\mathcal{S}_P^0$ from each $\mathcal{R}^\Delta$. Using explicit analytic continuations from

---

[2]Also called step or domain wall initial condition in the context of TASEP as a microscopic model.

section 4.9.2, we finally obtain that (6) is equivalent to the more explicit expression

$$\mathbb{P}_{\text{sw}}(h(x,t) > u) = \sum_{\Delta \sqsubset \mathbb{Z}+1/2} \Xi_x^\Delta \sum_{\substack{P \sqsubset \mathbb{Z}+1/2 \\ P \cap \Delta = \emptyset}} (\text{i}/4)^{2|P|} \left( \prod_{a \in P} \prod_{\substack{b \in P \cup \Delta \\ b \neq a}} \left( \frac{2\text{i}\pi a}{4} - \frac{2\text{i}\pi b}{4} \right)^2 \right)$$

$$\times \int_{c-\text{i}\pi}^{c+\text{i}\pi} \frac{\text{d}\nu}{2\text{i}\pi} \, \text{e}^{t\chi_P^\Delta(\nu) - u\chi_P'^\Delta(\nu) + \fint_{-\infty}^\nu \text{d}\nu \, \chi_P''^\Delta(\nu)^2}, \quad (10)$$

with $|P|$ the number of elements in $P$, $\chi_P^\Delta$ the restriction of $\chi^\Delta$ to the sheet $\mathbb{C}_P$ of $\mathcal{R}^\Delta$ given in (87), and $\fint$ the regularized integral subtracting the divergent logarithmic term at $-\infty$ like in (92). The expression (10) is derived in section 5.2 from earlier works [39] and [40] using the structure of the Riemann surfaces $\mathcal{R}^\Delta$ detailed in section 3.8.3 and explicit analytic continuations obtained in section 4.9.2. This shows in particular that the expressions from [39] and [40] about sharp wedge initial condition agree, which was missing so far.

## 2.4 Stationary initial condition

Exact results have also been obtained for one-point statistics of the KPZ height with stationary initial condition [39, 41], where $x \mapsto h(x,0)$ is a standard Brownian bridge. The formulas in that case are essentially the same as for sharp wedge initial condition, with only an additional harmless factor. Starting either with equation (7) of [39] (for $x = 0$) or with equation (2.1) of [41] for general $x$, we obtain by comparison to (6)

$$\mathbb{P}_{\text{stat}}(h(x,t) > u) = \sum_{\substack{\Delta \sqsubset \mathbb{Z}+1/2 \\ \Delta \equiv \Delta+1}} \Xi_x^\Delta \int_\gamma (\text{tr}_{\check{\rho}^\Delta} Z_{t,u}^{\Delta,\text{stat}})(\nu), \quad (11)$$

with

$$Z_{t,u}^{\Delta,\text{stat}}([\nu,P]) = -\sqrt{2\pi} \, \text{e}^{-\nu} \, \partial_u Z_{t,u}^{\Delta,\text{sw}}([\nu,P]) \quad (12)$$

and the same notations as in (6). A more explicit formula can be written by inserting the extra factor $-\sqrt{2\pi} \, \text{e}^{-\nu} \, \partial_u$ into (10).

## 2.5 Multiple-time statistics with sharp wedge initial condition

The joint distribution of the height at multiple times $0 < t_1 < \ldots < t_n$ and corresponding positions $x_j$ was obtained by Baik and Liu for sharp wedge initial condition in [42]. After some rewriting in section 5.3.1 based on explicit analytic continuations from section 4.9 and 4.10, we obtain

$$\mathbb{P}(h(x_1,t_1) > u_1, \ldots, h(x_n,t_n) > u_n) \quad (13)$$

$$= \left( \prod_{\ell=1}^n \sum_{\Delta_\ell \sqsubset \mathbb{Z}+1/2} \sum_{\substack{P_\ell \sqsubset \mathbb{Z}+1/2 \\ P_\ell \cap \Delta_\ell = \emptyset}} \right) \int_{c_1-\text{i}\pi}^{c_1+\text{i}\pi} \frac{\text{d}\nu_1}{2\text{i}\pi} \cdots \int_{c_n-\text{i}\pi}^{c_n+\text{i}\pi} \frac{\text{d}\nu_n}{2\text{i}\pi} \, \Xi_{x_1,\ldots,x_n}^{\Delta_1,\ldots,\Delta_n}(\nu_1,\ldots,\nu_n)$$

$$\times \left( \prod_{\ell=1}^n \text{e}^{(t_\ell - t_{\ell-1})\chi^{\Delta_\ell} - (u_\ell - u_{\ell-1})\chi'^{\Delta_\ell} + 2J^{\Delta_\ell}}(p_\ell) \right) \left( \prod_{\ell=1}^{n-1} \text{e}^{-2K^{\Delta_\ell,\Delta_{\ell+1}}}(p_\ell, p_{\ell+1}) \right),$$

with $t_0 = u_0 = 0$, $c_n < \ldots < c_1 < 0$, $p_\ell = [\nu_\ell, P_\ell]$ a point on the Riemann surface $\mathcal{R}^{\Delta_\ell}$, $\chi^\Delta$ and $\chi'^\Delta$ holomorphic functions on $\mathcal{R}^\Delta$ given in (86), (91), $\text{e}^{2J^\Delta}$ the meromorphic function on $\mathcal{R}^\Delta$

from (96) and $e^{2K^{\Delta,\Gamma}}$ a meromorphic function on $\mathcal{R}^{\Delta} \times \mathcal{R}^{\Gamma}$ defined by (118). The collection of Riemann surfaces $\mathcal{R}^{\Delta_\ell}$ in (13) is weighted by the meromorphic function on $\mathbb{C}^n$

$$\Xi_{x_1,\dots,x_n}^{\Delta_1,\dots,\Delta_n}(v_1,\dots,v_n) \tag{14}$$

$$= \left( \prod_{\ell=1}^{n} \sum_{\substack{A_\ell \sqsubset \Delta_\ell \\ |A_\ell|=|\Delta_\ell \setminus A_\ell|}} \right) \prod_{\ell=1}^{n} \left( (i/4)^{|\Delta_\ell|} V_{A_\ell}^2 V_{\Delta_\ell \setminus A_\ell}^2 e^{2i\pi(x_\ell - x_{\ell-1})\left( \sum_{a \in A_\ell} a - \sum_{a \in \Delta_\ell \setminus A_\ell} a \right)} \right)$$

$$\times \prod_{\ell=1}^{n-1} \frac{(1-e^{v_{\ell+1}-v_\ell})^{|\Delta_\ell|/2}(1-e^{v_\ell-v_{\ell+1}})^{|\Delta_{\ell+1}|/2}}{(1-e^{v_{\ell+1}-v_\ell}) V_{A_\ell,A_{\ell+1}}(v_\ell,v_{\ell+1}) V_{\Delta_\ell \setminus A_\ell, \Delta_{\ell+1} \setminus A_{\ell+1}}(v_\ell,v_{\ell+1})},$$

with $x_0 = 0$, $V_A$ the Vandermonde determinant (5) and

$$V_{A,B}(v,\mu) = \prod_{a \in A} \prod_{b \in B} \left( \frac{2i\pi a - v}{4} - \frac{2i\pi b - \mu}{4} \right). \tag{15}$$

Since $\Xi_{x_1,\dots,x_n}^{\Delta_1,\dots,\Delta_n}(v_1,\dots,v_n) = 0$ when any of the $|\Delta_\ell|$ is odd because of the constraints $|A_\ell| = |\Delta_\ell \setminus A_\ell|$, only sets $\Delta_\ell$ containing an even number of elements contribute to (13).

The same reasoning as the one between (10) and (6) allows to express (13) in terms of non-isomorphic Riemann surfaces and a trace over covering maps. One has

$$\mathbb{P}(h(x_1,t_1) > u_1, \dots, h(x_n,t_n) > u_n)$$

$$= \left( \prod_{\ell=1}^{n} \sum_{\substack{\Delta_\ell \sqsubset \mathbb{Z}+1/2 \\ \Delta_\ell \equiv \Delta_\ell + 1}} \right) \int_{\gamma_1} \dots \int_{\gamma_n} \Xi_{x_1,\dots,x_n}^{\Delta_1,\dots,\Delta_n}(v_1,\dots,v_n) \tag{16}$$

$$\times \operatorname{tr}_{\breve{\rho}^{\Delta_1}} \dots \operatorname{tr}_{\breve{\rho}^{\Delta_n}} \frac{\prod_{\ell=1}^{n} Z_{t_\ell-t_{\ell-1}, u_\ell-u_{\ell-1}}^{\Delta_\ell,\mathrm{sw}}(v_\ell)}{\prod_{\ell=1}^{n-1} e^{2K^{\Delta_\ell,\Delta_{\ell+1}}}(v_\ell,v_{\ell+1})},$$

with $\operatorname{tr}_{\breve{\rho}^{\Delta_\ell}}$ acting on $v_\ell$, $Z_{\delta t,\delta u}^{\Delta,\mathrm{sw}}$ the holomorphic differential from (8), and a loop $\gamma_\ell$ with winding number 1 around the infinite cylinder $\mathcal{C}$ for the variable $v_\ell$. Because of the trace, the integrand in (16) is meromorphic in $\mathcal{C}^n$. The loops $\gamma_\ell$ do not cross each other, as the order $\operatorname{Re} v_n < \dots < \operatorname{Re} v_1$ must be preserved because of the presence of simple poles at $v_{\ell+1} = v_\ell + 2i\pi m$, $m \in \mathbb{Z}$. Interestingly, it can be shown that such poles exist only when $\Delta_{\ell+1} = \Delta_\ell + m$ and $P_{\ell+1} = P_\ell + m$ (and only the sector $A_{\ell+1} = A_\ell + m$ of $\Xi_{x_1,\dots,x_n}^{\Delta_1,\dots,\Delta_n}$ contributes to them), corresponding to points $p_\ell = [v_\ell, P_\ell]$ and $p_{\ell+1} = [v_{\ell+1}, P_{\ell+1}]$ coinciding on the Riemann surface $\mathcal{R}^{\Delta_\ell} \sim \mathcal{R}^{\Delta_{\ell+1}}$, see section 5.3.2.

## 2.6 Discussion

In this section, we discuss various interpretations of the exact formulas given above for KPZ fluctuations with periodic boundaries.

### 2.6.1 Full dynamics from large deviations

In the long time limit, KPZ fluctuations in finite volume converge to a stationary state where the interface has the same statistics as a Brownian motion with appropriate boundary conditions. Large deviations corresponding to fluctuations of the height with an amplitude of order $t$ when $t \to \infty$ are on the other hand non-Gaussian and can be characterized by a generating function of the form [36, 43]

$$\langle e^{sh(x,t)} \rangle \simeq \theta(s) e^{t e(s)}, \tag{17}$$

where $e(s)$ involves an infinite sum of square roots. At finite time, the generating function of the height is given exactly by a sum of infinitely many terms of the same form,

$$\langle e^{sh(x,t)} \rangle = \sum_n \theta_n(s) e^{t\, e_n(s)}, \tag{18}$$

with $n$ an index labelling sheets of Riemann surfaces, see equations (122), (131). Known results for the spectrum of TASEP [44,45] indicate that the stationary contribution corresponds to the sheet $\mathbb{C}_\emptyset$ of $\mathcal{R}$ when $\operatorname{Re} s > 0$.

These observations suggest the possibility to guess the full finite time dynamics of KPZ fluctuations in finite volume from the solution of the static problem of stationary large deviations alone. For flat initial condition in particular, the functions $\theta_P(s)$, $e_P(s)$ (or, more properly, their analogues for the probability (4) after Fourier transform, see section 5.1.1) are simply analytic continuations of $\theta_\emptyset(s)$, $e_\emptyset(s)$ to all the sheets of the Riemann surface $\check{\mathcal{R}}$. The situation is less straightforward for sharp wedge and stationary initial conditions, where a natural interpretation is still missing for the coefficients $\Xi_x^\Delta$ weighting the Riemann surfaces $\mathcal{R}^\Delta$ covered by $\mathcal{R}$ in (6) and (11).

The stationary large deviations problem can be studied independently from the dynamics, using e.g. matrix product representations for discrete models [46,47]. This approach was recently exploited in [38] to express the factor $\theta(s)$ in (17) for general initial condition as the probability that a gas of infinitely many non-intersecting Brownian bridges with density $1/s$ stays under the graph of the initial condition $h(x,0) = h_0(x)$. More precisely, it was shown that

$$\theta(s) = \frac{\mathbb{P}(b_{-1} < h_0 | \ldots < b_{-2} < b_{-1})}{\mathbb{P}(b_{-1} < b_0 | \ldots < b_{-2} < b_{-1},\, b_1 < b_2 < \ldots)}, \tag{19}$$

where $b_j(x) - b_j(0)$, $j \in \mathbb{Z}$ are independent standard Brownian bridges with $b_j(0) = b_j(1)$, distances between consecutive endpoints $b_{j+1}(0) - b_j(0)$ are independent exponentially distributed random variables with parameter $s$, and $b_0(0) = h_0(0)$. Deriving exact formulas from the Brownian bridge representation is still an open problem, though, even for the simple initial conditions (flat, Brownian, sharp wedge) for which the result is known from Bethe ansatz, see however [48] for related work.

The idea that the contributions of the excited states of a theory should follow from that of the ground state by analytic continuation with respect to some parameter is not new, see for instance [49] for the quantum quartic oscillator, [50] for the Ising field theory on a circle (where the ground state energy is interestingly also given by an infinite sum of square roots, but with conjugate branch points paired), or [51] for models described by the thermodynamic Bethe ansatz. In the context of the Schrödinger equation for a particle in a potential, a unifying scheme appears to be exact WKB analysis [52], which uses tools from the theory of resurgent functions in order to reconstruct a single valued eigenfunction from the multivalued classical action. Such an approach might be useful for KPZ in order to derive known exact formulas without having to consider discrete models, by starting directly from the associated backward Fokker-Planck equation, a rather formal infinite dimensional linear partial differential equation acting on the functional space of allowed initial heights.

### 2.6.2 Particle-hole excitations

The finite sets of half-integers labelling the sheets of the Riemann surfaces $\check{\mathcal{R}}$, $\mathcal{R}^\Delta$ considered in this paper have a natural interpretation in terms of particle-hole excitations at both edges of a Fermi sea, see figure 2. This is most clearly seen on the expressions from [39] for the generating function $\langle e^{sh(x,t)} \rangle$ of the KPZ height discussed in sections 5.1.1 and 5.2.1.

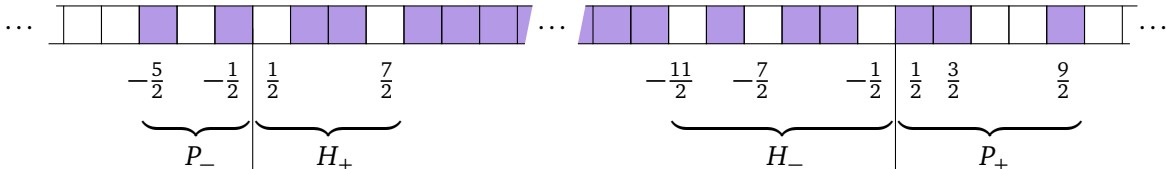

Figure 2: Picture of particle-hole excitations at both edges of the Fermi sea corresponding to sets $P$, $H$ of half-integers. The notations $P_\pm$, $H_\pm$ indicate the positive and negative elements of the sets.

From the exact Bethe ansatz solution of TASEP, eigenstates of the time evolution operator in the KPZ scaling regime are labelled by sets $P$ and $H$ corresponding to particle-hole excitations, interpreted respectively as momenta of quasiparticle and hole excitations relative to the Fermi momentum on both sides of the Fermi sea, see figure 2. Excitations only occur in particle-hole pairs, with "neutral charge": no particle or hole excitation alone occurs. The stationary state $P = H = \emptyset$ corresponds in particular to the completely filled Fermi sea.

For sharp wedge initial condition, the sets $P$ and $H$ must satisfy the constraint $|P|_\pm = |H|_\mp$ that the number of positive elements of $P$ is equal to the number of negative elements of $H$ and vice versa, see equations (131), (133). This corresponds to the fact that particle-hole excitations occur independently on both sides of the Fermi sea, i.e. quasiparticles at a finite distance from either side of the Fermi sea may be excited above the Fermi momentum but will stay at a finite distance of the same edge of the Fermi sea: excitation from one edge to the other (known as Umklapp processes in condensed matter physics) are suppressed for KPZ. It is remarkable that the constraints $|P|_\pm = |H|_\mp$ are automatically verified in (6), (11) from the way the collection of non-isomorphic Riemann surfaces $\mathcal{R}^\Delta$ are partitioned into sheets $\mathbb{C}_P$, with $H = P \ominus \Delta$ the symmetric difference of $P$ and $\Delta$ (union minus intersection), see section 5.2.1.

Flat initial condition (122), (123) corresponds to the special case $P = H$, where momenta of quasiparticles excitations on one side of the Fermi sea are identical to momenta of hole excitations on the other side of the Fermi sea. It is again quite remarkable that the resulting constraint $|P|_+ = |P|_-$ naturally appears for the sets $P$ labelling (half)-sheets of the Riemann surface $\check{\mathcal{R}}$, see figure 13 on the right. The extra constraint $P = H$ for flat initial condition is understood from TASEP as the fact that a specific microscopic state representing a flat interface has nonzero overlap only with Bethe eigenstates corresponding to particle-hole excitations satisfying the constraint [53]. This has the consequence to increase the spectral gap (i.e. to reduce the relaxation time) compared to a generic initial state, as was already recognized in [54].

Compared to the states contributing for flat initial condition, which have zero momentum, a non-empty symmetric difference $\Delta = P \ominus H$ corresponds to an imbalance between both sides of the Fermi sea, and is related through the coefficients $\Xi_x^\Delta$ in (7) to motion along the KPZ interface, with momentum $2\pi(\sum_{a \in P} a - \sum_{a \in H} a)$.

The interpretation of KPZ fluctuations in terms of particle-hole excitations close to the Fermi level is reminiscent of Luttinger liquid universality describing large scale dynamics of one-dimensional quantum fluids, with however several important distinctions. In addition to the absence mentioned above of Umklapp terms for KPZ, unlike in the Luttinger liquid setting, the dispersion relation, linear for the Luttinger liquid by construction after expanding around the Fermi level, is given for KPZ by $\kappa_a(v)^3 \sim |a|^{3/2}$ at large wave number $2\pi a$, indicating the existence of a singularity at the Fermi level. From a mathematical point of view, the difference amounts to the presence of polylogarithms with integer index (especially the dilogarithm $\mathrm{Li}_2$) for various quantities in the Luttinger liquid case, while half-integer polylogarithms appear for

KPZ.

### 2.6.3  KdV solitons

The exact formula for the probability $\mathbb{P}_{\text{flat}}(h(y,3t) > x)$ [3] with flat initial condition has a nice interpretation in terms of a solution to the Korteweg-de Vries (KdV) equation representing infinitely many solitons in interaction, see e.g. [55–57] for an introduction to classical non-linear integrable equations. As explained below, the relation to KdV suggests that the parameter $v$ appearing in various expressions in sections 2.2 to 2.5 should be interpreted as a moduli parameter for a class of degenerate hyperelliptic Riemann surfaces. The relation to KdV is most visible on the Fredholm determinant expression (22), (23) for $\mathbb{P}_{\text{flat}}(h(y,3t) > x)$, which has the same kind of Cauchy kernel (24) as KdV $N$-soliton $\tau$ functions with $N \to \infty$.

We recall that any determinant of the form $\tau(x,t) = \det(1 - M_N^{\text{KdV}}(x,t))$ where $M_N^{\text{KdV}}(x,t)$ is a $N \times N$ square matrix with matrix elements $M_N^{\text{KdV}}(x,t)_{a,b} = e^{2x\kappa_a + 2t\kappa_a^3 + \lambda_a}/(\kappa_a + \kappa_b)$ depending on $2N$ arbitrary coefficients $\kappa_a$, $\lambda_a$ is called a $\tau$ function for KdV, such that $u(x,t) = 2\partial_x^2 \log \tau(x,t)$ is a solution of the KdV equation

$$4\partial_t u = 6u\partial_x u + \partial_x^3 u \,. \tag{20}$$

Such a solution corresponds to $N$ solitons in interaction, the constants $\kappa_a$ determining the asymptotic velocities $-\kappa_a^2$ of the solitons when they are far away from each other, and the whole Cauchy determinant $\det(1 - M_N^{\text{KdV}}(x,t))$ describing how the solitons interact otherwise.

The KdV equation (20) belongs to a family of non-linear partial differential equations known as the KdV hierarchy. The $n$-th equation of the hierarchy, $n$ odd, involves derivatives with respect to the space variable $x$ and to a time variable $t_n$. The case $n = 3$ corresponds to (20), with $t_3 = t$. Soliton solutions to higher equations in the hierarchy are obtained by replacing $2t\kappa_a^3$ in $M_N^{\text{KdV}}(x,t)_{a,b}$ above by $2t_n\kappa_a^n$.

We observe that the probability $\mathbb{P}_{\text{flat}}(h(y,3t) > x)$ of the KPZ height with flat initial condition, independent of $y$, can be written as an integral over $v$ of a determinant (of Fredholm type, i.e. corresponding to an infinite dimensional operator) with a kernel of the same form as $M_\infty^{\text{KdV}}(x,t)$ above. Indeed, using (4), (64), (51), (184) and the Cauchy determinant identity

$$\det\left(\frac{1}{\kappa_a + \kappa_b}\right)_{a,b \in P} = \frac{\prod_{a > b \in P}(\kappa_a - \kappa_b)^2}{\prod_{a,b \in P}(\kappa_a + \kappa_b)} \,, \tag{21}$$

see section 5.1.2 for more details, one has

$$\mathbb{P}_{\text{flat}}(h(y,3t) > x) = \int_{c-i\pi}^{c+i\pi} \frac{dv}{2i\pi}\, \tau_{\text{flat}}(x,t;v) \,, \tag{22}$$

with the $\tau$ function defined by [4]

$$\tau_{\text{flat}}(x,t;v) = e^{3t\chi_\emptyset(v) - x\chi_\emptyset'(v) + I_0(v) + J_\emptyset(v)} \det(1 - M_{\text{flat}}(x,t;v)) \,, \tag{23}$$

where $\chi_\emptyset$ is defined in (57), $I_0$ in (68) and $J_\emptyset$ in (74). The operator $M_{\text{flat}}(x,t;v)$, acting on sequences indexed by $\mathbb{Z} + 1/2$, has the kernel

$$M_{\text{flat}}(x,t;v)_{a,b} = \frac{e^{2x\kappa_a(v) + 2t\kappa_a^3(v) + 2\int_{-\infty}^{v} dv \frac{\chi_\emptyset''(v)}{\kappa_a(v)}}}{\kappa_a(v)(\kappa_a(v) + \kappa_b(v))} \,, \tag{24}$$

---

[3] The change $(t,u,x) \to (3t,x,y)$ in this section is needed to conform to standard notations for KdV.

[4] The exponential of a linear function of $x$ in front of the Fredholm determinant does not contribute to $u_{\text{flat}} = 2\partial_x^2 \log \tau_{\text{flat}}$, which is thus still solution of the KdV equation.

with $\kappa_a(\nu)$ a specific branch of $\sqrt{4i\pi a - 2\nu}$ defined in (50). When $c < 0$, this is directly the result from [40], see equations (124), (125). When $c > 0$, a little more work is needed in order to rewrite into (22) the integral for $\nu$ between $c - i\infty$ and $c + i\infty$ of the Fredholm determinant in [39], which corresponds to a representation of $\check{\mathcal{R}}$ distinct from the one in (4), see section 5.1.1.

KPZ with flat initial condition thus involves a $\tau$ function for KdV, i.e. $u_{\text{flat}}(x, t; \nu) = 2\partial_x^2 \log \tau_{\text{flat}}(x, t; \nu)$ is a solution of the KdV equation (20) for any $\nu$. This solution corresponds to a gas of infinitely many solitons with (complex) velocities $-\kappa_a^2(\nu) = 2\nu - 4i\pi a$, $a \in \mathbb{Z} + 1/2$. The probability $\mathbb{P}_{\text{flat}}(h(y, 3t) > x)$ is obtained by averaging $\tau_{\text{flat}}(x, t; \nu)$ over the common velocity modulo $2i\pi$ of the solitons. Time variables for higher equations in the KdV hierarchy naturally appear in (24) as $t_{2m+1} = \chi_\emptyset^{(m+2)}(\nu)/(2m+1)!!$, see equation (187).

The possibility to interpret KPZ as a gas of solitons was put forward by Fogedby [58] starting with the WKB solution of the Fokker-Planck equation in the weak noise limit, with in particular the prediction of the dispersion relation $|k|^{3/2}$ as a function of momentum $k$, corresponding in our notations to $\kappa_a^3(\nu) \sim |a|^{3/2}$ for large $a$, see also [59] for recent related work.

Since flat initial condition corresponds to $h(y, 0) = 0$, the probability $\mathbb{P}_{\text{flat}}(h(y, 3t) > x)$ is expected to converge to $1_{\{x < 0\}}$ when $t \to 0$. Furthermore, since the KPZ height in finite volume at short time must have the same statistics as the KPZ fixed point on $\mathbb{R}$ [60], one should have $\mathbb{P}_{\text{flat}}(h(y, 4t) > -t^{1/3} x) \to F_1(x)$ when $t \to 0$, where $F_1$ is the GOE Tracy-Widom distribution from random matrix theory. This was checked numerically with good agreement in [39]. In terms of the KdV interpretation, we conjecture that the short time limit corresponds to the known scaling solution $(t/4)^{2/3} u(-(t/4)^{1/3} x, t/3) = V'(x) - V^2(x)$ of (20), where $V$ is a solution of the Painlevé II equation $V''(z) = 2V^3(z) + zV(z) + \alpha$ and $\alpha$ a constant which may depend on $\nu$, see e.g. [61].

The relation to KdV allows to interpret the integration variable $\nu$ in (1) as a moduli parameter for a class of singular hyperelliptic Riemann surfaces with infinitely many branch points. A soliton solutions of KdV can indeed be seen as the limit $\delta \to 0$ of a solution of KdV built in terms of the theta function of the hyperelliptic Riemann surface with branch points $0$, $\infty$, $\kappa_a^2 + \delta$, $\kappa_a^2 - \delta$ when branch points $\kappa_a^2 \pm \delta$ merge together on the Riemann surface, see [55, 56]. The $\infty$-soliton solution $u_{\text{flat}}$ corresponds in particular to the hyperelliptic Riemann surface with branch points $0$, $\infty$ and singular points $4i\pi a - 2\nu$, $a \in \mathbb{Z} + 1/2$, or equivalently to branch points $\nu$, $\infty$ and singular points $2i\pi a$, $a \in \mathbb{Z} + 1/2$. The Riemann surface $\mathcal{R}$ on which the parameter $\nu$ lives before taking the trace in (1) thus describes the monodromy of the branch point $\nu$ of the hyperelliptic Riemann surface above around the singular points $2i\pi a$.

For sharp wedge initial condition, the Fredholm determinant expressions for $\mathbb{P}_{\text{sw}}(h(2y, 3t) > x)$ from [39, 40] are instead reminiscent of soliton solutions for the Kadomtsev-Petviashvili (KP) equation $3\partial_y^2 u = \partial_x(4\partial_t u - (6u\partial_x u + \partial_x^3 u))$, a generalization of the KdV equation for a function $u(x, y, t)$ with two spatial dimensions, see e.g. [55–57]. The $\tau$ functions related to $u$ by $u(x, y, t) = 2\partial_x^2 \log \tau(x, y, t)$ and corresponding to $N$-soliton solutions of the KP equation are of the form $\tau(x, y, t) = \det(1 - M_N^{\text{KP}}(x, y, t))$, where the $N \times N$ matrix $M_N^{\text{KP}}(x, y, t)_{a,b} = e^{x(\kappa_a - \eta_b) + y(\kappa_a^2 - \eta_b^2) + t(\kappa_a^3 - \eta_b^3) + \lambda_a}/(\kappa_a - \eta_b)$ depends on $3N$ arbitrary constants $\kappa_a, \eta_b, \lambda_a$. For KPZ with sharp wedge initial condition, calculations similar to the ones leading to (22), see section 5.2.2 for more details, allow to rewrite (10) in terms of a Fredholm determinant as

$$\mathbb{P}_{\text{sw}}(h(2y, 3t) > x) = \int_{c-i\pi}^{c+i\pi} \frac{d\nu}{2i\pi} e^{t\chi_\emptyset(\nu) - u\chi_\emptyset'(\nu) + 2J_\emptyset(\nu)} \det(1 - M_{\text{sw}}(x, y, t; \nu)), \qquad (25)$$

with $M_{\text{sw}}(x, y, t; \nu) = L_{\text{sw}}(x, y, t; \nu)L_{\text{sw}}(x, -y, t; \nu)$ and

$$L_{\text{sw}}(x, y, t; \nu)_{a,b} = \frac{e^{x\kappa_a(\nu)+y\kappa_a^2(\nu)+t\kappa_a^3(\nu)+2\int_{-\infty}^{\nu} d\nu \frac{\chi_{\emptyset}''(\nu)}{\kappa_a(\nu)}}}{\kappa_a(\nu)(\kappa_a(\nu) + \kappa_b(\nu))} . \tag{26}$$

When $c < 0$, this is essentially the result from [40], see section 5.2.2. A similar Fredholm determinant was also given in [39] for $c > 0$, corresponding to a distinct representation of the Riemann surfaces in (10).

The dependency on $x$, $y$ and $t$ in (25) is essentially the same as for KP solitons, with $N \to \infty$, $\kappa_a = \kappa_a(\nu)$ and $\eta_b = -\kappa_b(\nu)$. The rest of the expression is similar to but different from the Cauchy determinant $M_N^{\text{KP}}$ required for KP solitons. Interestingly, proper $\infty$-soliton solutions of the KP hierarchy are known to appear for Laplacian growth [62], which belongs to a universality class of growing interfaces distinct from KPZ.

A paper by Quastel and Remenik [63] about KPZ fluctuations on $\mathbb{R}$ appeared shortly after our paper. There, the one-point cumulative distribution function with general initial condition is shown to be a $\tau$ function for the KP equation, without an extra integration like in (22), while multiple-point distributions at a given time correspond to a matrix generalization of KP. This suggest that there might still be a way to properly understand (25) in terms of KP solitons, maybe a matrix generalization such as the one in [63]. Additionally, the distinction between the solutions of Quastel-Remenik and ours for KPZ fluctuations is highly reminiscent of the one for the KdV / KP equations between solutions on the infinite line, where an extension to more singular initial conditions is required for KPZ [63], and quasi-periodic solutions involving compact Riemann surfaces, which appear to become non-compact for KPZ.

## 2.7 Conclusions

Several exact results for KPZ fluctuations with periodic boundaries have been reformulated in this paper in a compact way in terms of meromorphic differentials on Riemann surfaces related to polylogarithms with half-integer index. We believe that KPZ universality would benefit from a more systematic use of tools from algebraic geometry, especially more recent developments about non-compact Riemann surfaces of infinite genus [64]. Conversely, the very singular and universal nature of KPZ fluctuations suggests that objects appearing naturally for KPZ might also be of some interest in themselves for the field of algebraic geometry, especially when studying limits where the genus of Riemann surfaces goes to infinity.

A possible extension concerns the renormalization group flow $h_\lambda(x, t)$ from the equilibrium fixed point $\lambda \to 0$ to the KPZ fixed point $\lambda \to \infty$ considered in this paper. Whether the dynamics for finite $\lambda$ may also be expressed in a natural way in terms of Riemann surfaces is unclear at the moment. Hints of a duality [65] between the equilibrium fixed point in an infinite system and the KPZ fixed point for periodic boundaries, with half-integer polylogarithms describing large deviations on both sides [66], suggest however the existence of a tight structure holding everything together. Partial exact results relevant to finite $\lambda$ with periodic boundaries have been obtained using the replica solution [37] of the KPZ equation and a weakly asymmetric exclusion process [67,68] (see also [69,70] for recent exact results with arbitrary asymmetry). The appearance of half-integer polylogarithms and $\zeta$ functions in related contexts of non-intersecting lattice paths [71], largest eigenvalues in the real Ginibre ensemble [72,73] and return probabilities for the symmetric exclusion process [74] and quantum spin chains [75] on $\mathbb{Z}$ with domain wall initial condition might also have some connections to the equilibrium side of the duality.

Finally, the results of this paper are based on complicated asymptotics of Bethe ansatz formulas for TASEP in the limit where the number of lattice sites $L$ and the number of particles $N$ go to infinity with fixed density $N/L$ [39–42]. A natural question is whether TASEP with

finite $L$, $N$ can already be described in terms of (finite genus) Riemann surfaces, so that the infinite genus Riemann surface $\mathcal{R}$ would emerge in a more transparent fashion in the large $L$, $N$ limit. Tools from algebraic geometry have already been used in the study of the more complicated Bethe equations for the asymmetric exclusion process with hopping in both directions [76] and the related XXZ spin chains with finite anisotropy [77, 78]. The limit where the anisotropy of the spin chain goes to infinity, corresponding for the exclusion process to the TASEP limit, seems however a better starting point since the Bethe equations have a much simpler structure in that case, see e.g. [79, 80] for related works.

## 3 Riemann surfaces and ramified coverings

In this section, we recall a few classical results about (compact) Riemann surfaces and ramified coverings. The various properties are illustrated using two examples: hyperelliptic Riemann surfaces $\mathcal{H}_N$, which are the proper domain of definition for square roots of polynomials, and Riemann surfaces $\mathcal{R}_N$ defined from sums of square roots, which have the topology of $N-1$-dimensional hypercubes. The Riemann surfaces $\mathcal{R}_N$ are finite genus analogues of the non-compact Riemann surfaces $\mathcal{R}$ introduced in section 3.8 and used for KPZ fluctuations in section 2. We refer to [81–83] for good self-contained introductions to compact Riemann surfaces and ramified coverings.

### 3.1 Analytic continuation and Riemann surfaces

Let us consider a function $g_0$ analytic in the complex plane $\mathbb{C}$ except for the existence of branch cuts, i.e. paths in $\mathbb{C}$ across which $g_0$ is discontinuous. Extremities of branch cuts, called branch points, correspond to genuine singularities of the function $g_0$. The branch cuts themselves, on the other hand, are somewhat arbitrary. The domain of definition of $g_0$ can be extended by analytic continuation along paths crossing the branch cuts. In the favourable case considered in this paper, successive iterations of this procedure lead to functions $g_i$, $i \in I$ analytic in $\mathbb{C}$ except for the same branch cuts as $g_0$, such that the function $g_i$ on one side of a branch cut continues analytically to another function $g_j$ on the other side of the same branch cut. The collection of all branches $g_i$ represents a multivalued function. Multivalued basic special functions usually come with a standard choice for the principal value $g_0$.

Considering the domains of definition of the functions $g_i$ as distinct copies $\mathbb{C}_i$ of the complex plane [5], the Riemann surface $\mathcal{M}$ for the function $g_0$ is built by gluing together the *sheets* $\mathbb{C}_i$ along branch cuts, and we use the notation $[z, i]$, $z \in \mathbb{C}$ for the points on the sheet $\mathbb{C}_i$ of $\mathcal{M}$. More precisely, let $z_*$ be a branch point of $g_i$ and $\gamma$ a branch cut issued from $z_*$. Calling by "left" and "right" the two sides of $\gamma$, we glue the left side of the cut in the sheet $\mathbb{C}_i$ to the right side of the cut in the sheet $\mathbb{C}_j$ if $g_i$ is analytically continued from the left to $g_j$ across the cut, and we glue the right side of the cut in the sheet $\mathbb{C}_i$ to the left side of the cut in the sheet $\mathbb{C}_k$ if $g_i$ is analytically continued from the right to $g_k$ across the cut. Additionally, the branch points $[z_*, i]$, $[z_*, j]$, $[z_*, k]$ of the sheets $\mathbb{C}_i$, $\mathbb{C}_j$, $\mathbb{C}_k$ represent a single point on $\mathcal{M}$, $[z_*, i] = [z_*, j] = [z_*, k]$. The Riemann surface $\mathcal{M}$ is independent of the precise choice of branch cuts for $g_0$: the branch cuts only determine a partition of $\mathcal{M}$ into sheets $\mathbb{C}_i$, and the notation $[z, i]$ for the points of $\mathcal{M}$ thus depends implicitly on the choice of branch cuts.

---

[5]For the sake of simplicity, we consider in this paper *concrete Riemann surfaces*, defined in terms of analytic continuation of functions and close to Riemann's original presentation, and not the more abstract modern formalism in terms of an atlas of charts and transition functions. All the Riemann surfaces considered in this paper can be understood as the natural domain of definition of some explicit multivalued function, and thus come with a natural ramified covering from the Riemann surface to $\mathbb{C}$.

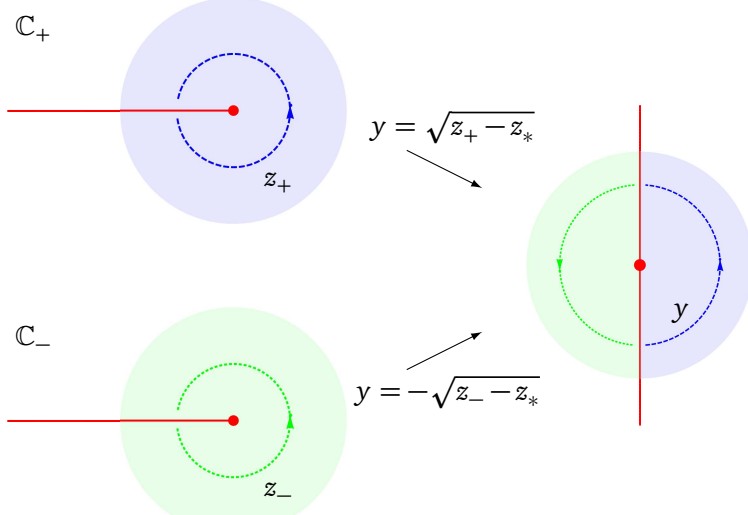

Figure 3: Neighbourhood of a point $q = [z_*, \mathbb{C}_+] = [z_*, \mathbb{C}_-]$ in a Riemann surface (right) such that $z_*$ is a branch point of the function $g_0$ from which the Riemann surface is built. The neighbourhood is formed by gluing together along the branch cut originating from $q$ two half-disks obtained from taking the square root of full disks from the sheets $\mathbb{C}_\pm$ (left). The complex numbers $z_\pm$ parametrize half a neighbourhood of $q$ in $\mathbb{C}_\pm$. The local parameter $y$ at $q$ is a complex number that fully parametrizes the neighbourhood of $q$.

A function $g$ can then be defined on the Riemann surface $\mathcal{M}$ by $g([z, i]) = g_i(z)$. Locally, the neighbourhood of any point of $\mathcal{M}$ looks like an open disk of $\mathbb{C}$, and the function $g$ is analytic there. This is obvious by construction, except around branch points where one needs to introduce a non-trivial local coordinate $y$ to parametrize the neighbourhood. We mainly consider in the following branch points $z_*$ of square root type, such that $g_i(z) \simeq \tilde{g}_i(z)\sqrt{z - z_*}$ when $z \to z_*$, where the $\tilde{g}_i$ are analytic and with a branch cut for the square root determined by the branch cuts of the $g_i$. A possible local parameter is then $y = \sqrt{z - z_*}$, and $g_i(z) \simeq y\,\tilde{g}_i(z_* + y^2)$ is indeed analytic around $y = 0$. A neighbourhood of $z_*$ in $\mathcal{M}$ may be built using the local parameter $y$ by gluing together two half-disks as in figure 3.

All the construction goes through in the presence of isolated poles, with analytic functions replaced by meromorphic functions. Additionally, it is often convenient to make Riemann surfaces compact by adding the points at infinity of the sheets, with appropriate local parameters ensuring that the neighbourhoods of these points are regular. In the simplest case where $g_0$ is a rational function without branch points and $\mathcal{M}$ is thus made of a single sheet $\mathbb{C}$, the associated compact Riemann surface is called [6] the Riemann sphere $\widehat{\mathbb{C}} = \mathbb{C} \cup \{\infty\}$.

## 3.2 The Riemann surfaces $\mathcal{H}_N$ and $\mathcal{R}_N$

We consider in this section two concrete examples of the construction above. Let $z_1, \ldots, z_N$ be distinct complex numbers, that are fixed in the following. We choose for simplicity of the pictures (and also because it will be the case of interest for KPZ) the $z_j$'s to be purely imaginary and equally spaced, $\mathrm{Im}\, z_1 < \ldots < \mathrm{Im}\, z_N$, but this choice is not essential here. We define the functions

$$h_+(z) = \sqrt{z - z_1} \times \ldots \times \sqrt{z - z_N} \tag{27}$$

---

[6] Other notations for $\widehat{\mathbb{C}}$ include $\overline{\mathbb{C}}$, $\mathbb{C}_\infty$, or $\mathbb{P}^1(\mathbb{C})$, $\mathbb{CP}^1$ when interpreted as the complex projective line.

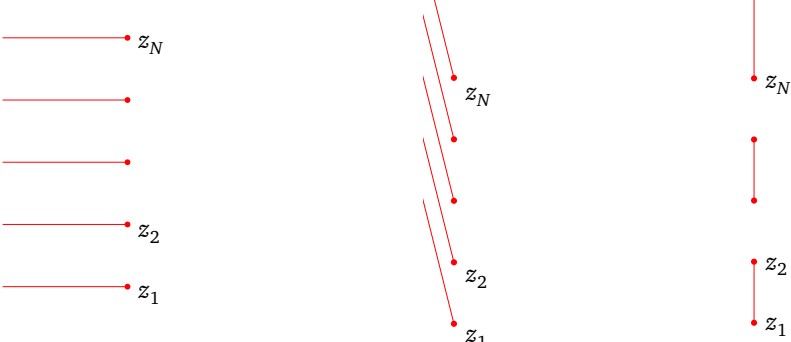

Figure 4: Three different choices of branch cuts (solid lines) for the function $h_+$ defined in (27) with $N = 5$. The branch points $z_j$ are represented by dots. The function $h_+$ is multiplied by $-1$ after crossing any branch cut.

and

$$f_\emptyset(z) = \sqrt{z - z_1} + \ldots + \sqrt{z - z_N} \tag{28}$$

of a complex variable $z$, which inherit branch cuts from the square roots (see respectively figures 4 and 5 for some possible choices of branch cuts). The functions $h_+$ and $f_\emptyset$ can be extended by the procedure described in the previous section to analytic functions $h$ and $f$ defined on compact Riemann surfaces $\mathcal{H}_N$ and $\mathcal{R}_N$. The Riemann surface $\mathcal{H}_N$, called hyperelliptic and used here mainly for illustrative purpose, has links to the KdV equation discussed in section 2.6.3. The Riemann surface $\mathcal{R}_N$, on the other hand, is a simplified, finite genus version of the Riemann surface $\mathcal{R}$ introduced in section 3.8, and in terms of which KPZ fluctuations are expressed in section 2.

We begin with the function $h_+$ defined in (27), analytic on a sheet called $\mathbb{C}_+$, with the choice of branch cuts on the left in figure 4. Analytic continuation across branch cuts gives $h_- = -h_+$, which lives on another sheet $\mathbb{C}_-$. The Riemann surface $\mathcal{H}_N$ is formed by the two sheets $\mathbb{C}_\pm$ glued together, and $h_+$ extends analytically to a function $h$ defined on $\mathcal{H}_N$ by $h([z, \pm]) = h_\pm(z)$. Locally, the neighbourhood of any point of $\mathcal{H}_N$ looks like an open disk of $\mathbb{C}$ (see figure 3 for the neighbourhood of $[z_j, \pm]$), and the function $h$ is analytic there. The points at infinity $[\infty, \pm]$ are poles of the function $h$. A local parameter $y$ for these points is $y = z^{-1}$ if $N$ is even and $y = z^{-1/2}$ if $N$ is odd. In the former case, the poles of $h$ at $[\infty, +]$ and $[\infty, -]$, which are distinct points of $\mathcal{H}_N$, are of order $N/2$. In the latter case, $\infty$ is a branch point of $h_\pm$ and the point $[\infty, +] = [\infty, -]$ is a pole of order $N$ of $h$. The function $h$ also has $N$ zeroes, the points $[z_j, +] = [z_j, -]$, $j = 1, \ldots, N$. The total number of poles of $h$, counted with multiplicity, is thus equal to its number of zeroes. This is in fact a general property valid for any meromorphic function on a compact Riemann surface.

Compared to the hyperelliptic case discussed above, analytic continuations across branch cuts of $f_\emptyset$ defined in (28) have a richer structure, since all the square roots are independent: each one may change sign independently across branch cuts. The corresponding Riemann surface $\mathcal{R}_N$ is thus made of $2^N$ sheets labelled by sets of integers between 1 and $N$, $P \subset [\![1, N]\!]$, indicating the square roots coming with a minus sign. It will be convenient in the following to distinguish two systems of sheets $\mathcal{G}_P$ and $\mathcal{F}_P$ partitioning $\mathcal{R}_N$, corresponding respectively to the choice of branch cuts on the left and on the right in figure 5. The points of $\mathcal{R}_N$ will be written as $[z, \mathcal{G}_P]$ or $[z, \mathcal{F}_P]$ when specifying the choice of sheets is needed, and simply as $[z, P]$ otherwise. The analytic function $f$ on $\mathcal{R}_N$ induced by $f_\emptyset$ is defined by $f([z, P]) = f_P(z)$,

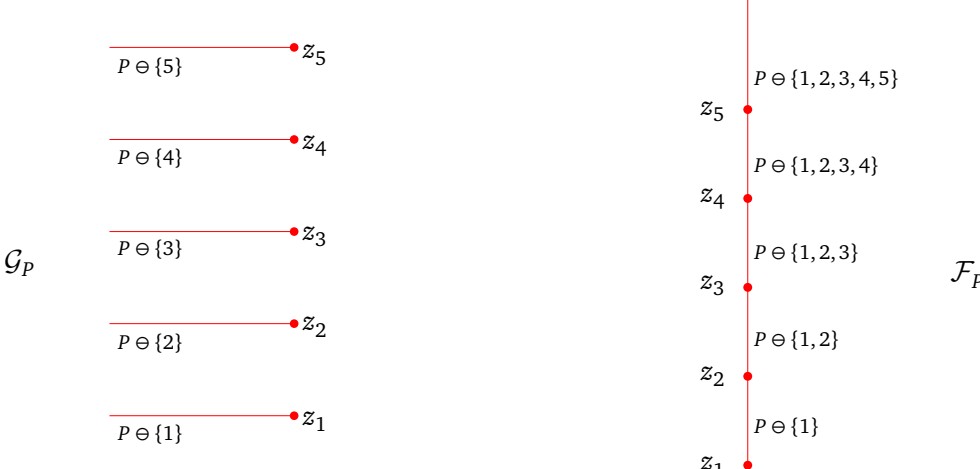

Figure 5: Two different choices of branch cuts (solid lines) for the function $f_\emptyset$ defined in (28) with $N = 5$. The branch points are represented by dots. The sets labelling the sheet reached after crossing branch cuts from either side starting from the sheet labelled by $P \subset \{1, 2, 3, 4, 5\}$ is indicated near the branch cuts.

with $f_P(z) = \sum_{j=1}^{N} \sigma_j(P) \sqrt{z - z_j}$ and

$$\sigma_a(P) = \begin{cases} 1 & a \notin P \\ -1 & a \in P \end{cases}. \tag{29}$$

The number of square roots that have changed sign in $f_P$ compared to $f_\emptyset$ is equal to the number $|P|$ of elements in $P$.

The connectivity of the sheets $\mathcal{G}_P$ and $\mathcal{F}_P$ in $\mathcal{R}_N$ following from analytic continuation can be expressed in terms of the symmetric difference operator $\ominus$, defined as union minus intersection:

$$P \ominus Q = (P \cup Q) \backslash (P \cap Q). \tag{30}$$

The symmetric difference operator $\ominus$ is associative, commutative and verifies $P \ominus P = \emptyset$ and [7] $P \ominus Q + n = (P + n) \ominus (Q + n)$ for $n \in \mathbb{Z}$. A collection of sets closed under union, intersection and complement forms a group with the operation $\ominus$. The identity element is the empty set $\emptyset$, and the maps $\sigma_a$ act as group homomorphisms, $\sigma_a(P \ominus Q) = \sigma_a(P) \sigma_a(Q)$.

Crossing the branch cut associated to $z_j$ from the sheet $\mathcal{G}_P$ leads to $\mathcal{G}_{P \ominus \{j\}}$ (see figure 5), and one has the local parameters $y = \pm \sqrt{z - z_j}$ with the same half-disk construction of figure 3 as for $\mathcal{H}_N$ around the points $[z_j, \mathcal{G}_P] = [z_j, \mathcal{G}_{P \ominus \{j\}}]$. In the sheet $\mathcal{F}_P$ on the other hand, crossing the branch cut between $z_j$ and $z_{j+1}$ (with $z_{N+1} = \infty$) leads to $\mathcal{F}_{P \ominus [\![1,j]\!]}$ (see figure 5). There, an additional difficulty for constructing local parameters is that the branch points $z_j$, $j = 2, \ldots, N-1$ lie on branch cuts, and must thus be labelled by an additional index l or r depending on whether the point is on the left side (l) or the right side (r) of the cut. A neighbourhood of $[(z_j)_r, \mathcal{F}_P] = [(z_j)_l, \mathcal{F}_{P \ominus [\![1,j]\!]}] = [(z_j)_r, \mathcal{F}_{P \ominus \{j\}}] = [(z_j)_l, \mathcal{F}_{P \ominus [\![1,j-1]\!]}]$ is constructed in figure 6 by gluing quarter disks together.

The points at infinity are branch points of the functions $f_P$. Considering the partition of $\mathcal{R}_N$ with sheets $\mathcal{F}_P$, one has $[\infty, \mathcal{F}_P] = [\infty, \mathcal{F}_{P \ominus [\![1,N]\!]}]$, see figures 5 and 7 right. For the sheets $\mathcal{G}_P$, points at infinity can be reached from $N$ directions in figure 5 left, and one has to distinguish points $[\infty_j, \mathcal{G}_P]$, $j = 1, \ldots, N$, with the identifications $[\infty_j, \mathcal{G}_P] = [\infty_{j-1}, \mathcal{G}_{P \ominus \{j\}}]$, see figure 7 left.

---

[7]We choose symmetric difference to have precedence over addition, so that $P \ominus Q + n$ means $(P \ominus Q) + n$.

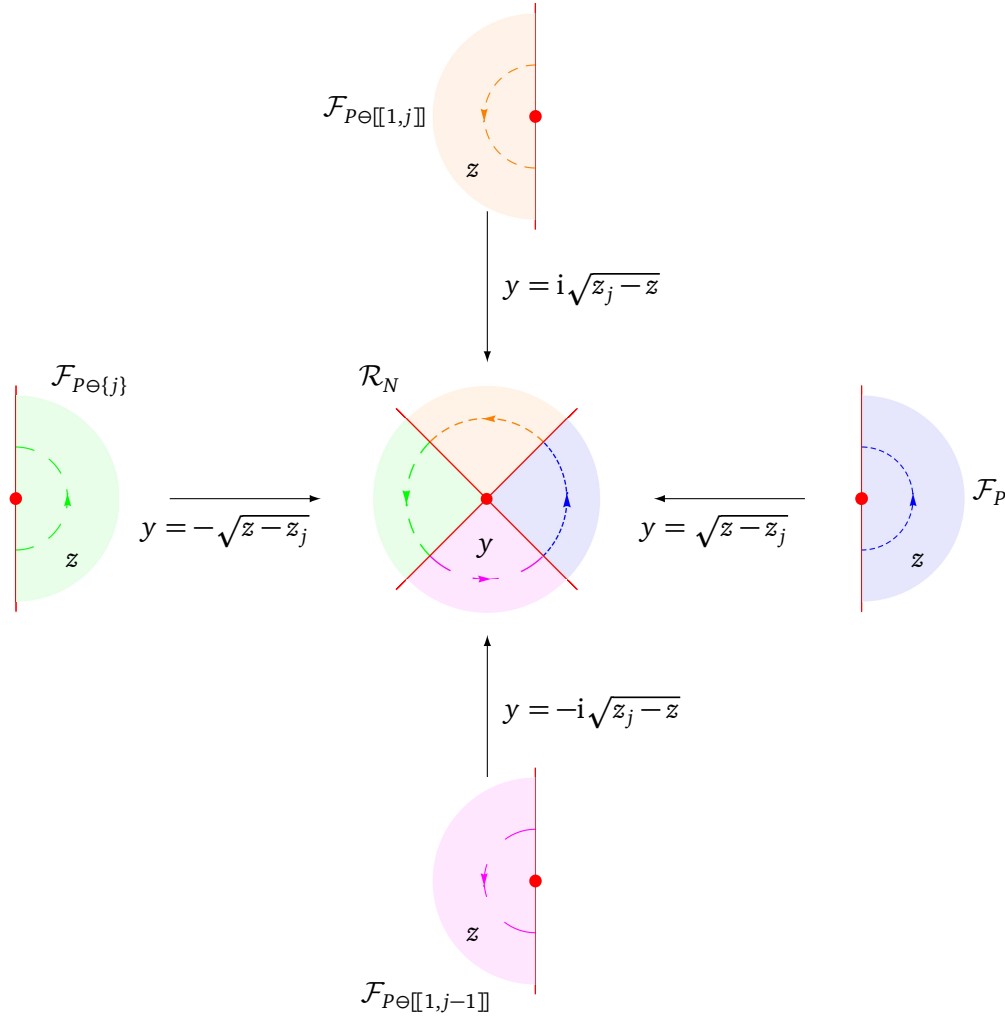

Figure 6: Neighbourhood of $q = [(z_j)_\mathrm{r}, \mathcal{F}_P] = [(z_j)_\mathrm{l}, \mathcal{F}_{P\ominus[\![1,j]\!]}] = [(z_j)_\mathrm{r}, \mathcal{F}_{P\ominus\{j\}}]$ $= [(z_j)_\mathrm{l}, \mathcal{F}_{P\ominus[\![1,j-1]\!]}]$, $2 \le j \le N-1$ in $\mathcal{R}_N$, formed by gluing four quarter-disks obtained from taking the square root of half-disks in the sheets $\mathcal{F}_P$, $\mathcal{F}_{P\ominus[\![1,j]\!]}$, $\mathcal{F}_{P\ominus\{j\}}$ and $\mathcal{F}_{P\ominus[\![1,j-1]\!]}$. The complex numbers $z$ parametrize quarters of neighbourhoods of $q$ in the various sheets. The local parameter $y$ fully parametrizes a neighbourhood of $q$, with $y = 0$ corresponding to $q$.

### 3.3 Genus

From a purely topological point of view, a Riemann surface is a two-dimensional connected manifold. In the case of a closed (i.e. without boundary), compact Riemann surface such as $\widehat{\mathbb{C}}$, $\mathcal{H}_N$ or $\mathcal{R}_N$ above, the manifold is fully characterized up to homeomorphisms (i.e. continuous deformations with continuous inverse) by a single non-negative integer, its genus $g$, corresponding to the maximal number of simple non-intersecting closed curves along which the manifold can be cut while still being connected. The case $g = 0$ corresponds to a sphere, $g = 1$ to a torus and $g \ge 2$ to a chain of $g$ tori glued together, see figure 8.

   The additional complex structure of Riemann surfaces detailing how sheets are glued together gives more freedom, and two Riemann surfaces of the same genus are not necessarily isomorphic (i.e. there may not exist a holomorphic homeomorphism with holomorphic inverse transforming one into the other). The genus 0 case is an exception, for which a single Riemann surface exists up to isomorphism, the Riemann sphere $\widehat{\mathbb{C}}$. For genus 1, the equivalence

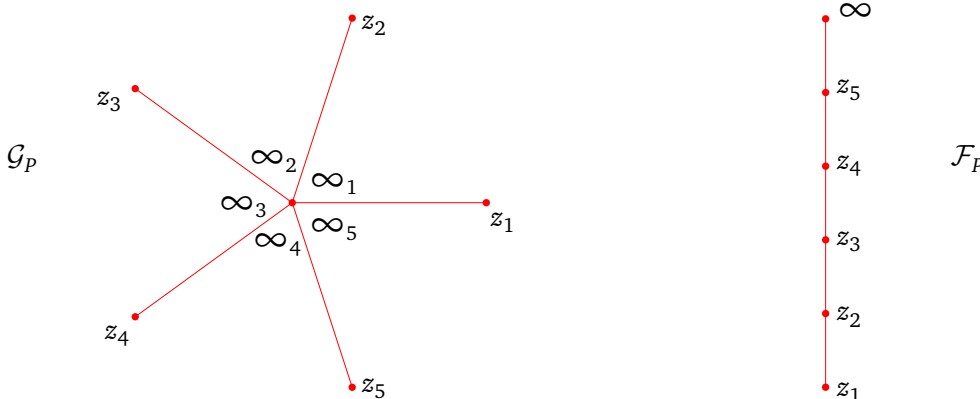

Figure 7: Compact representation of branch cuts for the sheets $\mathcal{G}_P$ (left) and $\mathcal{F}_P$ (right) of $\mathcal{R}_5$ after adding the points at infinity. The branch cuts are represented as straight lines for clarity, and distances are not meaningful.

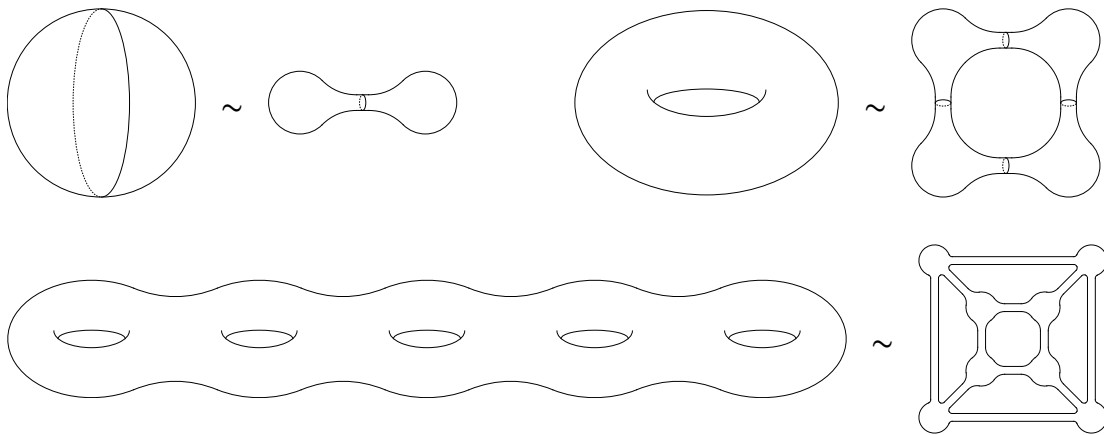

Figure 8: Sphere, torus, and surface with genus $g = 5$, along with hypercubes of dimensions 1, 2, 3 made of spheres connected with cylinders that can be mapped to them by continuous deformations.

classes up to isomorphism are indexed by a single complex number $\tau$ (defined up to modular transformations), such that the parallelogram with vertices $0, 1, 1 + \tau, \tau$ becomes a torus when opposite sides are glued together. For higher genus $g \geq 2$, the moduli space of all Riemann surfaces is parametrized by $3g - 3$ complex parameters.

The Riemann surfaces $\mathcal{H}_1$ and $\mathcal{H}_2$ are both isomorphic to the Riemann sphere, while $\mathcal{H}_3$ and $\mathcal{H}_4$ are tori. More generally, the genus of the hyperelliptic Riemann surface $\mathcal{H}_N$ is known to be equal to either $(N-1)/2$ or $(N-2)/2$ depending on the parity of $N$. Similarly, $\mathcal{R}_1$ (which is exactly the same as $\mathcal{H}_1$ since $f_\emptyset = h_+$ when $N = 1$) and $\mathcal{R}_2$ (see figure 9) are also isomorphic to the Riemann sphere, while $\mathcal{R}_3$ is a torus, see figure 10. More generally, the genus $g_N$ of $\mathcal{R}_N$ grows exponentially fast with $N$, as

$$g_N = 1 + (N-3)2^{N-2} \,, \tag{31}$$

which can be proved by induction on $N$. Indeed, cutting $\mathcal{R}_N$ along a path joining $[z_N, \mathcal{F}_P]$ and $[\infty, \mathcal{F}_P]$ in every sheet $\mathcal{F}_P$ splits $\mathcal{R}_N$ into two disconnected pieces (corresponding to sheets $\mathcal{F}_P$ with either $N \in P$ or $N \notin P$), each component having $2^{N-1}$ boundaries corresponding to the cycles $[\infty, \mathcal{F}_P] = [\infty, \mathcal{F}_{P \ominus [\![1,N]\!]}] \to [(z_N)_\mathrm{l}, \mathcal{F}_{P \ominus [\![1,N]\!]}] = [(z_N)_\mathrm{r}, \mathcal{F}_P] \to [\infty, \mathcal{F}_P]$. Both pieces are homeomorphic to $\mathcal{R}_{N-1}$ with $2^{N-1}$ boundaries, which are connected two by two

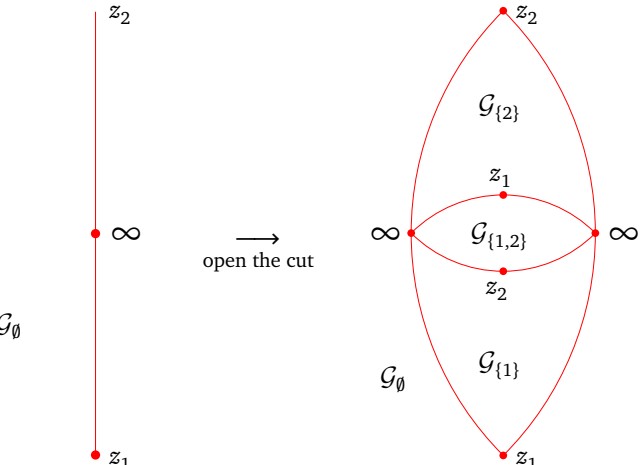

Figure 9: Representation of the surface corresponding to the Riemann surface $\mathcal{R}_2$. The sheet $\mathcal{G}_{\emptyset}$ is represented on the left, with a cut linking the points $[z_1, \mathcal{G}_{\emptyset}]$, $[z_2, \mathcal{G}_{\emptyset}]$ and $[\infty, \mathcal{G}_{\emptyset}]$. Opening the cut, all the other sheets $\mathcal{G}_{\{1\}}$, $\mathcal{G}_{\{2\}}$, $\mathcal{G}_{\{1,2\}}$ fit within the opening. The graph made by the cuts of all sheets is planar, and $\mathcal{R}_2$ is isomorphic to the Riemann sphere $\widehat{\mathbb{C}}$.

in $\mathcal{R}_N$. This leads to the recurrence relation $g_{N+1} = 2g_N + 2^{N-1} - 1$, since gluing together both pieces along a first boundary leads to a surface with twice the genus of $\mathcal{R}_{N-1}$, and each additional gluing adds a handle to the surface and hence increases the genus by 1. We observe that the Riemann surface $\mathcal{R}_N$ is thus homeomorphic to a $N-1$-dimensional hypercube [84] whose nodes are spheres and edges cylinders connecting the spheres, see figure 8.

### 3.4 Ramified coverings

Maps between Riemann surfaces acting as holomorphic functions on local parameters, called holomorphic maps, are a powerful tool in the study of Riemann surfaces. They allow in particular to define a notion of equivalence between Riemann surfaces having essentially the same complex structure: two Riemann surfaces are called isomorphic when there exists a bijective holomorphic map with holomorphic inverse between them.

Given two Riemann surfaces $\mathcal{M}$ and $\mathcal{N}$, a ramified covering (or branched covering, or simply covering map [8] here for simplicity) such that $\mathcal{M}$ covers $\mathcal{N}$ (or $\mathcal{N}$ is covered by $\mathcal{M}$) is a non-constant holomorphic map from $\mathcal{M}$ to $\mathcal{N}$, which is then surjective by analyticity. Ramified coverings allow to relate complicated Riemann surfaces to simpler Riemann surfaces. In particular, for any Riemann surface $\mathcal{M}$, there exists at least one ramified covering from $\mathcal{M}$ to the Riemann sphere $\widehat{\mathbb{C}}$, i.e. a non-constant meromorphic function on $\mathcal{M}$. Conversely, defining a concrete Riemann surface $\mathcal{M} = \{[z, i], z \in \widehat{\mathbb{C}}, i \in I\}$ by gluing domains of definition of branches $i \in I$ of a multivalued function gives the natural covering map $[z, i] \mapsto z$.

Let $\rho : \mathcal{M} \to \mathcal{N}$ be a branched covering and $p \in \mathcal{M}$. Then, there exists a unique positive integer $e_p \in \mathbb{N}^*$ such that one can choose local coordinates $y$ and $z$ for the neighbourhoods around $p$ and $\rho(p)$ in such a way that $z = y^{e_p}$. The integer $e_p$ is called the ramification index of the point $p$ for the covering map $\rho$. A ramification point of $\rho$ is then a point $p \in \mathcal{M}$ such that $e_p \geq 2$, with $e_p = 2$ corresponding to ramification points of square root type. Around such points, the mapping $\rho$ is not injective, and the inverse function $\rho^{-1}$ is multivalued. The branch points $q \in \mathcal{N}$ of $\rho$ are the images by $\rho$ of the ramification points $p \in \mathcal{M}$.

---

[8]Not to be confused with the topological notion of a covering, which does not have ramification points.

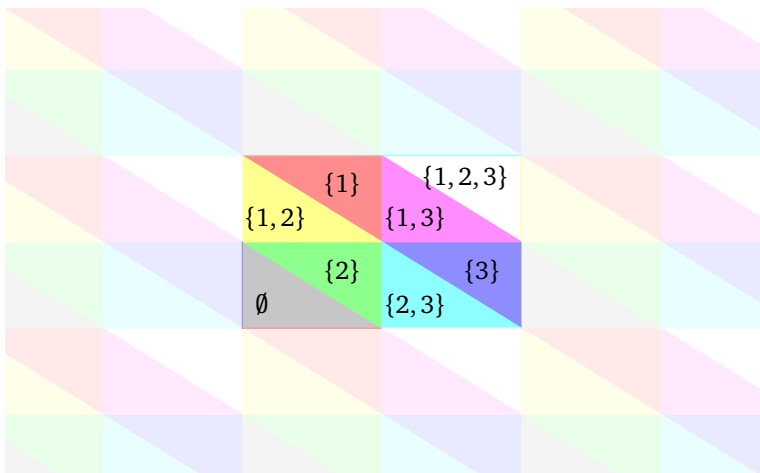

Figure 10: Representation of the torus homeomorphic to the Riemann surface $\mathcal{R}_3$. Opposite sides of the fundamental domain represented by the region in the middle are identified. The connectivity of the sheets $\mathcal{G}_P$ partitioning $\mathcal{R}_3$ is represented by the labels $P \subset \{1, 2, 3\}$ of the sheets.

Ramification points form a discrete subset of $\mathcal{M}$. Furthermore, if $\mathcal{M}$ is compact, the number of branch points is finite, and so is the number of preimages of any $q \in \mathcal{N}$ by $\rho$. In that case, there exists a unique positive integer $d \in \mathbb{N}^*$, the degree of the branched covering, such that for any $q \in \mathcal{N}$, $d = \sum_{p \in \rho^{-1}(q)} e_p$. For generic points $q \in \mathcal{N}$ not branch points of $\rho$, the set of preimages $\rho^{-1}(q) \subset \mathcal{M}$ has exactly $d$ distinct elements.

The construction of a concrete Riemann surface $\mathcal{M}$ by gluing together a discrete number of copies $\widehat{\mathbb{C}}_i$, $i \in I$ of the Riemann sphere along branch cuts of some function, which we used above to define $\mathcal{H}_N$ and $\mathcal{R}_N$, naturally gives the ramified covering $[z, i] \mapsto z$ from $\mathcal{M}$ to $\widehat{\mathbb{C}}$.

New meromorphic function can be built from known ones using ramified coverings. Let $\rho : \mathcal{M} \to \mathcal{N}$ be a branched covering and $\varphi_{\mathcal{M}}$, $\varphi_{\mathcal{N}}$ meromorphic functions defined respectively on $\mathcal{M}$ and $\mathcal{N}$. Then, the composition $\varphi_{\mathcal{N}} \circ \rho$ is a meromorphic function on $\mathcal{M}$. For instance, the ramified covering $\rho : [z, P] \mapsto [z, (-1)^{|P|}]$ from $\mathcal{R}_N$ to $\mathcal{H}_N$ generates from the function $h$ on $\mathcal{H}_N$ defined from (27) the function $h \circ \rho$ meromorphic on $\mathcal{R}_N$, which is essentially the same function as $h$ but defined on a bigger space: $\mathcal{H}_N$ is only the "minimal" closed, compact Riemann surface on which $h_\pm$ can be extended to a meromorphic function.

Conversely, tracing over preimages of $\rho$ defines a function $\mathrm{tr}_\rho \, \varphi_{\mathcal{M}}$ as

$$(\mathrm{tr}_\rho \, \varphi_{\mathcal{M}})(q) = \sum_{\substack{p \in \mathcal{M} \\ \rho(p) = q}} \varphi_{\mathcal{M}}(p) \,, \tag{32}$$

which can be shown [85] to be meromorphic on $\mathcal{N}$. This can be illustrated by considering the covering map $\rho : [z, P] \mapsto z$ from $\mathcal{R}_N$ to $\widehat{\mathbb{C}}$. Starting with the function $f$ meromorphic on $\mathcal{R}_N$ defined from (28), all the square roots cancel in the trace and one has $\mathrm{tr}_\rho \, f = 0$ which is indeed defined on $\widehat{\mathbb{C}}$. Less trivially, allowing essential singularities at infinity, the function $(\mathrm{tr}_\rho \, e^{\lambda f})(z) = 2^N \prod_{j=1}^N \cosh(\lambda \sqrt{z - z_j})$ is also analytic in $\mathbb{C}$.

The Riemann-Hurwitz formula gives a relation between ramification indices $e_p$ for a branched covering $\rho : \mathcal{M} \to \mathcal{N}$ of degree $d$ and the respective genus $g_{\mathcal{M}}$, $g_{\mathcal{N}}$ of the Riemann surfaces $\mathcal{M}$ and $\mathcal{N}$:

$$g_{\mathcal{M}} = d(g_{\mathcal{N}} - 1) + 1 + \frac{1}{2} \sum_{p \in \mathcal{M}} (e_p - 1) \,, \tag{33}$$

where only ramification points contribute to the sum. In particular, one has always $g_{\mathcal{M}} \geq g_{\mathcal{N}}$. Considering a triangulation of $\mathcal{M}$ with vertices at ramification points of $\rho$, the Riemann-Hurwitz formula is a simple consequence of the expression for the Euler characteristics $\chi = 2 - 2g$ in terms of the number of vertices, edges and faces of the triangulation.

The Riemann-Hurwitz formula allows one to recover the expression (31) for the genus of $\mathcal{R}_N$. We introduce the ramified covering $[z, P] \mapsto z$ from $\mathcal{R}_N$ to $\widehat{\mathbb{C}}$ for some choice of branch cuts. This covering has degree $d = 2^N$, and its ramification points, all with ramification index 2, are the $[z_j, P]$ and $[\infty, P]$. Each one is shared between two sheets (or four half-sheets, compare figures 3 and 6), so that the total number of ramification points is equal to $(N+1)2^{N-1}$. Taking $g_{\mathcal{N}} = 0$ in (33) since the target space is $\widehat{\mathbb{C}}$ gives again (31).

## 3.5 Quotient under group action

Quotients of Riemann surfaces by the action of their holomorphic automorphisms (i.e. bijective holomorphic maps from the Riemann surface to itself) generate new Riemann surfaces. Let $\mathcal{M}$ be a Riemann surface and $\mathfrak{h}$ a group of holomorphic automorphisms of $\mathcal{M}$ acting properly discontinuously on $\mathcal{M}$, i.e. for any point $p \in \mathcal{M}$, there exists a neighbourhood $U$ of $p$ in $\mathcal{M}$ such that the set $\{h \in \mathfrak{h}, h U \cap U \neq \emptyset\}$ is finite, with $h U = \{h q, q \in U\}$. The quotient $\mathcal{M}/\mathfrak{h}$ is then also a Riemann surface, whose points $q \in \mathcal{M}/\mathfrak{h}$ are identified to orbit of $p \in \mathcal{M}$ under the action of $\mathfrak{h}$, and the covering map $p \mapsto q$ from $\mathcal{M}$ to $\mathcal{M}/\mathfrak{h}$ is ramified at the fixed points of $\mathfrak{h}$.

Instead of considering the points of $\mathcal{M}/\mathfrak{h}$ as equivalence classes under the action of $\mathfrak{h}$, it is often convenient to choose a fundamental domain $\mathcal{F}$ in $\mathcal{M}$ such that $\{h\mathcal{F}, h \in \mathfrak{h}\}$ is a partition of $\mathcal{M}$. Then, $\mathcal{M}/\mathfrak{h}$ can be identified as $\mathcal{F}$ with additional boundary conditions. A genus 1 Riemann surface, which has the topology of a torus, can for instance be defined as the quotient of $\mathbb{C}$ by a group of translations in two directions, and the fundamental domain may always be chosen as a parallelogram whose opposite sides are glued together, see figure 10.

Given two Riemann surfaces $\mathcal{M}$, $\mathcal{N}$ and a covering map $\rho$ from $\mathcal{M}$ to $\mathcal{N}$, a holomorphic automorphism $h$ of $\mathcal{M}$ is called a deck transformation for $\rho$ if $h$ is compatible with $\rho$, i.e. $\rho \circ h = \rho$. A deck transformation is fully determined by the permutation it induces on $\rho^{-1}(q)$ with $q \in \mathcal{N}$ not a branch point of $\rho$.

## 3.6 Homotopy and homology

Let $\mathcal{M}$ be a Riemann surface and $p \in \mathcal{M}$. Closed loops on $\mathcal{M}$ with base point $p$ are continuous paths on $\mathcal{M}$ starting and ending at $p$. The set of equivalence classes of such loops under homotopy (i.e. continuous deformations) forms a group $\pi_1(\mathcal{M})$ for the concatenation of the paths, called the first homotopy group (or fundamental group), which is independent from the base point up to group isomorphism.

Let $\mathcal{M}$ and $\mathcal{N}$ be two Riemann surfaces, $\rho : \mathcal{M} \to \mathcal{N}$ a covering map, $p \in \mathcal{M}$ not a ramification point of $\rho$ and $\gamma$ a continuous path in $\mathcal{N}$ starting at $\rho(p)$ and avoiding the branch points of $\rho$. The lift $\gamma \cdot p$ of $\gamma$ to the point $p$ is the unique path in $\mathcal{M}$ starting at $p$ whose image by $\rho$ is $\gamma$. Considering a partition of $\mathcal{M}$ into sheets $\mathbb{C}_i$ such that any point $q \in \mathcal{N}$ has a single preimage under $\rho$ in each sheet $\mathbb{C}_i$, we also write $\gamma \cdot \mathbb{C}_i$ for the lift of the path $\gamma \subset \mathcal{N}$ to the preimage $p \in \mathbb{C}_i$ of the starting point of $\gamma$. Even if $\gamma$ is a closed path on $\mathcal{N}$, the lift $\gamma \cdot \mathbb{C}_i$ is not necessarily a loop on $\mathcal{M}$, since its endpoint may be in any sheet $\mathbb{C}_j$, but loops from any equivalence class in $\pi_1(\mathcal{M})$ may be obtained by the lifting procedure.

Considering the example of the covering map $\rho : [z, \mathcal{G}_P] \mapsto z$ from $\mathcal{R}_N$ to $\widehat{\mathbb{C}}$, we call $\theta_j$ the loop in $\widehat{\mathbb{C}}$ encircling only the branch point $z_j$, once in the counter-clockwise direction. Then, the lift $\theta_j^2 \cdot \mathcal{G}_P$ is homotopic to an empty loop on $\mathcal{R}_N$, see dashed path in figure 3. Any loop on $\mathcal{R}_N$ may be written up to homotopy as $\theta_{j_k} \dots \theta_{j_1} \cdot \mathcal{G}_P$, where each $\theta_j$ appears an even number

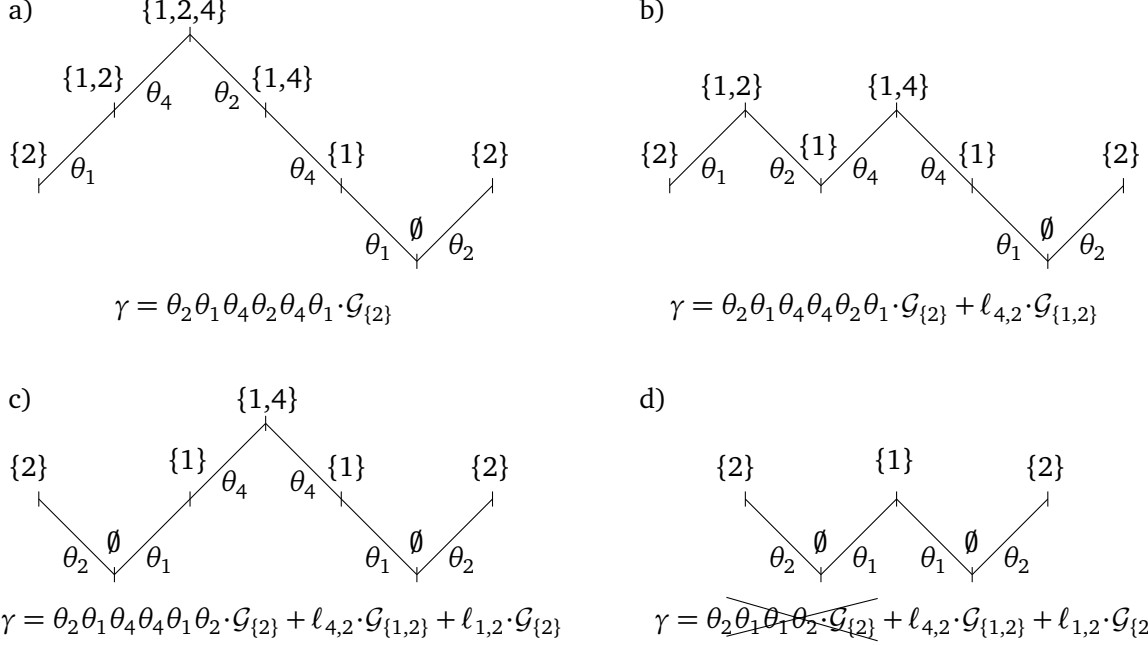

Figure 11: Homology class of a loop $\gamma = \theta_{j_k} \dots \theta_{j_1} \cdot \mathcal{G}_P$ on the Riemann surface $\mathcal{R}_N$ rewritten as a combination of loops $\ell_{a,b} \cdot \mathcal{G}_Q$.

of times in the product so that the final sheet $\mathcal{G}_Q$, $Q = P \ominus \{j_1\} \ominus \dots \ominus \{j_k\}$ is the same as the initial sheet $\mathcal{G}_P$.

Considering several loops based at a point $p$ of a Riemann surface $\mathcal{M}$, the homotopy class of their product $\gamma$ may depend on the order of the loops in $\gamma$. On the other hand, the integral of a holomorphic differential (see next section) over $\gamma$ is independent of the order of the loops in $\gamma$. This motivates the definition of the first homology group $H_1(\mathcal{M}, \mathbb{Z})$, a commutative version of the fundamental group $\pi_1(\mathcal{M})$, for which an additive notation is used for the concatenation of loops. For a compact Riemann surface of genus $g$, it is known that a minimal set of generators of $H_1(\mathcal{M}, \mathbb{Z})$ must have $2g$ elements.

An overcomplete set of generators for $H_1(\mathcal{R}_N, \mathbb{Z})$ is given by the loops $\ell_{a,b} \cdot \mathcal{G}_P$, $1 \leq a < b \leq N$, $P \subset [\![1, N]\!]$, with $\ell_{a,b} = \theta_b \theta_a \theta_b \theta_a$. Indeed, considering a general loop $\gamma = \theta_{j_k} \dots \theta_{j_1} \cdot \mathcal{G}_P$ of $\pi_1(\mathcal{R}_N)$, the sets $P = Q_0, Q_1, \dots, Q_{k-1}, Q_k = P$ indexing the sheets $\mathcal{G}_Q$ crossed by $\gamma$ are such that two consecutive sets $Q_i, Q_{i+1}$ may only differ by a single element, and their cardinals $|Q_i|, |Q_{i+1}|$ differ by $\pm 1$. Choosing an index $i$ such that $|Q_i|$ is a local maximum in the sequence, two situations can occur: if $Q_{i-1} = Q_{i+1}$, then $j_{i-1} = j_i$ and $\theta_{j_i} \theta_{j_{i-1}}$ is homotopic to the identity and can be erased. Otherwise $Q_{i-1} \neq Q_{i+1}$, and there exists $a, b \in [\![1, N]\!]$, $a \neq b$ such that $Q_i$ contains both $a$ and $b$, $Q_{i-1} = Q_i \setminus \{b\}$ and $Q_{i+1} = Q_i \setminus \{a\}$. The loop then contains the factor $\theta_b \theta_a$, which can be replaced by $\theta_a \theta_b$, reducing the value of $|Q_i|$ by 2 at the price of adding $\ell_{a,b} \cdot Q_{i-1}$ to the loop, which has the form desired, see figure 11 for an example.

## 3.7 Differential 1-forms

Let $\mathcal{M}$ be a concrete Riemann surface equipped with a covering map $\rho : [z, i] \mapsto z$ from $\mathcal{M}$ to $\widehat{\mathbb{C}}$. A meromorphic differential $\omega$ on $\mathcal{M}$, also called an Abelian differential, is a differential 1-form such that at any $p = [z, i] \in \mathcal{M}$ away from branch points of $\rho$ one can write $\omega(p) = h_i(z) dz$ with $h_i$ the branch of a meromorphic function $h$ on the sheet $\mathbb{C}_i$ of $\mathcal{M}$. At a ramification point $[z_*, i]$ of $\rho$ with ramification index $e \geq 2$, one has instead

$\omega([z_*, i]) = e\, y^{e-1}\, h_i(z_* + y^e)\mathrm{d}y$ in terms of the local coordinate $y$, $y^e = z - z_*$.

A meromorphic differential has a pole (respectively a zero) of order $n$ at $p = [z, i]$ away from ramification points if the function $h$ as above has a pole (resp. a zero) of order $n$ at $p$, and the residue of the pole is equal to the corresponding residue of $h_i$. For ramification points with local coordinate chosen as above, poles and zeroes of $\omega$ correspond to poles and zeroes of $e\, y^{e-1}\, h_i(z_* + y^e)$ at $y = 0$, and the residue of a pole, equal to the corresponding residue at $y = 0$, is independent from the choice of local coordinate. We observe that poles of $h$ at branch points may be cancelled in $\omega$ by the factor $y^{e-1}$ if the order of the pole is strictly lower than the ramification index. For a compact Riemann surface of genus $g$, the degree of a meromorphic differential, i.e. the total number of zeroes minus the total number of poles counted with multiplicity, is equal to $2g - 2$.

It is convenient to classify meromorphic differentials into three kinds depending on their poles. Meromorphic differentials of the first kind, also called holomorphic differentials, correspond to the special case where the differential has no poles. Holomorphic differentials are closed, i.e. the integral of a holomorphic differential on a path on $\mathcal{M}$ does not change if the path is deformed continuously while its endpoints are kept fixed. Equivalently, the integral of a holomorphic differential over a loop depends only on the equivalence class of the loop under homology. In particular, if the loop is homologous to $0 \in H_1(\mathcal{M}, \mathbb{Z})$, the integral is equal to $0 \in \mathbb{C}$ even though the loop may not be homotopic to an empty loop. Holomorphic differentials form a vector space $H^1(\mathcal{M}, \mathbb{C})$ of dimension $2g$, dual to the first homology group $H_1(\mathcal{M}, \mathbb{Z})$, and are the basic ingredients to build theta functions, a fundamental object in the theory of compact Riemann surfaces in terms of which $\tau$ functions for the KdV equation can for instance be built. For the Riemann surface $\mathcal{R}_N$, a basis of holomorphic differentials is given by the $\omega_{Q,k}$, $Q \subset [\![1, N]\!]$, $k$ integer with $0 \leq k \leq (|Q| - 3)/2$, equal at $[z, P]$ not a ramification point to $\omega_{Q,k}([z, P]) = z^k \mathrm{d}z / \prod_{\ell \in Q} \sqrt{z - z_\ell}$. One can check that there are indeed $2g_N$ holomorphic differentials $\omega_{Q,k}$, each one having degree $2g_N - 2$ with $g_N$ given by (31).

Meromorphic differentials of the second and third kind have poles. Meromorphic differentials of the second kind only have multiple poles with no residues, and are thus closed like holomorphic differentials. Meromorphic differentials of the third kind, on the other hand, also have poles with non-zero residue, and the integral over a small loop with winding number 1 around a pole is equal to $2i\pi$ times the residue of the pole, like for meromorphic functions on $\mathbb{C}$.

As with meromorphic functions, the trace of a meromorphic differential $\omega$ on $\mathcal{M}$ with respect to a covering map $\rho$ from $\mathcal{M}$ to $\mathcal{N}$ is defined as a sum over all preimages of $\rho$,

$$(\mathrm{tr}_\rho\, \omega)(q) = \sum_{\substack{p \in \mathcal{M} \\ \rho(p) = q}} \omega(p)\,. \tag{34}$$

If $\omega$ is holomorphic on $\mathcal{M}$, $\mathrm{tr}_\rho\, \omega$ is then holomorphic on $\mathcal{N}$ [85]. This observation is crucial for the application to KPZ in section 2 in order to move freely contours of integration between the left side and the right side of the cylinder $\mathcal{C}$.

## 3.8 Infinite genus limit

We consider in this section an infinite genus version $\mathcal{R}$ of the Riemann surface $\mathcal{R}_N$. Riemann surfaces $\check{\mathcal{R}}$ and $\mathcal{R}^\Delta$ in terms of which KPZ fluctuations are described in section 2 are constructed as quotients of $\mathcal{R}$ under the action of groups of holomorphic automorphisms.

### 3.8.1 Riemann surface $\mathcal{R}$

The Riemann surface $\mathcal{R}$ can be understood informally as a limit $N \to \infty$ of $\mathcal{R}_N$ with the choice

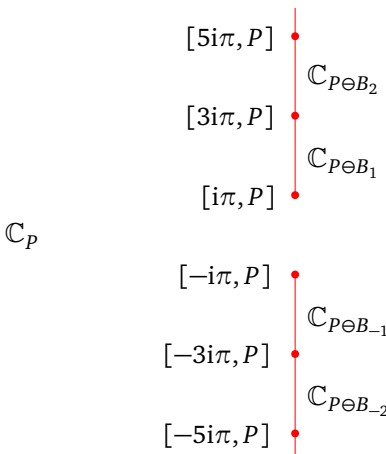

Figure 12: Choice of branch cuts (red, vertical lines) partitioning the Riemann surface $\mathcal{R}$ into the sheets $\mathbb{C}_P$, $P \sqsubset \mathbb{Z} + 1/2$. The ramification points of the covering map $\Pi$ from $\mathcal{R}$ to the infinite cylinder $\mathcal{C}$ are represented with red dots. The connectivity of the sheets is indicated near the cuts, with $B_n$ defined in (35) and $\ominus$ the symmetric difference operator from (30).

of branch points $2i\pi(\mathbb{Z} + 1/2)$ for the covering map $[v, P] \mapsto v$. Topologically, $\mathcal{R}$ is thus an infinite dimensional hypercube made of spheres connected by cylinders, see figure 8 for finite genus analogues $\mathcal{R}_N$. The Riemann surface $\mathcal{R}$ is understood more concretely in section 4.4 as a natural domain of definition for polylogarithms with half-integer index, which generalize the finite sum of square roots (28) on $\mathcal{R}_N$.

Branch cuts are chosen such that $\mathcal{R}$ is partitioned into infinitely many sheets $\mathbb{C}_P$ indexed by subsets $P$ of $\mathbb{Z} + 1/2$ [9], copies of the complex plane slit along the cut $(-i\infty, -i\pi] \cup [i\pi, i\infty)$ with the points on the cut belonging by convention to e.g. the left part of the cut. Introducing for $n \in \mathbb{Z}$ the sets

$$
\begin{array}{lll}
B_0 = \emptyset & n = 0 \\
B_n = \{1/2, 3/2, \ldots, n - 1/2\} & n > 0 & , \\
B_n = \{n + 1/2, \ldots, -3/2, -1/2\} & n < 0
\end{array}
\tag{35}
$$

the sheets $\mathbb{C}_P$ and $\mathbb{C}_{P \ominus B_n}$ are glued together along both sides of the cut $2i\pi(n - 1/2, n + 1/2)$ [10], with $\ominus$ the symmetric difference operator defined in (30), see figure 12.

Since the branch points $2i\pi a$, $a \in \mathbb{Z} + 1/2$ are equally spaced, there exists a covering map $\Pi : [v, P] \mapsto v - 2i\pi[\frac{\operatorname{Im} v}{2\pi}]$, with $[\frac{\operatorname{Im} v}{2\pi}]$ the integer closest to $\frac{\operatorname{Im} v}{2\pi}$, from $\mathcal{R}$ to the infinite cylinder $\mathcal{C} = \{v \in \mathbb{C}, v \equiv v + 2i\pi\}$, see figure 1. Using the covering map $\lambda : v \mapsto e^v$, the infinite cylinder is completely equivalent to $\mathbb{C}^* = \mathbb{C} \setminus \{0\}$, i.e. the Riemann sphere punctured twice $\mathbb{C}^* = \widehat{\mathbb{C}} \setminus \{0, \infty\}$.

A subtle issue concerns whether sheets indexed by infinite sets $P$ should be considered when building $\mathcal{R}$, as such sheets can not be reached from sheets indexed by finite sets without crossing infinitely many branch cuts. Since only sheets indexed by finite sets appear in our formulas for KPZ fluctuations in section 2, we avoid this issue completely and define $\mathcal{R}$ by gluing together only the sheets indexed by finite sets $P \sqsubset \mathbb{Z} + 1/2$ with cardinal $|P|$. A related issue concerns the status of the points at infinity in $\mathcal{R}$. Unlike in $\mathcal{R}_N$, we may not add these points to $\mathcal{R}$ since they correspond to an accumulation of ramification points for the covering

---

[9]The choice of sets of half-integers instead of integers to label the sheets is for symmetry between analytic continuations above and under the real axis for the functions of section 4.

[10]The choice of "vertical" branch cuts like for the sheets $\mathcal{F}_P$ of $\mathcal{R}_N$ instead of "horizontal" branch cuts like for the sheets $\mathcal{G}_P$ is for better compatibility with translations by integer multiples of $2i\pi$ later on.

map $[z, P] \mapsto z$. The two points at infinity on each sheet (on the left side and on the right side of the cut) are thus considered as punctures of $\mathcal{R}$, i.e. infinitesimal boundaries.

Homology classes of loops on $\mathcal{R}$ avoiding the punctures are generated by the same loops $\ell_{a,b} \cdot P$, $a < b \in \mathbb{Z} + 1/2$ as the ones defined for the finite genus analogue $\mathcal{R}_N$ in section 3.6, with $\theta_a$ now encircling $2i\pi a$. Additionally, paths between punctures play an important role for explicit computation of analytic continuations between the various sheets $\mathbb{C}_P$ for the functions needed to express KPZ fluctuations.

### 3.8.2 Riemann surface $\check{\mathcal{R}}$

Anticipating the fact that some functions on $\mathcal{R}$ defined in section 4 have special symmetries when their variable $[v, P] \in \mathcal{R}$ is replaced by $[v + 2i\pi, P]$, we are lead to define also the quotient $\check{\mathcal{R}}$ of $\mathcal{R}$ under the action of a group of translations.

Let us consider the bijective operators $T_{l|r}$ [11] acting on finite sets $P \sqsubset \mathbb{Z} + 1/2$ by $T_l P = P + 1$ and $T_r P = (P + 1) \ominus \{1/2\}$. For any $m \in \mathbb{Z}$, the iterated composition of these operators is given by

$$
\begin{aligned}
T_l^m P &= P + m, \\
T_r^m P &= (P + m) \ominus B_m \,.
\end{aligned}
\tag{36}
$$

The operators $T_{l|r}$ generate two groups $G_{l|r} = \{T_{l|r}^m, m \in \mathbb{Z}\}$ acting on $\mathbb{Z} + 1/2$. The empty set is invariant under $T_l$, and $\{\emptyset\}$ thus constitutes an orbit under the action of $G_l$. The other orbits under the action of $G_l$ are the infinite collections of sets $P$ obtained from one another by shifting all the elements by an integer $m$, and equivalence classes of sets in the same orbit may be labelled by e.g. sets $P$ whose smallest element is equal to $1/2$. Each orbit under the action of $G_r$, on the other hand, contains infinitely many elements. The identity

$$
|(P + m) \ominus B_m|_+ - |(P + m) \ominus B_m|_- = |P|_+ - |P|_- + m \,,
\tag{37}
$$

with $|P|_+$ (respectively $|P|_-$) denoting the number of positive (resp. negative) elements of the set $P$, indicates that each orbit under the action of $G_r$ contains a single element $P$ with $|P|_+ = |P|_-$, which may be used to label the equivalence class.

Let us now consider the map $\mathcal{T}$ defined on the left side of the sheet $\mathbb{C}_\emptyset$ of $\mathcal{R}$ by $\mathcal{T}[v, \emptyset] = [v + 2i\pi, \emptyset]$, $\mathrm{Re}\, v < 0$. The map $\mathcal{T}$ can be extended from the left side of $\mathbb{C}_\emptyset$ to $\mathcal{R}$ by lifting as follows: let $v_0 \in \mathbb{C}$ with $\mathrm{Re}\, v_0 < 0$ and $\gamma = \{\gamma(t), 0 \le t \le 1\}$, a path contained in $\mathbb{C} \setminus 2i\pi(\mathbb{Z} + 1/2)$ starting at $\gamma(0) = v_0$. The lifts $\gamma \cdot \emptyset$ and $(\gamma + 2i\pi) \cdot \emptyset$ are paths on $\mathcal{R}$ starting respectively at $[v_0, \emptyset]$ and $[v_0 + 2i\pi, \emptyset]$. Calling $[v, P]$ the endpoint of $\gamma \cdot \emptyset$, induction on the number of times $\gamma$ crosses the imaginary axis implies that the endpoint of $(\gamma + 2i\pi) \cdot \emptyset$ is $[v + 2i\pi, P + 1]$ if $\mathrm{Re}\, v < 0$ and $[v + 2i\pi, (P + 1) \ominus \{1/2\}]$ if $\mathrm{Re}\, v > 0$, independently of $v_0$ and $\gamma$. Checking carefully what happens when $v$ is on the imaginary axis, especially in a neighbourhood of points of the form $[(2i\pi a)_{l|r}, P]$, we observe that the map $\mathcal{T}$ defined by $\mathcal{T}([v, P]) = [v + 2i\pi, P + 1]$ when $\mathrm{Re}\, v < 0$, $\mathcal{T}([v, P]) = [v + 2i\pi, (P + 1) \ominus \{1/2\}]$ when $\mathrm{Re}\, v > 0$ and extended by continuity to $\mathrm{Re}\, v = 0$ is a homeomorphism of $\mathcal{R}$, and hence an automorphism since it is locally holomorphic. The map $\mathcal{T}$ is additionally a deck transformation for the covering map $\Pi$ from $\mathcal{R}$ to the infinite cylinder $\mathcal{C}$. The iterated composition $\mathcal{T}^m$, $m \in \mathbb{Z}$ is given by

$$
\mathcal{T}^m([v, P]) = \begin{cases} [v + 2i\pi m, T_l^m P] & \mathrm{Re}\, v < 0 \\ [v + 2i\pi m, T_r^m P] & \mathrm{Re}\, v > 0 \end{cases},
\tag{38}
$$

with $T_{l|r}^m$ defined in (36). The group $\check{\mathfrak{g}} = \{\mathcal{T}^m, m \in \mathbb{Z}\}$ acts properly discontinuously on $\mathcal{R}$, and we call $\check{\mathcal{R}} = \mathcal{R}/\check{\mathfrak{g}}$ the Riemann surface quotient of $\mathcal{R}$ under the action of $\check{\mathfrak{g}}$.

---

[11] We write l|r in the following as a shorthand for either the left side l or the right side r of a cut.

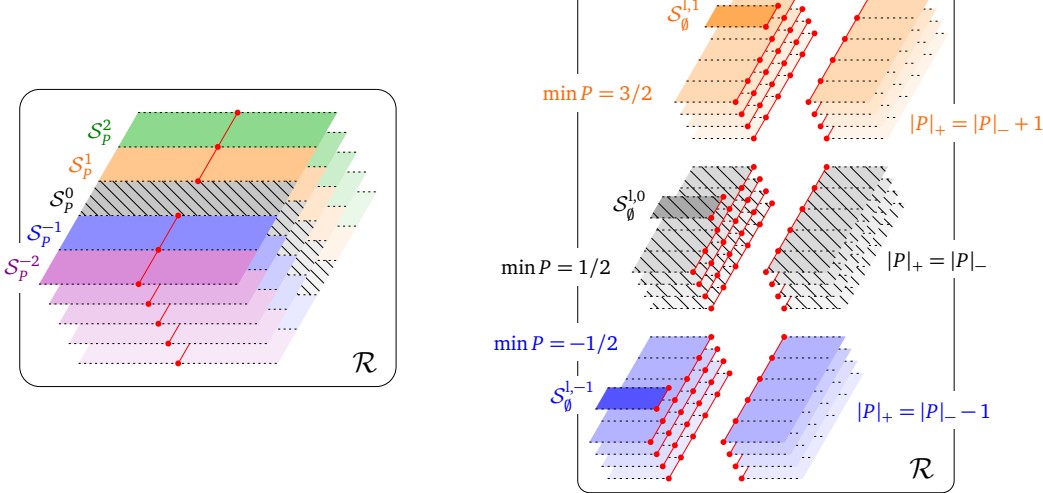

Figure 13: Two choices for a fundamental domain of $\mathcal{R}$ under the action of $\breve{\mathfrak{g}}$, from which $\check{\mathcal{R}} = \mathcal{R}/\breve{\mathfrak{g}}$ is built. The fundamental domain is the hatched portion. How sheets are glued together along branch cuts (red lines, with dots for the branch points) is not represented for clarity. The picture on the left represents all sheets $\mathbb{C}_P$, $P \sqsubset \mathbb{Z}+1/2$ above one another, with the fundamental domain made of the infinite strips $\mathcal{S}_P^0$ corresponding to points $[\nu, P]$ with $-\pi < \operatorname{Im} \nu \le \pi$. The picture on the right represents half-sheets $\mathbb{C}_P^l$ grouped according to the value of $\min P$ (if $P \ne \emptyset$; the half-sheet $\mathbb{C}_\emptyset^l$ is cut into half-infinite strips $\mathcal{S}_\emptyset^{l,m}$) and half-sheets $\mathbb{C}_P^r$ grouped by the value of $|P|_+ - |P|_-$.

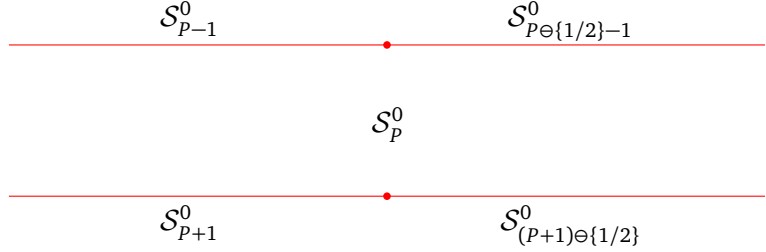

Figure 14: Connectivity of the infinite strips $\mathcal{S}_P^0$, $P \sqsubset \mathbb{Z}+1/2$ partitioning the Riemann surface $\check{\mathcal{R}}$. The red dots represent ramification points for the covering map $\check{\rho}$ sending all the strips to the infinite cylinder $\mathcal{C}$.

The Riemann surface $\check{\mathcal{R}}$ may be partitioned into half-sheets $\mathbb{C}_P^{l|r}$ by taking as a fundamental domain for $\breve{\mathfrak{g}}$ the left side of the sheets $\mathbb{C}_P$ of $\mathcal{R}$ with some choice of representatives $P$ for the orbits under the action of $G_l$ plus the right side of the sheets $\mathbb{C}_P$ with some choice $P$ of representatives for the orbits under the action of $G_r$, with the additional identification $\nu = \nu + 2\mathrm{i}\pi$ for the sheet $\mathbb{C}_\emptyset^l$ since $\{\emptyset\}$ is an orbit under the action of $G_l$, see figure 13 right. How sheets are glued together along the cuts depends on which representatives are chosen for the orbits.

Alternatively, we consider a partition of $\mathcal{R}$ into half-infinite strips

$$\mathcal{S}_P^{l,m} = \{[\nu, P], \operatorname{Re}\nu \le 0, 2\pi(m-1/2) < \operatorname{Im}\nu \le 2\pi(m+1/2)\}, \tag{39}$$
$$\mathcal{S}_P^{r,m} = \{[\nu, P], \operatorname{Re}\nu > 0, 2\pi(m-1/2) < \operatorname{Im}\nu \le 2\pi(m+1/2)\}.$$

The domain $\mathcal{S}_P^m = \mathcal{S}_P^{l,m} \cup \mathcal{S}_P^{r,m}$, contained in the sheet $\mathbb{C}_P$, is connected only when $m = 0$, see figure 13 left. A fundamental domain for the action of $\breve{\mathfrak{g}}$ in $\mathcal{R}$ may be chosen as the $\mathcal{S}_P^0$

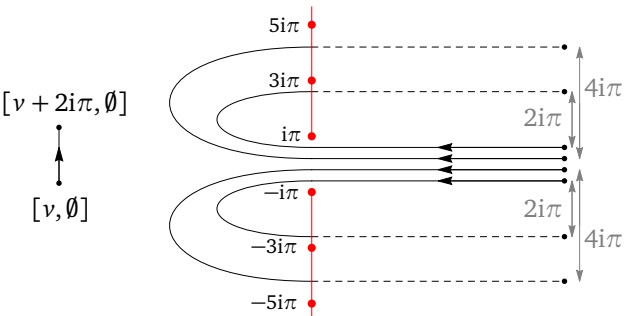

Figure 15: Examples of paths on $\mathcal{R}$ which are also closed loops on $\check{\mathcal{R}}$. The solid curves belong to $\mathbb{C}_\emptyset$ and the dashed lines to $\mathbb{C}_{B_m}$, $m = 2, 1, -1, -2$ from top to bottom.

indexed by all $P \sqsubset \mathbb{Z} + 1/2$. In the Riemann surface $\check{\mathcal{R}}$, the upper part $\text{Im } v = \pi$ of $\mathcal{S}_P^0$ is glued to the lower part $\text{Im } v \to -\pi$ of $\mathcal{S}_{P-1}^0$ on the left side, while the upper part of $\mathcal{S}_P^0$ is glued to the lower part of $\mathcal{S}_{P \ominus \{1/2\}-1}^0$ on the right side, see figure 14. This choice of a fundamental domain naturally defines the covering maps $\check{\Pi} : [v, P] \mapsto [v - 2i\pi[\frac{\text{Im } v}{2\pi}], P]$ from $\mathcal{R}$ to $\check{\mathcal{R}}$ and $\check{\rho} : [v, P] \mapsto v$ from $\check{\mathcal{R}}$ to the infinite cylinder $\mathcal{C}$. KPZ fluctuations with flat initial condition are expressed in section 2.2 in terms of the trace over $\check{\rho}$ of a holomorphic differential on $\check{\mathcal{R}}$.

The existence of the covering map $\check{\Pi}$ from $\mathcal{R}$ to $\check{\mathcal{R}}$ implies that any loop $\ell_{a,b} \cdot P$ on $\mathcal{R}$ project to a loop on $\check{\mathcal{R}}$. There exists however loops on $\check{\mathcal{R}}$ that may not be obtained in such a way, for instance starting from $\mathbb{C}_\emptyset$, the paths $[v - i\pi, \emptyset] \to [v + i\pi, \emptyset]$, $\text{Re } v < 0$ and $[v, \emptyset] \to [(2i\pi m)_l, \emptyset] = [(2i\pi m)_r, B_m] \to [v + 2i\pi m, B_m]$, $\text{Re } v > 0$, $m \in \mathbb{Z}$ represented in figure 15 are closed on $\check{\mathcal{R}}$ but not on $\mathcal{R}$.

### 3.8.3 Riemann surface $\mathcal{R}^\Delta$

We define in this section Riemann surfaces $\mathcal{R}^\Delta$, $\Delta \sqsubset \mathbb{Z} + 1/2$ such that the elements $a \in \Delta$ correspond to branch points $2i\pi a$ that have been "removed" compared to $\mathcal{R}$.

Let us consider the involutions $D_a$, $a \in \mathbb{Z} + 1/2$ acting on finite sets $P \sqsubset \mathbb{Z} + 1/2$ by

$$D_a P = P \ominus \{a\} . \tag{40}$$

For $\Delta \sqsubset \mathbb{Z} + 1/2$, we call $G^\Delta$ the commutative group generated by the $D_a$, $a \in \Delta$. The orbit of any $P \sqsubset \mathbb{Z} + 1/2$ under the action of $G^\Delta$ is the collection of all sets $Q \sqsubset \mathbb{Z} + 1/2$ such that $Q \setminus \Delta = P \setminus \Delta$, and orbits can thus be labelled by the sets $P$ such that $P \cap \Delta = \emptyset$.

The group $G^\Delta$ acting on sets can be upgraded to a group acting on the Riemann surface $\mathcal{R}$. By the commutativity of symmetric difference, the maps $\mathcal{D}_a$, $a \in \mathbb{Z} + 1/2$ defined by

$$\mathcal{D}_a[v, P] = [v, D_a P] \tag{41}$$

are holomorphic automorphisms of $\mathcal{R}$, and deck transformations for the covering map $\Pi$ from $\mathcal{R}$ to the infinite cylinder $\mathcal{C}$. The group $\mathfrak{g}^\Delta$ generated by the $\mathcal{D}_a$, $a \in \Delta$ acts properly discontinuously on $\mathcal{R}$, and the quotient $\mathcal{R}^\Delta = \mathcal{R}/\mathfrak{g}^\Delta$ is a Riemann surface. One has in particular $\mathcal{R}^\emptyset = \mathcal{R}$.

Choosing for fundamental domain the collection of sheets $\mathbb{C}_P$ with $P \cap \Delta = \emptyset$, the construction above defines the covering map $\Pi^\Delta : [v, P] \mapsto [v, P \setminus \Delta]$ from $\mathcal{R}$ to $\mathcal{R}^\Delta$, and $\mathcal{R}^\Delta$ may be partitioned into sheets $\mathbb{C}_P$, $P \cap \Delta = \emptyset$, images of the sheets of $\mathcal{R}$ by $\Pi^\Delta$, see figure 16. The sheet $\mathbb{C}_P$ of $\mathcal{R}^\Delta$ is glued to the sheet $\mathbb{C}_{P \ominus (B_n \setminus \Delta)}$ along the cut $v \in (2i\pi(n-1/2), 2i\pi(n+1/2))$.

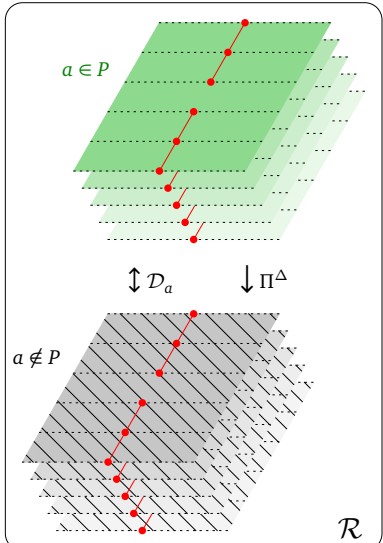
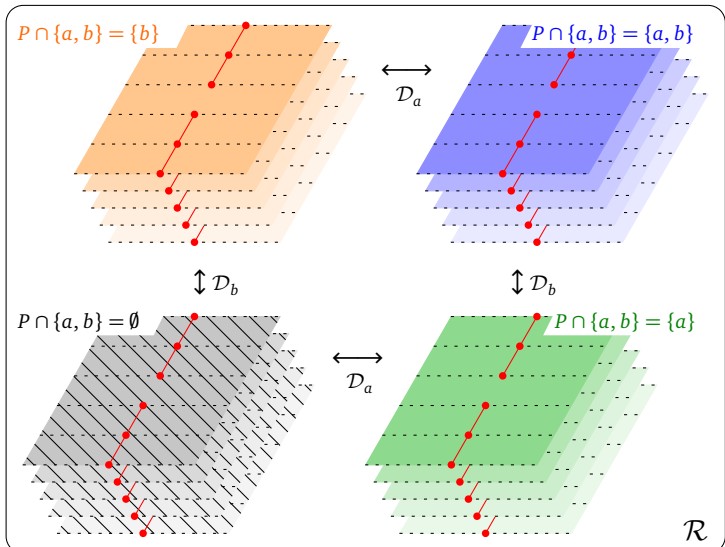

Figure 16: Choice of a fundamental domain of $\mathcal{R}$ under the action of $\mathfrak{g}^{\Delta}$ with $\Delta = \{a\}$ (left) and $\Delta = \{a, b\}$ (right). The sheets $\mathbb{C}_P$, partitioned along dashed lines into pairs of half-infinite strips $\mathcal{S}_P^m = \mathcal{S}_P^{l,m} \cup \mathcal{S}_P^{r,m}$ from (39), are grouped together according to the value of $P \cap \Delta$. The fundamental domain corresponding to $\mathcal{R}^{\Delta}$ is the hatched portion, made from the sheets $\mathbb{C}_P$ with $P \cap \Delta = \emptyset$. How sheets are glued together along branch cuts (red lines, with dots for the branch points) is not represented for clarity.

The sheets $\mathbb{C}_P$ may be further partitioned into pairs of half-infinite strips $\mathcal{S}_P^m = \mathcal{S}_P^{l,m} \cup \mathcal{S}_P^{r,m}$, $m \in \mathbb{Z}$ with the same notations (39) as in the previous section. Unlike for $\check{\mathcal{R}}$, we have not taken a quotient by translations of $2i\pi$ here, so that all values $m \in \mathbb{Z}$ must be taken in the partition. This defines the covering map $\rho^{\Delta} : [v, P] \mapsto v - 2i\pi[\frac{\text{Im} \, v}{2\pi}]$ from $\mathcal{R}^{\Delta}$ to the infinite cylinder $\mathcal{C}$, see figure 1 for a summary of useful covering maps.

### 3.8.4 Collection $\overline{\mathcal{R}}$ of Riemann surface $\mathcal{R}^{\Delta}$

The Riemann surfaces $\mathcal{R}^{\Delta}$ and $\mathcal{R}^{\Delta+1}$ are isomorphic since changing $\Delta$ to $\Delta + 1$ amounts to relabelling the sheets. For the application to KPZ in section 2, it is sometimes convenient, however, to consider sheets from all $\mathcal{R}^{\Delta}$ without the identification $\Delta \equiv \Delta + 1$. In order to do this, we introduce bijective operators $\overline{T}_{l|r}$ acting on pairs of sets $(P, \Delta)$, $P, \Delta \sqsubset \mathbb{Z} + 1/2$ by $\overline{T}_l(P, \Delta) = (P + 1, \Delta + 1)$ and $\overline{T}_r(P, \Delta) = ((P + 1) \ominus (\{1/2\} \setminus (\Delta + 1)), \Delta + 1)$. For any $m \in \mathbb{Z}$, the iterated composition of these operators is given by

$$\overline{T}_l^m(P, \Delta) = (P + m, \Delta + m), \tag{42}$$
$$\overline{T}_r^m(P, \Delta) = ((P + m) \ominus (B_m \setminus (\Delta + m)), \Delta + m) \,.$$

The operators $\overline{T}_{l|r}$ generate two groups $\overline{G}_{l|r} = \{\overline{T}_{l|r}^m, m \in \mathbb{Z}\}$ acting on subsets of $(\mathbb{Z} + 1/2) \times (\mathbb{Z} + 1/2)$. The sector $\{(P, \emptyset), P \sqsubset \mathbb{Z} + 1/2\}$ is an invariant subset under $\overline{G}_{l|r}$, which essentially reduce to the groups $G_{l|r}$ of the previous section there, and have in particular the same orbits. Outside of that sector, equivalence classes of pairs of sets in the same orbit under $\overline{G}_l$ may be labelled by pairs $(P, \Delta)$ with $\Delta$ arbitrary and $P$ such that its smallest element is equal to e.g. $1/2$.

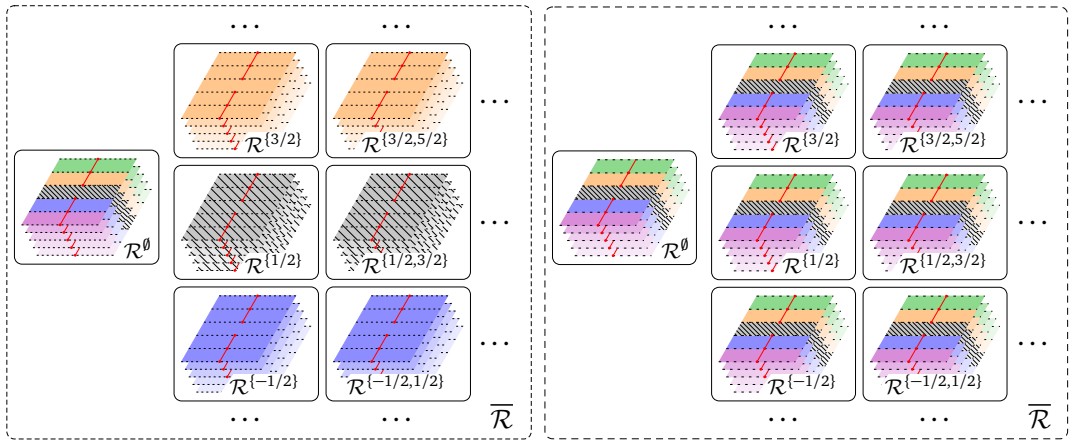

Figure 17: Two choices for the fundamental domain of the collection of Riemann surfaces $\overline{\mathcal{R}}$ under the action of $\overline{\mathfrak{g}}$. The fundamental domain is the hatched portion. How sheets are glued together is not represented for clarity. For the connected component $\mathcal{R}^{\emptyset} = \mathcal{R}$ of $\overline{\mathcal{R}}$, the fundamental domain $\check{\mathcal{R}}$ is chosen as in figure 13 left.

The situation is more complicated for $\overline{G}_{\mathrm{r}}$. Introducing $\lambda_{\pm}(P, \Delta) = |P|_{\pm} - |P \ominus \Delta|_{\mp}$, one has for any $P, \Delta \sqsubset \mathbb{Z} + 1/2$ the identity

$$(\lambda_{\pm} \circ \overline{T}_{\mathrm{r}}^{m})(P, \Delta) = \lambda_{\pm}(P, \Delta) \pm m , \tag{43}$$

which implies in particular that $|P| - |P \ominus \Delta|$ is invariant by $\overline{T}_{\mathrm{r}}^{m}$. The equivalence classes of pairs of sets $(P, \Delta)$ in the same orbit under $\overline{G}_{\mathrm{r}}$ may thus be labelled by pairs $(P, \Delta)$ with $|P|_{+} = |P \ominus \Delta|_{-}$. In the sector $|P| = |P \ominus \Delta|$, such pairs verify both $|P|_{\pm} = |P \ominus \Delta|_{\mp}$.

We consider the collection $\overline{\mathcal{R}}$ of all Riemann surfaces $\mathcal{R}^{\Delta}$, $\Delta \sqsubset \mathbb{Z} + 1/2$ and write $[v, (P, \Delta)] \in \overline{\mathcal{R}}$ for the point $[v, P] \in \mathcal{R}^{\Delta}$. The map $\overline{\mathcal{T}}$ whose iterated composition is given by

$$\overline{\mathcal{T}}^{m}[v, (P, \Delta)] = \begin{cases} [v + 2\mathrm{i}\pi m, \overline{T}_{\mathrm{l}}^{m}(P, \Delta)] & \mathrm{Re}\, v < 0 \\ [v + 2\mathrm{i}\pi m, \overline{T}_{\mathrm{r}}^{m}(P, \Delta)] & \mathrm{Re}\, v > 0 \end{cases} , \tag{44}$$

with $\overline{T}_{\mathrm{l}|\mathrm{r}}^{m}$ defined in (42) is a holomorphic automorphism of $\overline{\mathcal{R}}$ generating a group $\overline{\mathfrak{g}}$. The map $\overline{\mathcal{T}}^{m}$ restricts to an isomorphism between $\mathcal{R}^{\Delta}$ and $\mathcal{R}^{\Delta + m}$ for any $\Delta \sqsubset \mathbb{Z} + 1/2$. In particular, $\overline{\mathcal{T}}$ has the same action on $\mathcal{R}^{\emptyset} = \mathcal{R}$ as $\mathcal{T}$ defined in (38).

The quotient $\overline{\mathcal{R}}/\overline{\mathfrak{g}}$ corresponds to the collection of Riemann surfaces containing $\check{\mathcal{R}}$ and a representative $\mathcal{R}^{\Delta}$ from each equivalence class $\Delta \equiv \Delta + 1$ under isomorphism, see figure 17 left. Another choice of fundamental domain for the action of $\overline{\mathcal{T}}$ consists in taking $\check{\mathcal{R}}$ plus the infinite strips $\mathcal{S}_{P}^{0}$, $P \cap \Delta = \emptyset$ from all $\mathcal{R}^{\Delta}$ without the identification $\Delta \equiv \Delta + 1$, see figure 17 right.

# 4 Functions on the Riemann surfaces $\mathcal{R}, \check{\mathcal{R}}, \mathcal{R}^{\Delta}$

In this section, we study polylogarithms with half-integer index and several functions built from them living on the Riemann surfaces $\mathcal{R}, \check{\mathcal{R}}, \mathcal{R}^{\Delta}$ defined in section 3.8, and used for KPZ fluctuations in section 2. The results presented until section 4.6 are not new and merely serve to introduce notations. The explicit analytic continuations performed from section 4.7 on the specific functions needed for KPZ are presumably new.

## 4.1 $2i\pi(\mathbb{Z}+1/2)$-continuable functions, translation and analytic continuations

Throughout section 4, we consider functions analytic in the domain

$$\mathbb{D} = \mathbb{C} \setminus ((-i\infty, -i\pi] \cup [i\pi, i\infty)), \tag{45}$$

and which may be continued analytically along any path in $\mathbb{C}$ avoiding the points in $2i\pi(\mathbb{Z}+1/2)$, i.e. such paths never encounter branch points, poles or essential singularities. We borrow the terminology $2i\pi(\mathbb{Z}+1/2)$-continuable from the theory of resurgent function for this class of functions. In the presence of branch points, which must necessarily belong to $2i\pi(\mathbb{Z}+1/2)$, new functions analytic in $\mathbb{D}$ are generated by crossing branch cuts.

We introduce the notation $A_n^l f$ (respectively $A_n^r f$) for the function obtained from a $2i\pi(\mathbb{Z}+1/2)$-continuable function $f$ after crossing the (potential) branch cut $(2i\pi(n-1/2), 2i\pi(n+1/2))$, $n \in \mathbb{Z}$ from left to right (resp. from right to left), i.e. when the function $f$ is continued analytically along the path $x + 2i\pi n$ with $x$ increasing from $0^-$ to $0^+$ (resp. decreasing from $0^+$ to $0^-$), see figure 18 right. Both $A_n^{l|r} f$ are understood as analytic functions in $\mathbb{D}$. When the analytic continuation from both sides gives the same result, which happens if the branch points of $f$ are of square root type, we write $A_n$ instead of $A_n^l$ or $A_n^r$. Since $f$ is assumed to be analytic in $\mathbb{D}$, there is no branch cut between $-i\pi$ and $i\pi$, and one has $A_0^{l|r} f = f$. Furthermore, the operators $A_n^l$ and $A_n^r$ are inverse of each other, $A_n^l A_n^r f = A_n^r A_n^l f = f$.

We will be interested in the following in the interplay between analytic continuation and translation by integer multiples of $2i\pi$. Since we are working with functions analytic in $\mathbb{D}$, one has to distinguish the effect of translations on the left and on the right, as two points apart of $2i\pi$ moved from the left side to the right side end up crossing distinct branch cuts. We introduce translation operators $T_{l|r}$ acting on $2i\pi(\mathbb{Z}+1/2)$-continuable functions $f$, such that $T_l f$ and $T_r f$ are analytic in $\mathbb{D}$ and verify respectively $(T_l f)(v) = f(v + 2i\pi)$ when $\text{Re } v < 0$ and $(T_r f)(v) = f(v + 2i\pi)$ when $\text{Re } v > 0$, see figure 18 right.

One has for any $m, n \in \mathbb{Z}$ the identities $A_{m+n}^l = T_r^{-m} A_n^l T_l^m$ and $A_{m+n}^r = T_l^{-m} A_n^r T_r^m$. In particular, since $A_0^{l|r}$ is the identity operator, we observe that the analytic continuation can be deduced from translations on both sides:

$$A_n^l = T_r^{-n} T_l^n, \tag{46}$$
$$A_n^r = T_l^{-n} T_r^n .$$

These identities are used in the following as a convenient way to derive analytic continuations of functions defined as integrals of meromorphic differentials.

## 4.2 Polylogarithms

The polylogarithm of index $s \in \mathbb{C}$ is defined for $|z| < 1$ by the series

$$\text{Li}_s(z) = \sum_{k=1}^{\infty} \frac{z^k}{k^s} . \tag{47}$$

When $s$ is a non-positive integer, $s \in -\mathbb{N}$, $\text{Li}_s(z)$ reduces to a rational function of $z$ with a pole at $z = 1$. Otherwise, analytic continuation beyond the unit disk allows to extend $\text{Li}_s$ to an analytic function in $\mathbb{C} \setminus [1, \infty)$, the principal value of $\text{Li}_s$, with a branch point at $z = 1$ and a branch cut traditionally chosen to be the real numbers larger than 1. For $s = 1$, $\text{Li}_1(z) = -\log(1-z)$, and the branch point is of logarithmic type. The function $\text{Li}_1$ can thus be extended to an analytic function on a Riemann surface built from infinitely many sheets $\mathbb{C}_k$, $k \in \mathbb{Z}$, such that the top part of the cut in $\mathbb{C}_k$ is glued to the bottom part of the cut in $\mathbb{C}_{k+1}$.

Analytic continuations in the variable $z$ of $\mathrm{Li}_s(z)$ when $s \neq 1$ is more involved [86]. Indeed, after analytic continuation from below the cut $[1, \infty)$, the function $\mathrm{Li}_s(z)$ becomes $\mathrm{Li}_s(z) - \frac{2\mathrm{i}\pi(\log z)^{s-1}}{\Gamma(s)}$ with $\Gamma$ the Euler gamma function. The power $s-1$ in the extra term leads to the same branch point $z = 1$ as $\mathrm{Li}_s$, while the logarithm gives an additional branch point at $z = 0$, and makes the structure of the Riemann surface more complicated since further analytic continuation must take into account how $(\log z)^{s-1}$ varies across branch cuts.

Because of the extra logarithm obtained from analytic continuation, it is useful to consider instead [12] the function $\mathrm{Li}_s(-\mathrm{e}^\nu)$. In terms of the variable $\nu$, this function has an alternative expression as the complete Fermi-Dirac integral

$$\mathrm{Li}_s(-\mathrm{e}^\nu) = -\frac{1}{\Gamma(s)} \int_0^\infty \mathrm{d}u \, \frac{u^{s-1}}{\mathrm{e}^{u-\nu} + 1} \tag{48}$$

for $\mathrm{Re}\, s > 0$, and in terms of the Hurwitz zeta function $\zeta(s, u) = \sum_{k=0}^\infty (u + k)^{-s}$ as

$$\mathrm{Li}_s(-\mathrm{e}^\nu) = \frac{\Gamma(1-s)}{(2\pi)^{1-s}} \left( \mathrm{i}^{1-s} \zeta\left(1-s, \frac{1}{2} + \frac{\nu}{2\mathrm{i}\pi}\right) + \mathrm{i}^{s-1} \zeta\left(1-s, \frac{1}{2} - \frac{\nu}{2\mathrm{i}\pi}\right) \right). \tag{49}$$

The function $\mathrm{Li}_s(-\mathrm{e}^\nu)$ has the branch points $2\mathrm{i}\pi a$, $a \in \mathbb{Z} + 1/2$. Crossing the branch cut associated to $2\mathrm{i}\pi a$ in the anti-clockwise direction relative to the branch point transforms $\mathrm{Li}_s(-\mathrm{e}^\nu)$ into $\mathrm{Li}_s(-\mathrm{e}^\nu) - \frac{2\mathrm{i}\pi(\nu - 2\mathrm{i}\pi a)^{s-1}}{\Gamma(s)}$, which has the same branch points as the principal value: the function $\nu \mapsto \mathrm{Li}_s(-\mathrm{e}^\nu)$ thus belongs to the class of $2\mathrm{i}\pi(\mathbb{Z} + 1/2)$-continuable functions defined in the previous section if the branch cut is chosen as $(-\mathrm{i}\infty, -\mathrm{i}\pi] \cup [\mathrm{i}\pi, \mathrm{i}\infty)$.

In the much studied case where $s \geq 2$ is an integer, the extra terms obtained after crossing branch cuts are polynomials in $\nu$, and are thus inert by analytic continuation: crossing a branch cut twice in the same direction simply adds the same extra term once more. All branch points are thus of logarithmic type. When $s$ is not an integer, the situation becomes more complicated since the extra terms are multiplied by a phase after analytic continuation. After setting notations for square roots with specific branch cuts in the next section, we focus in section 4.4 on the case where $s$ is a half-integer, $s \in \mathbb{Z} + 1/2$, which is the case of interest for KPZ.

## 4.3 Square roots $\kappa_a(\nu)$

Before considering polylogarithms with half-integer index, we introduce for convenience a square root function $\kappa_a$, $a \in \mathbb{Z} + 1/2$ with branch point $2\mathrm{i}\pi a$. We define

$$\kappa_a(\nu) = \sqrt{4\mathrm{i}\pi a} \sqrt{1 - \frac{\nu}{2\mathrm{i}\pi a}} = \sqrt{\mathrm{sgn}(a)\mathrm{i}} \sqrt{|4\pi a| + \mathrm{sgn}(a)2\mathrm{i}\nu}, \tag{50}$$

with the usual branch cut $\mathbb{R}^- = (-\infty, 0]$ for the square roots so that the branch cut of $\kappa_a$ is the interval $\mathrm{sgn}(a)2\mathrm{i}\pi(|a|, \infty)$. In particular, $\kappa_a$ is analytic in the domain $\mathbb{D}$ defined in (45). The expression (50) for $\kappa_a$ reduces to $\kappa_a(\nu) = \sqrt{4\mathrm{i}\pi a - 2\nu}$ when $\mathrm{Re}\, \nu < 0$ and $\kappa_a(\nu) = \mathrm{sgn}(a)\mathrm{i}\sqrt{2\nu - 4\mathrm{i}\pi a}$ when $\mathrm{Re}\, \nu > 0$.

We list a few useful properties of the functions $\kappa_a$. The derivative $\kappa_a'$, also analytic in $\mathbb{D}$, is equal to

$$\kappa_a' = -\frac{1}{\kappa_a}. \tag{51}$$

For $\nu \in \mathbb{D}$, the possible locations of $\kappa_a(\nu)$ in the complex plane are such that both $\log \kappa_a$ and $\log(\kappa_a + \kappa_b)$, $a, b \in \mathbb{Z} + 1/2$ are analytic in $\mathbb{D}$ with the branch cut of the logarithm

---

[12]The minus sign in front of $\mathrm{e}^\nu$ is introduced to make formulas symmetric under complex conjugation.

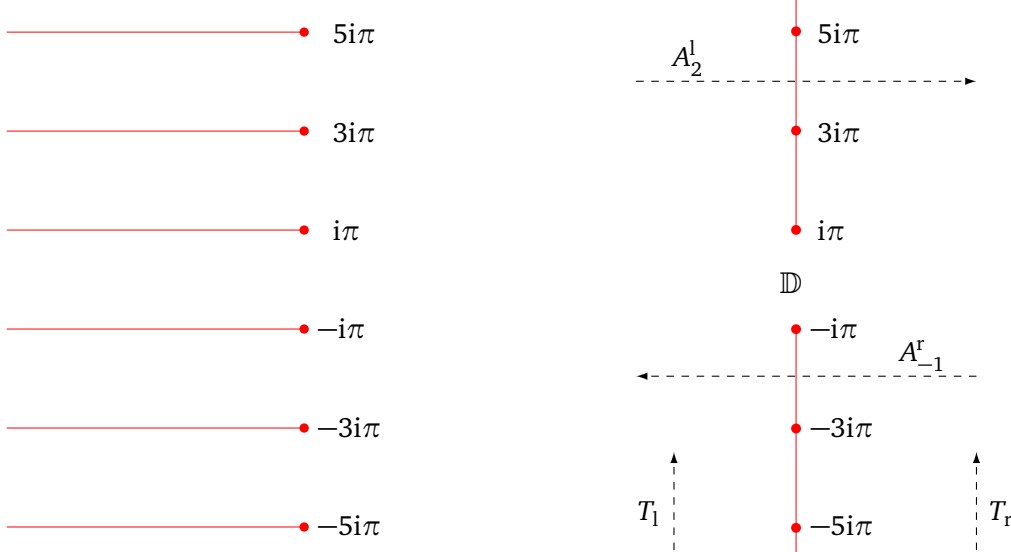

Figure 18: Two possible choices of branch cuts for the function $\chi_\emptyset$. The choice on the left, corresponding to the principal value of the polylogarithm $\text{Li}_{5/2}$ in (56), is not convenient because of the presence of symmetries $v \mapsto v + 2i\pi$. The choice on the right, which is the one actually used in the paper, defines by removing the cuts a space $\mathbb{D} = \mathbb{C} \setminus ((-i\infty, -i\pi] \cup [i\pi, i\infty))$ on which the chosen determination of $\chi_\emptyset$ is analytic. Examples of paths of analytic continuation for the translation operators $T_l$, $T_r$ and the analytic continuation operators $A_n^l$, $A_n^r$, $n \in \mathbb{Z}$ defined in section 4.1 are also indicated on the right with dashed arrows.

taken as $\mathbb{R}^-$, see appendix C.1. Finally, shifting $v$ by an integer multiple of $2i\pi$ is equivalent to shifting $a$, up to a possible minus sign: $\kappa_a(v + 2i\pi n) = \kappa_{a-n}(v)$ when $\text{Re}\, v < 0$ and $\kappa_a(v + 2i\pi n) = \sigma_a(B_n)\kappa_{a-n}(v)$ when $\text{Re}\, v > 0$, with $\sigma_a$ given in (29) and $B_n$ in (35). For the logarithm of $\kappa_a$, using (180), one has instead $\log \kappa_a(v+2i\pi n) = \log \kappa_{a-n}(v)$ when $\text{Re}\, v < 0$ and $\log \kappa_a(v + 2i\pi n) = \log \kappa_{a-n}(v) + 1_{\{a \in B_n\}} i\pi \, \text{sgn}(n)$ when $\text{Re}\, v > 0$. In terms of the translation operators of section 4.1, these identities can be written as

$$T_l^n \kappa_a = \kappa_{a-n}, \tag{52}$$
$$T_r^n \kappa_a = \sigma_a(B_n)\kappa_{a-n} \, .$$

Using (46), this leads for the analytic continuation from either side of the cut to

$$A_n \kappa_a = \sigma_a(B_n)\kappa_a \, , \tag{53}$$

which is already obvious from the definition of $\kappa_a$. The analytic continuation gives the same result from both sides of the cut since the branch point of $\kappa_a$ is of square root type. Similarly, one has for the logarithm of $\kappa_a$

$$T_l^n \log \kappa_a = \log \kappa_{a-n}, \tag{54}$$
$$T_r^n \log \kappa_a = \log \kappa_{a-n} + 1_{\{a \in B_n\}} i\pi \, \text{sgn}(n) \, ,$$

which using (46), leads to distinct analytic continuations from either side,

$$A_n^l \log \kappa_a = \log \kappa_a - 1_{\{a \in B_n\}} i\pi \, \text{sgn}(n), \tag{55}$$
$$A_n^r \log \kappa_a = \log \kappa_a + 1_{\{a \in B_n\}} i\pi \, \text{sgn}(n) \, ,$$

and the branch point of $\log \kappa_a$ is of logarithmic type.

## 4.4 Half-integer polylogarithms and function $\chi$ on $\mathcal{R}$

We introduce the function

$$\chi_{\emptyset}(v) = -\frac{\text{Li}_{5/2}(-e^v)}{\sqrt{2\pi}}. \tag{56}$$

The branch points of $\chi_{\emptyset}$ are the $2i\pi a$, $a \in \mathbb{Z} + 1/2$. If the principal value of $\text{Li}_{5/2}$ is chosen in (56), the branch cut associated to the branch point $a$ is $2i\pi a + (-\infty, 0]$, see figure 18 left. We choose instead the branch cut $(-i\infty, -i\pi] \cup [i\pi, i\infty)$ in the following, see figure 18 right, so that $\chi_{\emptyset}$ is analytic in the domain $\mathbb{D}$ defined in (45). Using (49), this can be done explicitly by writing $\chi_{\emptyset}(v)$ as

$$\chi_{\emptyset}(v) = \frac{8\pi^{3/2}}{3}\left(e^{i\pi/4}\zeta\left(-\frac{3}{2}, \frac{1}{2} + \frac{v}{2i\pi}\right) + e^{-i\pi/4}\zeta\left(-\frac{3}{2}, \frac{1}{2} - \frac{v}{2i\pi}\right)\right), \tag{57}$$

which is indeed analytic for $v \in \mathbb{D}$ if the usual branch cut $\mathbb{R}^-$ is chosen for the Hurwitz $\zeta$ function $\zeta(-3/2, \cdot)$.

From (47), the polylogarithm expression (56) gives for large $|v|$, $\text{Re}\, v < 0$ the convergent expansion

$$\chi_{\emptyset}(v) \simeq -\frac{1}{\sqrt{2\pi}}\sum_{j=1}^{\infty}\frac{(-1)^j e^{jv}}{j^{5/2}} \simeq \frac{e^v}{\sqrt{2\pi}}. \tag{58}$$

Using the asymptotic expansion for the Hurwitz zeta function in terms of Bernoulli numbers $B_r$,

$$\zeta(s, u) \simeq -\frac{1}{1-s}\sum_{r=0}^{\infty}\binom{1-s}{r}\frac{B_r}{u^{r+s-1}}, \tag{59}$$

when $|u| \to \infty$ away from the negative real axis $\mathbb{R}^-$, the expression (57) for $\chi_{\emptyset}$ gives the asymptotic expansion on the other side of the branch cut,

$$\chi_{\emptyset}(v) \simeq \frac{32\pi^{3/2}}{15}\sum_{r=0}^{\infty}\binom{5/2}{2r}\frac{(-1)^r(2^{1-2r}-1)B_{2r}}{(\frac{v}{2\pi})^{2r-5/2}} \simeq \frac{(2v)^{5/2}}{15\pi} + \frac{\pi\sqrt{2v}}{6} - \frac{7\pi^3}{360(2v)^{3/2}}, \tag{60}$$

when $|v| \to \infty$, $\text{Re}\, v > 0$. The function $\chi_{\emptyset}$ has thus an essential singularity at infinity: on the left side of the cut, the convergent expansion (58) for $\chi_{\emptyset}(v)$ is given as a series in $e^v$, while on the right side of the cut, the expansion (60) of $\chi_{\emptyset}(v)$ has a vanishing radius of convergence in the variable $1/v$.

From the Hurwitz zeta representation (57), it is possible to rewrite $\chi_{\emptyset}$ as an infinite sum of powers $3/2$. In terms of the square root functions $\kappa_a$ defined in (50), the identity $\zeta(s, u+1) = \zeta(s, u) - u^{-s}$, valid for any $s \in \mathbb{C} \setminus \{1\}$, even though $\zeta(s, u) = \sum_{k=0}^{\infty}(u+k)^{-s}$ only holds when $\text{Re}\, s > 1$, leads for any non-negative integer $M$ to

$$\chi_{\emptyset}(v) = \frac{8\pi^{3/2}}{3}\left(e^{i\pi/4}\zeta\left(-\frac{3}{2}, M + \frac{1}{2} + \frac{v}{2i\pi}\right) + e^{-i\pi/4}\zeta\left(-\frac{3}{2}, M + \frac{1}{2} - \frac{v}{2i\pi}\right)\right) \tag{61}$$
$$-\sum_{a=-M+1/2}^{M-1/2}\frac{\kappa_a^3(v)}{3},$$

where the sum is over half integers $a \in \mathbb{Z} + 1/2$ between $-M + 1/2$ and $M - 1/2$. In the expression above, the first term containing the $\zeta$ functions is analytic in the strip $-(M + 1/2)\pi < \text{Re}\, v < (M + 1/2)\pi$: the only branch points in the strip are contributed by the sum. Our choice of branch cuts for the functions $\kappa_a$ and $\zeta(-3/2, \cdot)$ then agrees with the requirement that the expression (61) must be analytic in $\mathbb{D}$.

Taking $M \to \infty$ and using the asymptotic expansion (59), we finally obtain $\chi_\emptyset$ as

$$\chi_\emptyset(v) = \lim_{M \to \infty} \left( -\frac{4(2\pi M)^{5/2}}{15\pi} - \frac{2v(2\pi M)^{3/2}}{3\pi} \right. \tag{62}$$
$$\left. + \frac{(\pi^2 + 3v^2)\sqrt{2\pi M}}{6\pi} - \sum_{a=-M+1/2}^{M-1/2} \frac{\kappa_a^3(v)}{3} \right).$$

In this expression, each term of the sum is analytic for $v \in \mathbb{D}$ with our choice of branch cut for the functions $\kappa_a$. The two choices of branch cuts in figure 18 are thus analogous to the ones in figure 5 for the finite sum of $m$ square roots defined in (28).

Analytic continuation of $\chi_\emptyset$ across the branch cut $(2i\pi(n-1/2), 2i\pi(n+1/2))$ changes the signs of a finite number of terms in the infinite sum representation (62). After a finite number of branch cut crossings, the function $\chi_\emptyset$ is replaced by

$$\chi_P(v) = \lim_{M \to \infty} \left( -\frac{4(2\pi M)^{5/2}}{15\pi} - \frac{2v(2\pi M)^{3/2}}{3\pi} + \frac{(\pi^2 + 3v^2)\sqrt{2\pi M}}{6\pi} \right. \tag{63}$$
$$\left. - \sum_{a=-M+1/2}^{M-1/2} \sigma_a(P)\frac{\kappa_a^3(v)}{3} \right),$$

$P \sqsubset \mathbb{Z}+1/2$, where the sign $\sigma_a(P)$ is defined in (29). The set $P$ contains the indices $a \in \mathbb{Z}+1/2$ for which the sign of $\kappa_a^3(v)$ has been flipped an odd number of times after crossing branch cuts, and depends on the path along which the analytic continuation is taken. When $P$ is the empty set $\emptyset$, $\chi_P$ reduces to $\chi_\emptyset$. The difference between $\chi_P$ and $\chi_\emptyset$ is a finite sum,

$$\chi_P(v) = \chi_\emptyset(v) + \sum_{a \in P} \frac{2\kappa_a^3(v)}{3} . \tag{64}$$

The expressions (63) and (64) for $\chi_P$ are manifestly analytic in $\mathbb{D}$ with our choice of branch cuts for $\chi_\emptyset$ and $\kappa_a$.

In terms of the operators $A_n$ defined in section 4.1, analytic continuations across branch cuts of $\chi_P$ are simply given by

$$A_n \chi_P = \chi_{P \ominus B_n} , \tag{65}$$

where the symmetric difference operator $\ominus$ and the set $B_n$ are defined in (30) and (35). Since all the branch cuts of $\chi_P$ are of square root type, the analytic continuations are independent of the side from which the branch cut is crossed.

The functions $\chi_P$ on $\mathbb{D}$ define a function $\chi$ analytic [13] on the Riemann surface $\mathcal{R}$ of section 3.8.1 by

$$\chi([v,P]) = \chi_P(v) . \tag{66}$$

Derivatives of $\chi_\emptyset$ can also be extended to functions on $\mathcal{R}$. The function $\chi'$, defined by $\chi'([v,P]) = \chi_P'(v)$ is still analytic on $\mathcal{R}$. The function $\chi''$, defined by $\chi''([v,P]) = \chi_P''(v)$ is only meromorphic on $\mathcal{R}$, as it has poles at the points $[(2i\pi a)_{l|r}, P]$.

## 4.5 Symmetries of $\chi$

The expansion $\mathrm{Li}_s(z) = \sum_{k=1}^\infty z^k/k^s$, valid for $|z| < 1$, indicates that $\chi_\emptyset(v)$ is periodic with period $2i\pi$ in the sector $\mathrm{Re}\, v < 0$. This is no longer true when $\mathrm{Re}\, v > 0$ since the points $v$ and

---

[13]We recall that the points at infinity, where $\chi_P$ has an essential singularity, are understood as punctures and do not belong to $\mathcal{R}$.

$\nu + 2i\pi$ end up in distinct sheets of $\mathcal{R}$ when moved continuously from $\mathrm{Re}\,\nu < 0$, which leads to more complicated symmetries.

The action of translations on $\chi_P$ can be deduced from its expression (63) as an infinite sum. Recalling the translation operators $T_{l|r}$ from section 4.1 and using $\kappa_a(\nu) \simeq \sqrt{4i\pi a} - \nu/\sqrt{4i\pi a}$ when $|a| \to \infty$, the identities (52) lead to

$$
\begin{aligned}
T_l^{-n}\chi_P &= \chi_{P+n} \\
T_r^{-n}\chi_P &= \chi_{(P+n)\ominus B_n}
\end{aligned}
\tag{67}
$$

for any $n \in \mathbb{Z}$, with $B_n$ defined in (35). More explicitly, $\chi_P(\nu - 2i\pi n) = \chi_{P+n}(\nu)$ when $\mathrm{Re}\,\nu < 0$ and $\chi_P(\nu - 2i\pi n) = \chi_{(P+n)\ominus B_n}(\nu)$ when $\mathrm{Re}\,\nu > 0$. The identities (67) are compatible with analytic continuation (65) through (46) since $(P \ominus B_n) - n = (P - n) \ominus B_{-n}$.

In terms of the map $\mathcal{T}$ defined in (38), the identities (67) correspond to the symmetry $\chi \circ \mathcal{T} = \chi$ for the function $\chi$ on $\mathcal{R}$. Since the Riemann surface $\check{\mathcal{R}}$ is defined as the quotient of $\mathcal{R}$ by the group generated by $\mathcal{T}$, this means that $\chi$ may also be defined as an analytic function on $\check{\mathcal{R}}$. In the following, we use the same notation $\chi$ for both the function defined on $\mathcal{R}$ and on $\check{\mathcal{R}}$, and similarly for the functions $\chi'$ and $\chi''$ built from the derivatives $\chi'_P$ and $\chi''_P$.

## 4.6 Function $I_0$

We consider for $\nu \in \mathbb{D}$ the function

$$
I_0(\nu) = -\frac{1}{4}\int_{-\infty}^{\nu}\frac{d\nu}{1+e^{-\nu}} = -\frac{1}{4}\int_{-\infty}^{\nu}d\nu\left(\frac{1}{2} + \sum_{a\in\mathbb{Z}+1/2}\frac{1}{\nu - 2i\pi a}\right),
\tag{68}
$$

with a path of integration contained in $\mathbb{D}$, see figure 19. Since the integrand is analytic in $\mathbb{D}$, $I_0$ is independent from the path of integration. Because of the poles located at $2i\pi a$, $a \in \mathbb{Z}+1/2$, however, the function $I_0$ is defined with the branch cut $(-i\infty, -i\pi] \cup [i\pi, i\infty)$. The integral can be performed explicitly, and one has the alternative expression

$$
I_0(\nu) = \begin{cases} -\frac{1}{4}\log(1+e^{\nu}) & \mathrm{Re}\,\nu < 0 \\ -\frac{\nu}{4} - \frac{1}{4}\log(1+e^{-\nu}) & \mathrm{Re}\,\nu > 0 \end{cases}.
\tag{69}
$$

An expression manifestly analytic in $\mathbb{D}$ follows from the reflection formula for the Euler $\Gamma$ function $\Gamma(\frac{1}{2} - iz)\Gamma(\frac{1}{2} + iz) = \pi/\cosh(\pi z)$,

$$
I_0(\nu) = -\frac{\log(2\pi)}{4} - \frac{\nu}{8} + \frac{1}{4}\log\Gamma\left(\frac{1}{2} - \frac{\nu}{2i\pi}\right) + \frac{1}{4}\log\Gamma\left(\frac{1}{2} + \frac{\nu}{2i\pi}\right).
\tag{70}
$$

Here, $\log\Gamma(z)$ is the principal value of the $\log\Gamma$ function [14], which is analytic for $z \in \mathbb{C} \setminus \mathbb{R}^-$. Alternatively, the $\log\Gamma$ function may be written as an infinite sum of logarithms, and one has

$$
I_0(\nu) = -\frac{\log 2}{4} - \frac{\nu}{8} - \frac{1}{4}\sum_{a\in\mathbb{Z}+1/2}\log\left(1 - \frac{\nu}{2i\pi a}\right).
\tag{71}
$$

When $\mathrm{Re}\,\nu < 0$, the function $I_0$ verifies from (69) the identity $I_0(\nu + 2i\pi n) = I_0(\nu)$ for $n \in \mathbb{Z}$. When $\mathrm{Re}\,\nu > 0$, one has instead $I_0(\nu + 2i\pi n) = I_0(\nu) - i\pi n/2$. In terms of the translation operators defined in section 4.1, one can write

$$
\begin{aligned}
T_l^n I_0 &= I_0, \\
T_r^n I_0 &= I_0 - i\pi n/2\,.
\end{aligned}
\tag{72}
$$

---

[14]And not merely the logarithm of the $\Gamma$ function for some choice of branch cut for the logarithm.

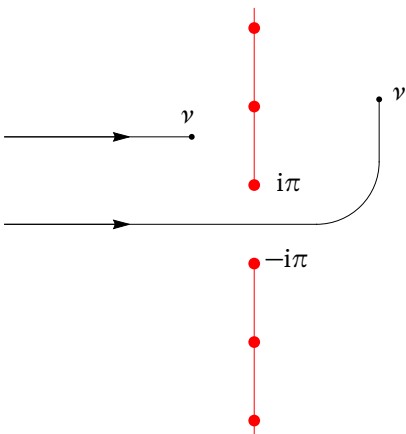

Figure 19: Examples of paths of integration in $\mathbb{D}$ for $I_0(\nu)$ in (68) and for $J_P(\nu)$ in (74), so that the functions are analytic in $\mathbb{D}$. The vertical, red lines represent the branch cuts $\mathbb{C} \setminus \mathbb{D}$. The bigger, red dots are the branch points $2i\pi a$, $a \in \mathbb{Z} + 1/2$.

The function $I_0$ is $2i\pi(\mathbb{Z} + 1/2)$-continuable, as defined in section 4.1. Analytic continuations across the branch cut can then be written solely in terms of the translation operators on both sides as (46), and we obtain

$$A_n^l I_0 = I_0 + i\pi n/2, \tag{73}$$
$$A_n^r I_0 = I_0 - i\pi n/2 \,,$$

which can also be proved more directly from e.g. (69).

We note that (73) implies that the domain of definition of the function $I_0$ may not be extended to the Riemann surfaces $\mathcal{R}$ or $\check{\mathcal{R}}$. This is a consequence of the presence of logarithmic branch points, coming from the integration of poles, instead of the square root branch points required for $\mathcal{R}$. The domain of definition of the function $e^{2I_0}$, studied below in section 4.8, can on the other hand be extended to both $\mathcal{R}$ and $\check{\mathcal{R}}$.

## 4.7 Functions $J_P$

We consider for $\nu \in \mathbb{D}$ and $P \sqsubset \mathbb{Z} + 1/2$ the function

$$J_P(\nu) = \frac{1}{2} \fint_{-\infty}^{\nu} d\nu \, \chi_P''(\nu)^2 = \lim_{\Lambda \to \infty} \left( -|P|^2 \log \Lambda + \frac{1}{2} \int_{-\Lambda}^{\nu} d\nu \, \chi_P''(\nu)^2 \right), \tag{74}$$

with $\chi_P$ given in (64), (57) and a path of integration contained in $\mathbb{D}$, see figure 19. The regularized integral $\fint$ is defined for convenience by subtracting the divergent logarithmic term at $-\infty$ coming from (64), (58), with $|P|$ the cardinal of $P$. The integrand in (74) is analytic in $\mathbb{D}$, and $J_P$ is independent from the path of integration. The function $J_P$ has logarithmic singularities at $2i\pi a$, $a \in \mathbb{Z} + 1/2$ since $\chi_P''(\nu)^2 d\nu$ is a meromorphic differential of the third kind with simple poles at the $2i\pi a$. Additionally, $J_\emptyset$ has the large $|\nu|$ asymptotics $J_\emptyset(\nu) \simeq \frac{e^{2\nu}}{4\pi}$ when $\text{Re}\,\nu < 0$ and $J_\emptyset(\nu) \simeq \frac{\nu^2}{\pi^2} - \frac{\log \nu}{6}$ when $\text{Re}\,\nu > 0$.

From (67), the function $J_P$ transforms under translations of $2i\pi$ as $J_P(\nu - 2i\pi n) = J_{P+n}(\nu)$ when $\text{Re}\,\nu < 0$, since the path of integration may be chosen such that $\text{Re}\,\nu < 0$ everywhere along the path. The situation is more complicated on the other side. Shifting the integration variable by $2i\pi n$ in (74), one has $J_P(\nu - 2i\pi n) = \frac{1}{2} \fint_{-\infty}^{\nu} d\nu \, \chi_P''(\nu - 2i\pi n)^2$, where the path of integration crosses the imaginary axis in the interval $(2i\pi(n-1/2), 2i\pi(n+1/2))$. Introducing

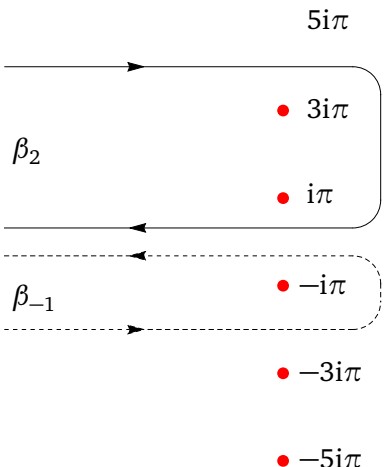

Figure 20: Paths $\beta_2$ (solid curve) and $\beta_{-1}$ (dashed curve) from $-\infty$ to $-\infty$ in $\mathbb{C}$.

$\epsilon > 0$, $\epsilon \to 0$, the path of integration can be split into a path from $-\infty$ to $2i\pi n - \epsilon$ with $\operatorname{Re} \nu < 0$ plus a path from $2i\pi n + \epsilon$ to $\nu$ with $\operatorname{Re} \nu > 0$. The translation identities (67) for $\chi_P$ then imply $J_P(\nu - 2i\pi n) = \frac{1}{2} \int_{-\infty}^{2i\pi n - \epsilon} d\nu \, \chi''_{P+n}(\nu)^2 + \frac{1}{2} \int_{2i\pi n + \epsilon}^{\nu} d\nu \, \chi''_{(P+n) \ominus B_n}(\nu)^2$. Completing the second term by adding the integral on a path from $-\infty$ to $2i\pi n + \epsilon$ crossing the imaginary axis between $-i\pi$ and $i\pi$, we arrive according to (65) at the integral of the meromorphic differential $\chi''(p)^2 d\nu$ on a path of the Riemann surface $\mathcal{R}$ (or $\check{\mathcal{R}}$), from the puncture $[-\infty, P + n]$ to the puncture $[-\infty, (P + n) \ominus B_n]$, regularized in the usual way at both punctures:

$$J_P(\nu - 2i\pi n) = J_{(P+n) \ominus B_n}(\nu) + \frac{1}{2} \oint_{\beta_n \cdot (P+n)} \chi''(p)^2 d\nu \,, \tag{75}$$

with $\oint_{\beta_n \cdot (P+n)} \chi''(p)^2 d\nu = \int_{-\infty}^{2i\pi n - \epsilon} d\nu \, \chi''_{P+n}(\nu)^2 - \int_{-\infty}^{2i\pi n + \epsilon} d\nu \, \chi''_{(P+n) \ominus B_n}(\nu)^2$. Here, $\beta_n$ is a path from $-\infty$ to $-\infty$ encircling the elements of $2i\pi B_n$ in the clockwise direction if $n > 0$ and in the anticlockwise direction if $n < 0$, see figure 20, while the path $\beta_0$ is empty. The path $\beta_n \cdot Q$, $Q \sqsubset \mathbb{Z} + 1/2$ lifting $\beta_n$ to $\mathcal{R}$ through the covering $[\nu, P] \mapsto \nu$ is contained in the sheets $\mathbb{C}_Q \cup \mathbb{C}_{Q \ominus B_n}$ and links $[-\infty, Q]$ to $[-\infty, Q \ominus B_n]$.

The integral $\oint_{\beta_n \cdot (P+n)} \chi''(\nu)^2 d\nu$ is computed in appendix A. We find

$$\frac{1}{2} \oint_{\beta_n \cdot P} \chi''(\nu)^2 d\nu = W_{P \ominus B_n} - W_{P-n}, \tag{76}$$

with

$$W_P = i\pi \Big( |P|_+^2 - |P|_-^2 - \sum_{b \in P} b \Big) - 2|P| \log 2 + \frac{1}{2} \sum_{\substack{b,c \in P \\ b \neq c}} \log \frac{\pi^2 (b - c)^2}{4} \,. \tag{77}$$

Here, $|P|_+$ (respectively $|P|_-$) denotes the number of positive (resp. negative) elements of $P$.

The identity (76) leads to $J_P(\nu - 2i\pi n) = J_{(P+n) \ominus B_n}(\nu) + W_{(P+n) \ominus B_n} - W_P$ when $\operatorname{Re} \nu > 0$. In terms of the translation operators defined in section 4.1, one has

$$T_l^{-n} J_P = J_{P+n}, \tag{78}$$
$$T_r^{-n}(J_P + W_P) = J_{(P+n) \ominus B_n} + W_{(P+n) \ominus B_n} \,.$$

The function $J_P$ has the same branch points $2i\pi a$, $a \in \mathbb{Z} + 1/2$ as $\chi_P$. Analytic continuation across the branch cut $(-i\infty, -i\pi) \cup (i\pi, i\infty)$ can then be written solely in terms of the

translation operators as in (46). Using $P \ominus B_n - n = (P - n) \ominus B_{-n}$, we obtain

$$A_n^{\mathrm{l}} J_P = J_{P \ominus B_n} + W_{P \ominus B_n} - W_{P-n}, \tag{79}$$
$$A_n^{\mathrm{r}} J_P = J_{P \ominus B_n} + W_{P \ominus B_n - n} - W_P .$$

Thus, $J_P$ is related to $J_{P \ominus B_n}$ by analytic continuation across $(2\mathrm{i}\pi(n-1/2), 2\mathrm{i}\pi(n+1/2))$, just like the functions $\chi_P$ and $\chi_{P \ominus B_n}$. The extra terms $W_{P \ominus B_n} - W_{P-n}$ and $W_{P \ominus B_n - n} - W_P$ are however distinct in general, as can be seen from the identity (205) in appendix E, and $J_P$ may not be extended to an analytic function on $\mathcal{R}$. Since the difference between the extra terms is from (205) an integer multiple of $\mathrm{i}\pi$, it is natural to consider the function $\mathrm{e}^{J_P}$ instead. This is done in the next section.

## 4.8  Functions $\mathrm{e}^{2I}$, $\mathrm{e}^{I+J}$ and $\mathrm{e}^{2J}$ defined on $\check{\mathcal{R}}$

We consider in this section functions $\mathrm{e}^{2I}$, $\mathrm{e}^{I+J}$ and $\mathrm{e}^{2J}$ [15] built from the functions $I_0$ and $J_P$ defined respectively in (68) and (74), and which are shown below to be well defined on the Riemann surface $\check{\mathcal{R}}$. These functions are used as building blocks for KPZ fluctuations in section 2.

We begin with the function $I_0$ from section 4.6. Equation (73) implies that the analytic continuation of the function $\mathrm{e}^{2I_0}$ across the cut is independent of the side from which the continuation is made, $A_n \mathrm{e}^{2I_0} = (-1)^n \mathrm{e}^{2I_0}$. It is then possible to extend $\mathrm{e}^{2I_0}$ to a meromorphic function $\mathrm{e}^{2I}$ on $\mathcal{R}$, defined as

$$\mathrm{e}^{2I}([\,v, P]) = (-1)^{|P|} \, \mathrm{e}^{2I_0(v)} . \tag{80}$$

The relations $|P \ominus B_n| = |P| + |B_n| - 2|P \cap B_n|$ and $|B_n| = |n|$ indeed imply that the change of sign from $A_n$ is equivalent to replacing $P$ by $P \ominus B_n$ in (80). Furthermore, the function $\mathrm{e}^{2I}$ verifies the symmetry relation $\mathrm{e}^{2I} \circ \mathcal{T} = \mathrm{e}^{2I}$ with $\mathcal{T}$ defined in (38), and is thus also well defined on the Riemann surface $\check{\mathcal{R}}$. In fact, since $\mathrm{e}^{2I_0(v)} = \pm\sqrt{1 + \mathrm{e}^v}$, the function $\mathrm{e}^{2I}$ can even be defined on the hyperelliptic-like Riemann surface with branch points $2\mathrm{i}\pi a$, $a \in \mathbb{Z} + 1/2$.

The function $\mathrm{e}^{2I}$ is meromorphic on $\check{\mathcal{R}}$, and has simple poles at the points $[(2\mathrm{i}\pi a)_{\mathrm{l|r}}, P]$, $a \in \mathbb{Z} + 1/2$: for $v$ close to $2\mathrm{i}\pi a$, $\mathrm{e}^{2I}([\,v, P]) \simeq \pm \mathrm{i}/\sqrt{v - 2\mathrm{i}\pi a}$, which corresponds to a pole in the local coordinate $y = \sqrt{v - 2\mathrm{i}\pi a}$. The differential defined at $p = [\,v, P] \in \check{\mathcal{R}}$ away from ramification points by $\mathrm{e}^{2I}(p) \, \mathrm{d}v$ is on the other hand holomorphic at $[(2\mathrm{i}\pi a)_{\mathrm{l|r}}, P]$, $a \in \mathbb{Z} + 1/2$, since the differential $\mathrm{d}v$ becomes proportional to $y \, \mathrm{d}y$ in the local coordinate above at $[(2\mathrm{i}\pi a)_{\mathrm{l|r}}, P]$ and compensates the pole of the function $\mathrm{e}^{2I}$.

We now consider the functions $J_P$ from section 4.7. Using the identities (207), (211) for the quantities $W_P$ and the relation $|P \ominus B_n| = |P| + \sum_{a \in B_n} \sigma_a(P)$, we obtain after some simplifications

$$A_n^{\mathrm{l}}\Big(\frac{(-1)^{|P|} V_P^2}{4^{|P|}} \, \mathrm{e}^{J_P}\Big) = \mathrm{i}^{-n} \frac{(-1)^{|P \ominus B_n|} V_{P \ominus B_n}^2}{4^{|P \ominus B_n|}} \, \mathrm{e}^{J_{P \ominus B_n}}, \tag{81}$$
$$A_n^{\mathrm{r}}\Big(\frac{(-1)^{|P|} V_P^2}{4^{|P|}} \, \mathrm{e}^{J_P}\Big) = \mathrm{i}^{n} \frac{(-1)^{|P \ominus B_n|} V_{P \ominus B_n}^2}{4^{|P \ominus B_n|}} \, \mathrm{e}^{J_{P \ominus B_n}} ,$$

where $V_P$ is the Vandermonde determinant (5). Comparison with analytic continuations (73) for $I_0$ implies that the function $\mathrm{e}^{I+J}$, defined for $v \in \mathbb{D}$, $P \sqsubset \mathbb{Z} + 1/2$ by

$$\mathrm{e}^{I+J}([\,v, P]) = \frac{(-1)^{|P|} V_P^2}{4^{|P|}} \, \mathrm{e}^{I_0(v) + J_P(v)} , \tag{82}$$

---

[15] We emphasize that only the combination of symbols $\mathrm{e}^{2I}$, $\mathrm{e}^{I+J}$ and $\mathrm{e}^{2J}$ are defined here: $I$ and $J$ alone are not, as there is no meaningful way to extend $I_0$ and $J_P$ to $\check{\mathcal{R}}$.

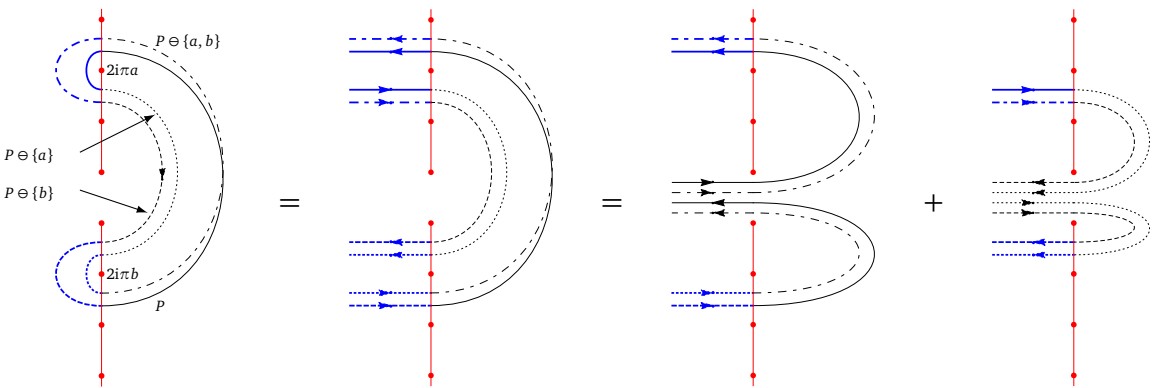

Figure 21: Decomposition of a loop $\ell_{a,b} \cdot P$, $b < 0 < a$ in terms of paths of the form $\beta_n \cdot Q$. Paths on various sheets $\mathbb{C}_P$ are represented differently.

is well defined on the Riemann surface $\mathcal{R}$. Furthermore, using (72), (78) and (211), the function $e^{I+J}$ verifies the symmetry $e^{I+J} \circ \mathcal{T} = e^{I+J}$, so that $e^{I+J}$ is also well defined on $\check{\mathcal{R}}$.

The function $e^{I+J}$ is meromorphic on $\check{\mathcal{R}}$, with the same simple poles as $e^{2I}$ at the points $[(2i\pi a)_{l|r}, P]$, $a \in \mathbb{Z}+1/2$. The poles come from logarithmic singularities at the points $\nu = 2i\pi a$ in the integral representations (68) and (74) for the functions $I_0$ and $J_P$. From the infinite sum representation (162) for $\chi_P''$, the functions $I_0$ and $J_P$ contribute each half of the logarithmic terms, and one has $e^{I+J}([\nu, P]) \propto 1/\sqrt{\nu - 2i\pi a}$ when $\nu \to 2i\pi a$. The meromorphic differential defined at $p = [\nu, P] \in \check{\mathcal{R}}$ away from ramification points by $e^{I+J}(p) \, d\nu$ is holomorphic at $[(2i\pi a)_{l|r}, P]$, $a \in \mathbb{Z} + 1/2$ for the same reason as for $e^{2I}$.

Finally, the function $e^{2J} = (e^{I+J})^2/e^{2I}$, more explicitly

$$e^{2J}([\nu, P]) = (-1)^{|P|} 2^{-4|P|} V_P^4 \, e^{2J_P(\nu)} , \tag{83}$$

is also well defined and meromorphic on $\check{\mathcal{R}}$, with simple poles at the points $[(2i\pi a)_{l|r}, P]$, $a \in \mathbb{Z} + 1/2$, while the differential $e^{2J}(p) \, d\nu$ is holomorphic.

The functions $e^{2I}$, $e^{2J}$ and $e^{I+J}$ are defined through $I_0(\nu)$, $J_P(\nu)$ as integrals on a path contained in $\mathbb{D}$ between $-\infty$ and $\nu$ according to (68), (74). An alternative point of view, which takes into account from the start that $\nu$ lives on $\check{\mathcal{R}}$, consists instead in writing directly $e^{2I}(p)$, $e^{2J}(p)$, $e^{I+J}(p)$ in terms of meromorphic differentials integrated on a path of $\check{\mathcal{R}}$ between $[-\infty, \emptyset]$ and $p$. Defining a meromorphic function $\varphi$ on $\check{\mathcal{R}}$ by $\varphi([\nu, P]) = -\frac{1}{2} \frac{1}{1+e^{-\nu}}$ and the meromorphic differentials of the third kind [16]

$$\begin{aligned}
\Omega_{2I}(q) &= \varphi(q) d\nu, \\
\Omega_{2J}(q) &= \chi''(q)^2 d\nu, \\
\Omega_{I+J}(q) &= \Big(\frac{\varphi(q)}{2} + \frac{\chi''(q)^2}{2}\Big) d\nu ,
\end{aligned} \tag{84}$$

at $q = [\nu, P]$ away from ramification points, one has

$$\begin{aligned}
e^{2I}(p) &= \exp\Big( \int_{[-\infty, \emptyset]}^{p} \Omega_{2I} \Big), \\
e^{2J}(p) &= \exp\Big( \int_{[-\infty, \emptyset]}^{p} \Omega_{2J} \Big), \\
e^{I+J}(p) &= \exp\Big( \int_{[-\infty, \emptyset]}^{p} \Omega_{I+J} \Big).
\end{aligned} \tag{85}$$

---

[16] With additional essential singularities for the points at infinity.

The fact that the functions $e^{2I}$, $e^{2J}$ and $e^{I+J}$ are well defined on $\check{\mathcal{R}}$ imply that the integrals on any loop in $\check{\mathcal{R}}$ of the meromorphic differentials $\Omega_{2I}$, $\Omega_{2J}$, $\Omega_{I+J}$ are integer multiples of $2i\pi$. For small loops around a point $[(2i\pi a)_{l|r}, P]$, this is a consequence of Cauchy's residue theorem on the Riemann surface, both $\varphi(q)d\nu$ and $\chi''(q)^2 d\nu$ having a residue $-1$ at that point [17]. For non-contractible loops, this is a non-trivial statement. For $\Omega_{2J}$, it can be checked directly for a given loop by expressing its homology class as a sum of paths of the form $\beta_n \cdot P$ from figure 20 and using (76) to compute the integrals. For example, considering the loops $\ell_{a,b} \cdot P$ which generate all loops on $\check{\mathcal{R}}$ that are also closed on $\mathcal{R}$ and are defined in section 3.8.1, see also figure 21, we obtain after some simplifications $\int_{\ell_{a,b}} \Omega_{2J} = 4i\pi(-1 + \sigma_a(P)\sigma_b(P)(\frac{\mathrm{sgn}\,a - \mathrm{sgn}\,b}{2} - \mathrm{sgn}(a-b)\frac{1+\mathrm{sgn}(ab)}{2}))$, which is an integer multiple of $4i\pi$. For loops on $\check{\mathcal{R}}$ corresponding to open paths on $\mathcal{R}$, the integral is generally only an integer multiple of $2i\pi$: for instance, the integral of $\Omega_{2J}$ on the loop $[\nu, \{1/2\}] \to [(4i\pi)_l, \{1/2\}] = [(4i\pi)_r, \{3/2\}] \to [\nu + 2i\pi, \{3/2\}]$, $\mathrm{Re}\,\nu < 0$ is equal to $-2i\pi$.

## 4.9 Functions on the Riemann surfaces $\mathcal{R}^\Delta$

In this section, we study functions on the Riemann surfaces $\mathcal{R}^\Delta$ built in section 3.8.3 by quotienting $\mathcal{R}$ by a group of involutions indexed by the elements of $\Delta \sqsubset \mathbb{Z}+1/2$. These functions are used in sections 2.3 and 2.4 for KPZ fluctuations with sharp wedge and stationary initial condition.

### 4.9.1 Function $\chi^\Delta$

Let $\Delta \sqsubset \mathbb{Z}+1/2$. The covering map $\Pi^\Delta : [\nu, P] \mapsto [\nu, P \setminus \Delta]$ from $\mathcal{R}$ to $\mathcal{R}^\Delta$ allows to define the function $\chi^\Delta = 2^{-|\Delta|}\,\mathrm{tr}_{\Pi^\Delta}\,\chi$ analytic on $\mathcal{R}^\Delta$, see section 3.4. More explicitly, for any $\nu \in \mathbb{C}$ and $Q \sqsubset \mathbb{Z}+1/2$, $Q \cap \Delta = \emptyset$, one has $\chi^\Delta([\nu, Q]) = 2^{-|\Delta|}\sum_{\substack{P \sqsubset \mathbb{Z}+1/2 \\ P \setminus \Delta = Q}} \chi_P(\nu) = 2^{-|\Delta|}\sum_{A \subset \Delta} \chi_{Q \cup A}(\nu)$. Using (64), we find

$$\chi^\Delta([\nu, P]) = \chi_P^\Delta(\nu)\,, \tag{86}$$

where $\chi_P^\Delta$, given by [18]

$$\chi_P^\Delta(\nu) = \frac{\chi_P(\nu) + \chi_{P \ominus \Delta}(\nu)}{2}\,, \tag{87}$$

generalizes the function $\chi_P$ of section 4.5. The function $\chi_P^\Delta$ has the infinite sum representation

$$\chi_P^\Delta(\nu) = \lim_{M \to \infty}\left(-\frac{4(2\pi M)^{5/2}}{15\pi} - \frac{2\nu(2\pi M)^{3/2}}{3\pi} + \frac{(\pi^2 + 3\nu^2)\sqrt{2\pi M}}{6\pi} \right. \tag{88}$$
$$\left. - \sum_{a=-M+1/2}^{M-1/2} 1_{\{a \notin \Delta\}}\,\sigma_a(P)\frac{\kappa_a^3(\nu)}{3}\right),$$

with $\sigma_a$ defined in (29) and $\kappa_a$ in (50), and verifies $\chi_P^\Delta = \chi_{P \setminus \Delta}^\Delta$.

Since $\chi^\Delta$ is analytic on $\mathcal{R}^\Delta$, the function $\chi_P^\Delta$ defined on $\mathbb{D}$ has the analytic continuation

$$A_n \chi_P^\Delta = \chi_{P \ominus (B_n \setminus \Delta)}^\Delta \tag{89}$$

---

[17]The function $\nu \mapsto -\frac{1}{2}\frac{1}{1+e^{-\nu}}$ has residues $-1/2$ in $\mathbb{C}$ instead. The distinction with $\varphi$ comes from the fact that the projection of a loop on $\check{\mathcal{R}}$ encircling $[(2i\pi a)_{l|r}, P]$ to the complex plane by the covering map $[\nu, P] \mapsto \nu$ encircles $2i\pi a$ an even number of times, see figure 3. In terms of the local coordinate $y = \sqrt{\nu - 2i\pi a}$ at $[(2i\pi a)_{l|r}, P]$, this is a consequence of $d\nu/(\nu - 2i\pi a) = 2dy/y$.

[18]For later convenience, we define $\chi_P^\Delta$ in (87) in terms of $P \ominus \Delta$ instead of $P \cup \Delta$ when $P \cap \Delta \neq \emptyset$.

across branch cuts. Alternatively, (89) can be proved directly from (65) using the general identity $\chi_P^\Delta = \chi_{P\ominus A}^\Delta$, $A \subset \Delta$ with $A = B_n \cap \Delta$. From (67), the functions $\chi_P^\Delta$ also verify the shift identities

$$
\begin{aligned}
T_{\mathrm l}^{-n}\chi_P^\Delta &= \chi_{P+n}^{\Delta+n}, \\
T_{\mathrm r}^{-n}\chi_P^\Delta &= \chi_{(P+n)\ominus(B_n\backslash(\Delta+n))}^{\Delta+n},
\end{aligned}
\tag{90}
$$

where we used again $\chi_P^\Delta = \chi_{P\ominus A}^\Delta$, $A \subset \Delta$ for the second line. In terms of the collection $\overline{\mathcal{R}}$ of all Riemann surfaces $\mathcal{R}^\Delta$ of section 3.8.3, of the function $\overline{\chi}([\nu,(P,\Delta)]) = \chi_P^\Delta(\nu)$ defined on $\overline{\mathcal{R}}$, and of the holomorphic map $\overline{\mathcal{T}}$ on $\overline{\mathcal{R}}$ defined in (44), the identity (90) simply rewrites as $\overline{\chi}\circ\overline{\mathcal{T}} = \overline{\chi}$.

Finally, we also define functions $\chi'^\Delta$, $\chi''^\Delta$ from derivatives of $\chi_P^\Delta$ as

$$
\begin{aligned}
\chi'^\Delta([\nu,P]) &= \chi_P'^\Delta(\nu), \\
\chi''^\Delta([\nu,P]) &= \chi_P''^\Delta(\nu).
\end{aligned}
\tag{91}
$$

The function $\chi'^\Delta$ is analytic on $\mathcal{R}^\Delta$ while $\chi''^\Delta$ is meromorphic on $\mathcal{R}^\Delta$. Both verify the same translation symmetries as $\chi^\Delta$.

### 4.9.2 Functions $J_P^\Delta$

We now consider $J_P^\Delta$ generalizing the function $J_P$ of section 4.7, defined from the second derivative $\chi_P''^\Delta$ of $\chi_P^\Delta$ as

$$
J_P^\Delta(\nu) = \frac{1}{2}\int_{-\infty}^{\nu} \mathrm d\nu\, \chi_P''^\Delta(\nu)^2
\tag{92}
$$

$$
= \lim_{\Lambda\to\infty}\left(-\frac{(|P|+|P\ominus\Delta|)^2}{4}\log\Lambda + \frac{1}{2}\int_{-\Lambda}^{\nu}\mathrm d\nu\,\chi_P''^\Delta(\nu)^2\right),
$$

for $\nu \in \mathbb{D}$, with a path of integration contained in $\mathbb{D}$. One has $J_P^\Delta = J_{P\backslash\Delta}^\Delta$.

The same reasoning as in section 4.7 gives for any $n \in \mathbb{Z}$ the identities $J_P^\Delta(\nu - 2\mathrm i\pi n) = J_{P+n}^{\Delta+n}(\nu)$ when $\mathrm{Re}\,\nu < 0$ and $J_P^\Delta(\nu - 2\mathrm i\pi n) = J_{(P+n)\ominus(B_n\backslash(\Delta+n))}^{\Delta+n}(\nu) + \frac{1}{2}\int_{\beta_n\cdot(P+n)}\chi''^{\Delta+n}(p)\mathrm d\nu$ with $\beta_n\cdot(P+n)$ the lift to the sheet $P+n$ of $\mathcal{R}^{\Delta+n}$ of the path in the complex plane $\beta_n$ from figure 20. The integral over $\beta_n\cdot(P+n)$ can be computed similarly as in appendix A. We obtain eventually

$$
\begin{aligned}
T_{\mathrm l}^{-n}J_P^\Delta &= J_{P+n}^{\Delta+n}, \\
T_{\mathrm r}^{-n}J_P^\Delta &= J_{(P+n)\ominus(B_n\backslash(\Delta+n))}^{\Delta+n} + W_{(P+n)\ominus(B_n\backslash(\Delta+n))}^{\Delta+n} - W_P^\Delta,
\end{aligned}
\tag{93}
$$

where

$$
W_P^\Delta = \frac{\mathrm i\pi}{4}\left((|P|_+ + |P\ominus\Delta|_+)^2 - (|P|_- + |P\ominus\Delta|_-)^2\right) - \frac{\mathrm i\pi}{2}\left(\sum_{b\in P}b + \sum_{b\in P\ominus\Delta}b\right)
\tag{94}
$$

$$
-(|P|+|P\ominus\Delta|)\log 2 + \frac{1}{4}\sum_{\substack{b,c\in P \\ b\neq c}}\log\frac{\pi^2(b-c)^2}{4} + \frac{1}{4}\sum_{\substack{b,c\in P\ominus\Delta \\ b\neq c}}\log\frac{\pi^2(b-c)^2}{4}
$$

generalizes the constants $W_P = W_P^\emptyset$ of (77). Using (46), the analytic continuation of $J_P^\Delta$ across branch cuts is given by

$$
\begin{aligned}
A_n^{\mathrm l}J_P^\Delta &= J_{P\ominus(B_n\backslash\Delta)}^\Delta + W_{P\ominus(B_n\backslash\Delta)}^\Delta - W_{P-n}^{\Delta-n}, \\
A_n^{\mathrm r}J_P^\Delta &= J_{P\ominus(B_n\backslash\Delta)}^\Delta + W_{P\ominus(B_n\backslash\Delta)-n}^{\Delta-n} - W_P^\Delta.
\end{aligned}
\tag{95}
$$

As in section 4.7, the analytic continuation from each side of the branch cuts does not give the same result, so that $J_P^\Delta$ may not be extended to $\mathcal{R}^\Delta$. For the application to KPZ in sections 2.3 and 2.4, however, only $e^{2J_P^\Delta}$ is needed. Using the identity (213), we find that the function $e^{2J^\Delta}$ generalizing $e^{2J} = e^{2J^\emptyset}$ and defined by

$$e^{2J^\Delta}([\nu, P]) = (i/4)^{2|P\setminus\Delta|}\left(\prod_{a\in P\setminus\Delta}\prod_{\substack{b\in P\cup\Delta\\b\neq a}}\left(\frac{2i\pi a}{4} - \frac{2i\pi b}{4}\right)^2\right)e^{2J_P^\Delta(\nu)} \tag{96}$$

is well defined on $\mathcal{R}^\Delta$. The prefactor of $e^{2J_P^\Delta(\nu)}$, which depends on $P$ and $\Delta$ only through $\Delta$ and $P\setminus\Delta$ since $P\cup\Delta = (P\setminus\Delta)\cup\Delta$, has been chosen equal to 1 when $P=\emptyset$. The function $e^{2J^\Delta}$ can then be expressed alternatively as

$$e^{2J^\Delta}(p) = \exp\left(\fint_{[-\infty,\emptyset]}^{p} \Omega_{2J}^\Delta\right), \tag{97}$$

independently from the path between $[-\infty,\emptyset]$ and $p$ in $\mathcal{R}^\Delta$, with $\Omega_{2J}^\Delta$ the meromorphic differential of the third kind

$$\Omega_{2J}^\Delta([\nu, P]) = \chi''^\Delta([\nu, P])^2\, d\nu. \tag{98}$$

The function $e^{2J^\Delta}$ has the simple poles $[(2i\pi a)_{l|r}, P]$, $a\in(\mathbb{Z}+1/2)\setminus\Delta$, $P\sqsubset(\mathbb{Z}+1/2)\setminus\Delta$, while the differential $e^{2J^\Delta}([\nu, P])\,d\nu$ is holomorphic on $\mathcal{R}^\Delta$.

Additionally, for any $m\in\mathbb{Z}$, the function $e^{2J^\Delta}$ verifies from (93), (212)

$$e^{2J^\Delta}([\nu-2i\pi m, P]) = \begin{cases} e^{2J^{\Delta+m}}([\nu, P+m]) & \mathrm{Re}\,\nu<0 \\ e^{2J^{\Delta+m}}([\nu,(P+m)\ominus(B_m\setminus(\Delta+m))]) & \mathrm{Re}\,\nu>0 \end{cases}. \tag{99}$$

Extending $e^{2J^\Delta}$ to the collection $\overline{\mathcal{R}}$ of all $\mathcal{R}^\Delta$ by $\overline{e^{2J}}([\nu,(P,\Delta)]) = e^{2J^\Delta}([\nu, P])$, the identity (99) is equivalent to the symmetry $\overline{e^{2J}}\circ\overline{\mathcal{T}} = \overline{e^{2J}}$ with $\overline{\mathcal{T}}$ defined in (44).

## 4.10   Functions $K_{P,Q}$, $K_{P,Q}^{\Delta,\Gamma}$ and $e^{2K^{\Delta,\Gamma}}$

In this section, we study functions on products of Riemann surfaces needed for joint statistics of the KPZ height at multiple times.

### 4.10.1   Functions $K_{P,Q}$

Given two finite sets of half-integers $P$ and $Q$, and $(\nu,\mu)$ in the simply connected domain

$$\mathbb{D}_2 = \{(\nu,\mu)\in\mathbb{D}\times\mathbb{D}, (\mathrm{Re}\,\nu\neq\mathrm{Re}\,\mu) \text{ or } (\mathrm{Re}\,\nu=\mathrm{Re}\,\mu \text{ and } \mathrm{Im}(\nu-\mu)\in(0,2\pi))\}, \tag{100}$$

we consider the functions

$$K_{P,Q}(\nu,\mu) = \frac{1}{2}\int_{-\infty}^{0} du\, \chi_P''(u+\nu)\chi_Q''(u+\mu) \tag{101}$$

$$= \lim_{\Lambda\to\infty}\left(-|P||Q|\log\Lambda + \frac{1}{2}\int_{-\Lambda}^{0} du\, \chi_P''(u+\nu)\chi_Q''(u+\mu)\right).$$

The path of integration is chosen such that $u+\nu$ and $u+\mu$ both stay in $\mathbb{D}$, which is possible [19] if $(\nu,\mu)\in\mathbb{D}_2$, see figure 22, and $K_{P,Q}$ is thus analytic in $\mathbb{D}_2$. When $P=Q$, one recovers $J_P(\nu) = \lim_{\mu\to\nu}K_{P,P}(\nu,\mu)$.

---

[19]At this point, $K_{P,Q}(\nu,\mu)$ is in fact analytic in a domain larger than $\mathbb{D}_2$: in the sector $\mathrm{Re}\,\nu=\mathrm{Re}\,\mu$, $\mathrm{Im}(\nu-\mu)$ need not be restricted when the real parts are negative, and one requires only $\mathrm{Im}(\nu-\mu)\in(-2\pi,2\pi)$ when the real parts are positive. Additional terms contributed later on by analytic continuation are however only analytic in $\mathbb{D}_2$, and it is thus convenient to add this restriction from the beginning.

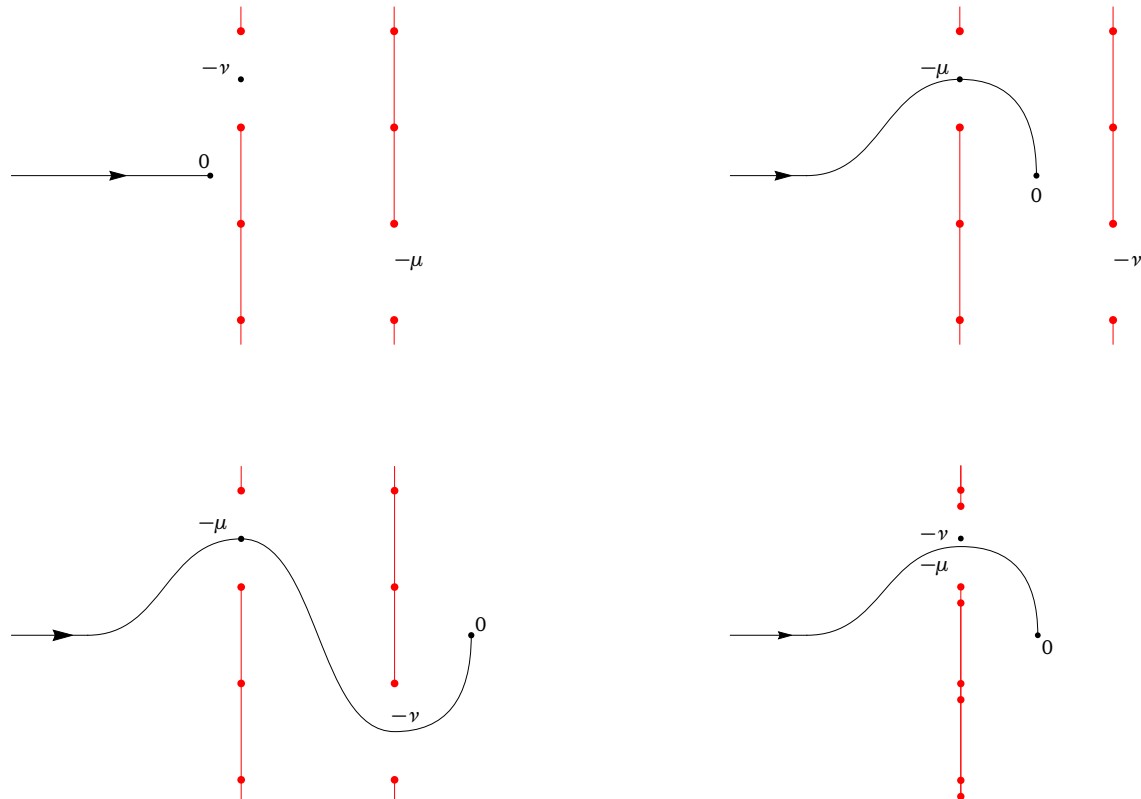

Figure 22: Possible choices for the path of integration in the definition (101) for $K_{P,Q}$. The red, vertical lines are the branch cuts in the variable $u$ of $\chi_P''(u + v)$ and $\chi_Q''(u + \mu)$, and the red dots on the cuts are the associated branch points $-v + 2i\pi a$, $-\mu + 2i\pi a$, $a \in \mathbb{Z} + 1/2$. The black curves, ending at $u = 0$, represent the path for the variable $u$. From left to right, top to bottom, the four graphs represent respectively the situations $\operatorname{Re}\mu < \operatorname{Re}v < 0$; $\operatorname{Re}v < 0 < \operatorname{Re}\mu$; $\operatorname{Re}\mu > \operatorname{Re}v > 0$; $\operatorname{Re}v = \operatorname{Re}\mu > 0$ with $-\pi < \operatorname{Im}(v - \mu) < \pi$.

Crossing the branch cuts $\mu, v \in (-i\infty, -i\pi) \cup (i\pi, i\infty)$, $v - \mu \in (-i\infty, 0) \cup (i\pi, i\infty)$ produces new functions analytic in $\mathbb{D}_2$. In order to obtain explicit formulas for the various analytic continuations of $K_{P,Q}(v, \mu)$, it is useful to study first shifts by integer multiples of $2i\pi$ in the variables $v$ and $\mu$. When both $\operatorname{Re}v < 0$ and $\operatorname{Re}\mu < 0$, the path of integration in (101) can be chosen such that $\operatorname{Re}(u + v) < 0$, $\operatorname{Re}(u + \mu) < 0$ everywhere, see figure 22 top left, and one has from (67) the identity $K_{P,Q}(v - 2i\pi n, \mu - 2i\pi m) = K_{P+n,Q+m}(v, \mu)$. Similar reasonings leads to $K_{P,Q}(v - 2i\pi n, \mu) = K_{P+n,Q}(v, \mu)$ when $\operatorname{Re}v < 0$ and $K_{P,Q}(v, \mu - 2i\pi m) = K_{P,Q+m}(v, \mu)$ when $\operatorname{Re}\mu < 0$. Shifting variables with a positive real part is more complicated. Indeed, the argument of the functions $\chi_P''$, $\chi_Q''$ has then a positive real part on some portions of the path of integration in (101) and a negative real part on other portions of the path, so that (67) gives several distinct expressions for the shifts along the path. After some rewriting, one finds in all cases the identity

$$K_{P,Q}(v - 2i\pi n, \mu - 2i\pi m) = K_{\tilde{P},\tilde{Q}}(v, \mu) \tag{102}$$

$$+ \frac{1}{2} \oint_{\gamma_{n,m}} du \, \mathcal{A}_u(\chi_{P+n}''(\cdot + v)\chi_{Q+m}''(\cdot + \mu)),$$

with $\tilde{P} = P + n$ when $\operatorname{Re}v < 0$, $\tilde{P} = (P + n) \ominus B_n$ when $\operatorname{Re}v > 0$ and $\tilde{Q} = Q + m$ when $\operatorname{Re}\mu < 0$,

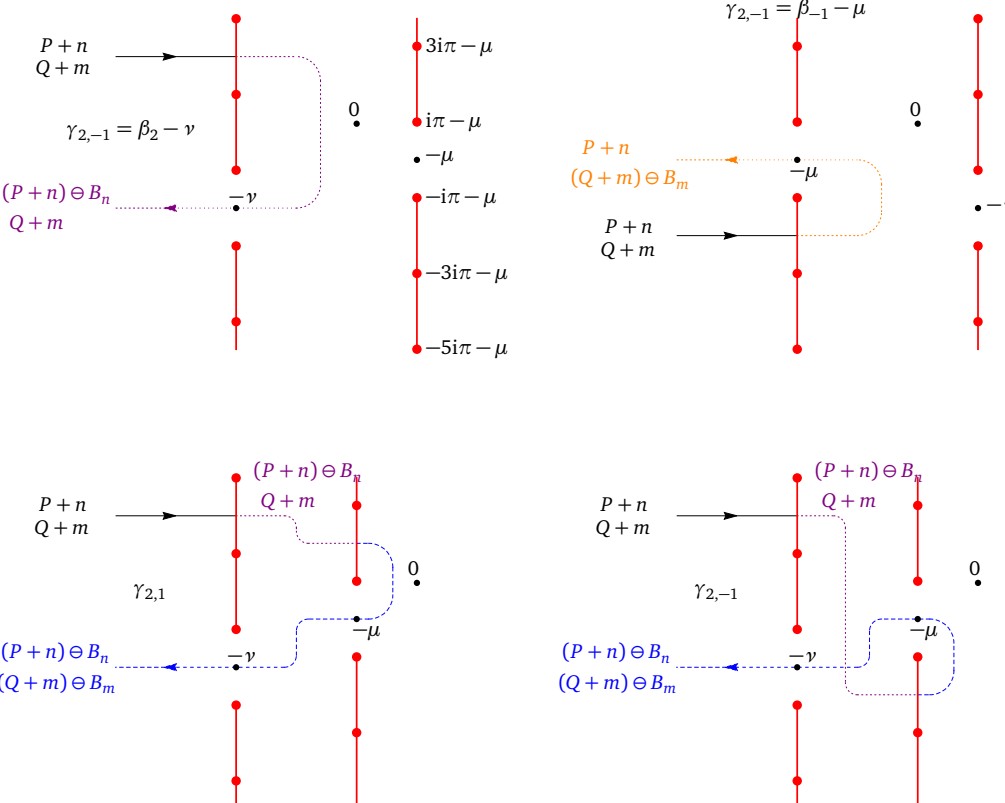

Figure 23: Path $\gamma_{n,m}$ in (102) plotted for some choices of $\nu, \mu \in \mathbb{D}$, $n, m \in \mathbb{Z}$. From left to right, top to bottom, the graphs represent $\gamma_{2,-1} = \beta_2 - \nu$ for $\mathrm{Re}\,\mu < 0 < \mathrm{Re}\,\nu$, $\gamma_{2,-1} = \beta_{-1} - \mu$ for $\mathrm{Re}\,\nu < 0 < \mathrm{Re}\,\mu$, $\gamma_{2,1}$ for $0 < \mathrm{Re}\,\mu < \mathrm{Re}\,\nu$, and $\gamma_{2,-1}$ for $0 < \mathrm{Re}\,\mu < \mathrm{Re}\,\nu$. The smaller, black dots represent the points $0, -\nu, -\mu$. The bigger, red dots represent the branch points $-\nu + 2\mathrm{i}\pi a$, $-\mu + 2\mathrm{i}\pi a$, $a \in \mathbb{Z} + 1/2$ of $\chi''_{P+n}(\cdot + \nu)\chi''_{Q+m}(\cdot + \mu)$, and the vertical, red lines the associated branch cuts. The solid / dotted / dashed portions of the path $\gamma_{n,m}$ correspond to distinct values of the sets $\hat{P}, \hat{Q}$ in $\chi''_{\hat{P}}(u + \nu)\chi''_{\hat{Q}}(u + \mu) = \mathcal{A}_u(\chi''_{P+n}(\cdot + \nu)\chi''_{Q+m}(\cdot + \mu))$ which are indicated next to the paths.

$\tilde{Q} = (Q + m) \ominus B_m$ when $\mathrm{Re}\,\mu > 0$. We used the notation $\int_\gamma \mathrm{d}u\, \mathcal{A}_u f$ for the integral of the analytic continuation of a function $f$ along a path $\gamma$. The path $\gamma_{n,m}$ in the complex plane depends on the real parts of $\nu$ and $\mu$, see figure 23. When $\mathrm{Re}\,\nu$ and $\mathrm{Re}\,\nu$ are both negative, the path is empty and the integral in (102) is equal to zero. When $\mathrm{Re}\,\nu$ and $\mathrm{Re}\,\nu$ do not have the same sign, the path reduces to $\gamma_{n,m} = \beta_n - \nu$ if $\mathrm{Re}\,\mu < 0 < \mathrm{Re}\,\nu$ and $\gamma_{n,m} = \beta_m - \mu$ if $\mathrm{Re}\,\nu < 0 < \mathrm{Re}\,\mu$, with $\beta_n$ defined in section 4.7, compare with figure 20. Finally, when $\mathrm{Re}\,\nu$ and $\mathrm{Re}\,\mu$ are both positive, the path goes from $-\infty$ to $-\infty$ and crosses the lines $\mathrm{Re}(u + \nu) = 0$ and $\mathrm{Re}(u + \mu) = 0$ along the path at successive points $2\mathrm{i}\pi n - \nu$, $2\mathrm{i}\pi m - \mu$, $-\mu$, $-\nu$ (respectively $2\mathrm{i}\pi m - \mu$, $2\mathrm{i}\pi n - \nu$, $-\nu$, $-\mu$) if $\mathrm{Re}\,\nu > \mathrm{Re}\,\mu$ (resp. $\mathrm{Re}\,\nu < \mathrm{Re}\,\mu$).

It is sufficient for the following to consider shifts of $\nu$ and $\mu$ separately, and compute the integrals over the paths $\gamma_{n,0}$ and $\gamma_{0,m}$ only. Anticipating the result, we define the function

$$W_{P,Q}(z) = 2|P|I_0(z) + 2\sum_{a \in P}\sum_{b \in Q}\log\left(\frac{\kappa_{b-a+1/2}(z)}{\sqrt{8}}\right), \tag{103}$$

and the coefficients

$$Z_{P,Q} = \sum_{a \in P} \left( -i\pi(a - 1/2) + 2i\pi \, \text{sgn}(a) \, |Q \cap B_{a-1/2}| \right). \tag{104}$$

The function $W_{P,Q}$ is analytic in $\mathbb{D}$ so that $(\nu, \mu) \mapsto W_{P,Q}(\mu - \nu + i\pi)$ and $(\nu, \mu) \mapsto W_{P,Q}(\nu - \mu - i\pi)$ are both analytic in $\mathbb{D}_2$. Additionally, using (54), (72) and $\text{sgn}(n)|(Q - a + 1/2) \cap B_n| = \text{sgn}(n + a)|Q \cap B_{n+a-1/2}| - \text{sgn}(a)|Q \cap B_{a-1/2}|$, one has

$$W_{P,Q}(z + 2i\pi n) = W_{P+n,Q}(z) + 1_{\{\text{Re} \, z > 0\}} (Z_{P+n,Q} - Z_{P,Q}). \tag{105}$$

The integral over $\gamma_{n,0}$ is computed in appendix B. There, one finds

$$\frac{1}{2} \oint_{\gamma_{n,0}} du \, \mathcal{A}_u(\chi''_{P+n}(\cdot + \nu)\chi''_Q(\cdot + \mu)) \tag{106}$$

$$= 1_{\{\text{Re} \, \nu > 0\}} \Big( W_{(P+n)\ominus B_n,Q}(\mu - \nu + i\pi) - W_{P+n,Q}(\mu - \nu + i\pi)$$

$$+ 1_{\{\text{Re} \, \mu > \text{Re} \, \nu\}} (Z_{(P+n)\ominus B_n,Q} - Z_{P+n,Q}) \Big)$$

and

$$\frac{1}{2} \oint_{\gamma_{0,m}} du \, \mathcal{A}_u(\chi''_P(\cdot + \nu)\chi''_{Q+m}(\cdot + \mu)) \tag{107}$$

$$= 1_{\{\text{Re} \, \mu > 0\}} \Big( W_{(Q+m)\ominus B_m+1,P}(\nu - \mu - i\pi) - W_{Q+m+1,P}(\nu - \mu - i\pi)$$

$$+ 1_{\{\text{Re} \, \nu > \text{Re} \, \mu\}} (Z_{(Q+m)\ominus B_m+1,P} - Z_{Q+m+1,P}) \Big).$$

As usual, it is useful to interpret (102), (106), (107) in terms of translation operators mapping $K_{P,Q}$ to other functions analytic in $\mathbb{D}_2$. We define the operators $T^n_{i^{\pm}_{l|r}}$ with $i = 1$ corresponding to translation in the first variable, $i = 2$ to translation in the second variable, and the indices $l|r, \pm$ indicating the sector for $(\text{Re} \, \nu, \text{Re} \, \mu)$ in which the translation is initially applied before reconstructing functions defined in $\mathbb{D}_2$ by analytic continuation, see table 1. All eight operators are in principle distinct, even though some of them coincide on $K_{P,Q}$. Writing $\nu$ for the first variable and $\mu$ for the second variable of the function $K_{P,Q}$, one has

$$T^{-n}_{1^{\pm}_l} K_{P,Q} = K_{P+n,Q}, \tag{108}$$

$$T^{-n}_{1^{-}_r} K_{P,Q} = K_{(P+n)\ominus B_n,Q} + W_{(P+n)\ominus B_n,Q}(\mu - \nu + i\pi) - W_{P+n,Q}(\mu - \nu + i\pi),$$

$$+ Z_{(P+n)\ominus B_n,Q} - Z_{P+n,Q}$$

$$T^{-n}_{1^{+}_r} K_{P,Q} = K_{(P+n)\ominus B_n,Q} + W_{(P+n)\ominus B_n,Q}(\mu - \nu + i\pi) - W_{P+n,Q}(\mu - \nu + i\pi)$$

for translations in the first variable and

$$T^{-m}_{2^{\pm}_l} K_{P,Q} = K_{P,Q+m}, \tag{109}$$

$$T^{-m}_{2^{-}_r} K_{P,Q} = K_{P,(Q+m)\ominus B_m} + W_{(Q+m)\ominus B_m+1,P}(\nu - \mu - i\pi) - W_{Q+m+1,P}(\nu - \mu - i\pi),$$

$$+ Z_{(Q+m)\ominus B_m+1,P} - Z_{Q+m+1,P}$$

$$T^{-m}_{2^{+}_r} K_{P,Q} = K_{P,(Q+m)\ominus B_m} + W_{(Q+m)\ominus B_m+1,P}(\nu - \mu - i\pi) - W_{Q+m+1,P}(\nu - \mu - i\pi)$$

for translations in the second variable.

Analytic continuation for the first and second variable across the branch cuts $(2i\pi(n - 1/2), 2i\pi(n + 1/2))$ can be expressed in terms of operators $A^{i^{\pm}_{l|r}}_n$, with $i = 1$ and $i = 2$

Table 1: Translation operators $T^n$, $T^m$ and analytic continuation operators $A_n$, $A_m$ defined on functions $f$ analytic in $\mathbb{D}_2$. The translation operators verify $(T^n f)(\nu, \mu) = f(\nu + 2i\pi n, \mu)$ (first four operators) or $(T^m f)(\nu, \mu) = f(\nu, \mu + 2i\pi m)$ (last four operators) in the specified sector for $(\operatorname{Re}\nu, \operatorname{Re}\mu)$. The operators $A_n$ (first four operators) correspond to analytic continuation in the first variable across the cut $\nu \in (2i\pi(n - 1/2), 2i\pi(n + 1/2))$ and the operators $A_m$ (last four operators) correspond to analytic continuation in the second variable across the cut $\mu \in (2i\pi(m - 1/2), 2i\pi(m + 1/2))$, starting from the specified sector for $(\operatorname{Re}\nu, \operatorname{Re}\mu)$.

| Operators | | Sector |
|:---:|:---:|:---:|
| $T^n_{1^-_l}$ | $A^{1^-_l}_n$ | $\operatorname{Re}\nu < \min(0, \operatorname{Re}\mu)$ |
| $T^n_{1^+_l}$ | $A^{1^+_l}_n$ | $\operatorname{Re}\mu < \operatorname{Re}\nu < 0$ |
| $T^n_{1^-_r}$ | $A^{1^-_r}_n$ | $0 < \operatorname{Re}\nu < \operatorname{Re}\mu$ |
| $T^n_{1^+_r}$ | $A^{1^+_r}_n$ | $\max(0, \operatorname{Re}\mu) < \operatorname{Re}\nu$ |
| $T^m_{2^-_l}$ | $A^{2^-_l}_m$ | $\operatorname{Re}\mu < \min(0, \operatorname{Re}\nu)$ |
| $T^m_{2^+_l}$ | $A^{2^+_l}_m$ | $\operatorname{Re}\nu < \operatorname{Re}\mu < 0$ |
| $T^m_{2^-_r}$ | $A^{2^-_r}_m$ | $0 < \operatorname{Re}\mu < \operatorname{Re}\nu$ |
| $T^m_{2^+_r}$ | $A^{2^+_r}_m$ | $\max(0, \operatorname{Re}\nu) < \operatorname{Re}\mu$ |

corresponding to analytic continuation respectively in the first and in the second variable, and the indices $l|r$, $\pm$ indicating the sector for $(\operatorname{Re}\nu, \operatorname{Re}\mu)$ from which the analytic continuation is performed, see table 1. The same reasoning as in section 4.1 gives

$$A^{i^\pm_l}_n = T^{-n}_{i^\pm_r} T^n_{i^\pm_l} \tag{110}$$

$$A^{i^\pm_r}_n = T^{-n}_{i^\pm_l} T^n_{i^\pm_r} \tag{111}$$

for $i = 1, 2$. Using (108), (109), we observe that the analytic continuation for $K_{P,Q}$ is in fact independent of the side $l|r$ from which the analytic continuation is made, and we simply write $A^{i^\pm}_n$ instead of $A^{i^\pm_{l|r}}_n$. Writing $\nu$ for the first variable and $\mu$ for the second variable of $K_{P,Q}$, we obtain

$$A^{1^-}_n (K_{P,Q} + W_{P,Q}(\mu - \nu + i\pi) + Z_{P,Q}) = K_{P \ominus B_n, Q} + W_{P \ominus B_n, Q}(\mu - \nu + i\pi) + Z_{P \ominus B_n, Q}$$
$$A^{1^+}_n (K_{P,Q} + W_{P,Q}(\mu - \nu + i\pi)) = K_{P \ominus B_n, Q} + W_{P \ominus B_n, Q}(\mu - \nu + i\pi) \tag{112}$$

for analytic continuations in the first variable and

$$A^{2^-}_n (K_{P,Q} + W_{Q+1,P}(\nu - \mu - i\pi) + Z_{Q+1,P})$$
$$= K_{P, Q \ominus B_n} + W_{Q \ominus B_n + 1, P}(\nu - \mu - i\pi) + Z_{Q \ominus B_n + 1, P}$$
$$A^{2^+}_n (K_{P,Q} + W_{Q+1,P}(\nu - \mu - i\pi)) = K_{P, Q \ominus B_n} + W_{Q \ominus B_n + 1, P}(\nu - \mu - i\pi) \tag{113}$$

for analytic continuations in the second variable. In particular, one has $(A^{1^-}_n)^2 K_{P,Q} = (A^{1^+}_n)^2 K_{P,Q} = (A^{2^-}_n)^2 K_{P,Q} = (A^{2^+}_n)^2 K_{P,Q} = K_{P,Q}$, and the branch points $\nu = 2i\pi a$, $\mu = 2i\pi a$, $a \in \mathbb{Z} + 1/2$ of $K_{P,Q}$ are of square root type.

Analytic continuation for $\nu$ across $(\mu + 2i\pi n, \mu + 2i\pi(n+1))$ is represented by the operator $D_n^{l,\pm}$ (respectively $D_n^{r,\pm}$) if the analytic continuation is made from the sector $\mathrm{Re}\,\nu < 0$ (resp. $\mathrm{Re}\,\nu > 0$), the sign $-$ corresponding to $\nu$ crossing the cut from the left and the sign $+$ from the right. Analytic continuation for $\mu$ across $(\nu + 2i\pi(n-1), \nu + 2i\pi n)$ is represented by the operator $D_{-n}^{l,\pm}$ (respectively $D_{-n}^{r,\pm}$) if the analytic continuation is made from the sector $\mathrm{Re}\,\mu < 0$ (resp. $\mathrm{Re}\,\mu > 0$). In terms of translation operators, one has

$$D_n^{l,\pm} = T_{1_l^\mp}^{-n} T_{1_l^\pm}^n, \tag{114}$$
$$D_n^{r,\pm} = T_{1_r^\mp}^{-n} T_{1_r^\pm}^n,$$

and one finds

$$
\begin{aligned}
D_n^{l,\pm} K_{P,Q} &= K_{P,Q}, \\
D_n^{r,-} K_{P,Q} &= K_{P,Q} + Z_{P \ominus B_n - n, Q} - Z_{P - n, Q}, \\
D_n^{r,+} K_{P,Q} &= K_{P,Q} - Z_{P \ominus B_n - n, Q} + Z_{P - n, Q}.
\end{aligned}
\tag{115}
$$

### 4.10.2 Functions $K_{P,Q}^{\Delta,\Gamma}$

We now introduce a generalization of $K_{P,Q}$ depending on two finite sets $\Delta, \Gamma \sqsubset \mathbb{Z} + 1/2$ similar to the generalization from $J_P$ to $J_P^\Delta$ in section 4.9.2. We define

$$K_{P,Q}^{\Delta,\Gamma}(\nu,\mu) = \frac{1}{2} \int_{-\infty}^0 \mathrm{d}u\, \chi_P''^\Delta(u+\nu) \chi_Q''^\Gamma(u+\mu), \tag{116}$$

with $\chi_P^\Delta$ defined in (87). We show in the following that $e^{2K_{P,Q}^{\Delta,\Gamma}}$ can be extended to a meromorphic function on $\mathcal{R}^\Delta \times \mathcal{R}^\Gamma$. More precisely, introducing the function

$$\Upsilon_{P,Q}^{\Delta,\Gamma}(\nu,\mu) = \frac{e^{2K_{P,Q}^{\Delta,\Gamma}(\nu,\mu)}}{(1 - e^{\mu-\nu})^{|P \setminus \Delta|}(1 - e^{\nu-\mu})^{|Q \setminus \Gamma|}} \tag{117}$$

$$\times \frac{\left(\prod_{a \in P \setminus \Delta} \prod_{b \in Q \setminus \Gamma} \left(\frac{2i\pi b - \mu}{4} - \frac{2i\pi a - \nu}{4}\right)\right) \left(\prod_{a \in P \cup \Delta} \prod_{b \in Q \cup \Gamma} \left(\frac{2i\pi b - \mu}{4} - \frac{2i\pi a - \nu}{4}\right)\right)}{\left(\prod_{a \in \Delta} \prod_{b \in \Gamma} \left(\frac{2i\pi b - \mu}{4} - \frac{2i\pi a - \nu}{4}\right)\right)},$$

which verifies $\Upsilon_{P,Q}^{\Delta,\Gamma}(\nu,\mu) = \Upsilon_{P \setminus \Delta, Q \setminus \Gamma}^{\Delta,\Gamma}(\nu,\mu)$, the function $e^{2K^{\Delta,\Gamma}}$ defined by

$$(e^{2K^{\Delta,\Gamma}})([\nu, P], [\mu, Q]) = \Upsilon_{P,Q}^{\Delta,\Gamma}(\nu,\mu) \tag{118}$$

is shown below to be meromorphic on $\mathcal{R}^\Delta \times \mathcal{R}^\Gamma$ when $|\Delta| \in 2\mathbb{N}$ and $|\Gamma| \in 2\mathbb{N}$, which is the case needed for KPZ fluctuations.

The function $K_{P,Q}^{\Delta,\Gamma}$ can be expressed in terms of $K_{P,Q}$ as

$$K_{P,Q}^{\Delta,\Gamma}(\nu,\mu) = \frac{K_{P,Q}(\nu,\mu)}{4} + \frac{K_{P \ominus \Delta, Q}(\nu,\mu)}{4} + \frac{K_{P,Q \ominus \Gamma}(\nu,\mu)}{4} + \frac{K_{P \ominus \Delta, Q \ominus \Gamma}(\nu,\mu)}{4}, \tag{119}$$

and verifies $K_{P,Q}^{\Delta,\Gamma}(\nu,\mu) = K_{P\setminus\Delta,Q\setminus\Gamma}^{\Delta,\Gamma}(\nu,\mu)$. Tedious computations using (108), (109), (112), (113), (115) then lead for shifts by integer multiples of $2\mathrm{i}\pi$ to

$$
\begin{aligned}
T_{1_{l}^{\pm}}^{-n}\Upsilon_{P,Q}^{\Delta,\Gamma} &= \Upsilon_{P+n,Q}^{\Delta+n,\Gamma}, \\
T_{1_{r}^{\pm}}^{-n}\Upsilon_{P,Q}^{\Delta,\Gamma} &= \Upsilon_{(P+n)\ominus(B_{n}\setminus(\Delta+n)),Q}^{\Delta+n,\Gamma}, \\
T_{2_{l}^{\pm}}^{-n}\Upsilon_{P,Q}^{\Delta,\Gamma} &= \Upsilon_{P,Q+n}^{\Delta,\Gamma+n}, \\
T_{2_{r}^{\pm}}^{-n}\Upsilon_{P,Q}^{\Delta,\Gamma} &= (-1)^{|\Delta||B_{-n}\setminus\Gamma|}\Upsilon_{P,(Q+n)\ominus(B_{n}\setminus(\Gamma+n))}^{\Delta,\Gamma+n}
\end{aligned}
\tag{120}
$$

and for the analytic continuations to [20]

$$
\begin{aligned}
A_{n}^{1}\Upsilon_{P,Q}^{\Delta,\Gamma} &= \Upsilon_{P\ominus(B_{n}\setminus\Delta),Q}^{\Delta,\Gamma}, \\
A_{n}^{2}\Upsilon_{P,Q}^{\Delta,\Gamma} &= (-1)^{|\Delta||B_{n}\setminus\Gamma|}\Upsilon_{P,Q\ominus(B_{n}\setminus\Gamma)}^{\Delta,\Gamma}, \\
D_{n}\Upsilon_{P,Q}^{\Delta,\Gamma} &= \Upsilon_{P,Q}^{\Delta,\Gamma}.
\end{aligned}
\tag{121}
$$

The signs above disappear when $|\Delta|$ and $|\Gamma|$ are even numbers, and thus $\Upsilon_{P,Q}^{\Delta,\Gamma}$ can indeed be extended to a function meromorphic on $\mathcal{R}^{\Delta}\times\mathcal{R}^{\Gamma}$ in that case.

Additionally, extending $\mathrm{e}^{2K^{\Delta,\Gamma}}$ to the collection $\overline{\mathcal{R}}^{2}$ of all $\mathcal{R}^{\Delta}\times\mathcal{R}^{\Gamma}$ by $\overline{\mathrm{e}^{2K}}([\nu,(P,\Delta)],[\mu,(Q,\Gamma)]) = \mathrm{e}^{2K^{\Delta,\Gamma}}([\nu,P],[\mu,Q])$, see section 3.8.4, the function $\overline{\mathrm{e}^{2K}}$ verifies from (120) the symmetries $\overline{\mathrm{e}^{2K}}\circ(\overline{\mathcal{T}}\otimes 1) = \overline{\mathrm{e}^{2K}}\circ(1\otimes\overline{\mathcal{T}}) = \overline{\mathrm{e}^{2K}}$ with $\overline{\mathcal{T}}$ defined in (44).

# 5 Relation with known formulas for KPZ

In this section, we show that several known results about KPZ fluctuations in finite volume with periodic boundary conditions are equivalent to the results given in section 2 in terms of the Riemann surfaces introduced in section 3.8 and meromorphic functions on them from section 4. For flat initial condition considered in section 2.2, only the Riemann surface $\check{\mathcal{R}}$ is needed. For sharp wedge and stationary initial conditions, discussed in sections 2.3, 2.4 and 2.5, a summation over the Riemann surfaces $\mathcal{R}^{\Delta}$, $\Delta\sqsubset\mathbb{Z}+1/2$ is needed.

## 5.1 Flat initial condition

In this section, we show that the expression (4) for the probability of the KPZ height with flat initial condition is equivalent to exact results from [39, 40].

### 5.1.1 Relation with the generating function of the height

The expression (4) follows directly from that of the generating function $\langle\mathrm{e}^{sh(x,t)}\rangle_{\mathrm{flat}}$, $s > 0$ obtained in [39], equation (8) [21], based on earlier works [44, 45, 53, 54, 87–90] on the Bethe ansatz solution of TASEP with periodic boundaries. One has

$$
\langle\mathrm{e}^{sh(x,t)}\rangle_{\mathrm{flat}} = s\sum_{\substack{P\sqsubset\mathbb{Z}+1/2 \\ |P|_{+}=|P|_{-}}}\frac{V_{P}^{2}}{4^{|P|}}\frac{\mathrm{e}^{t\chi_{P}(\nu_{P}(s))}\,\mathrm{e}^{\frac{1}{2}\int_{-\infty}^{\nu_{P}(s)}\mathrm{d}\nu\,\chi_{P}''(\nu)^{2}}}{\chi_{P}''(\nu_{P}(s))\,\mathrm{e}^{\nu_{P}(s)/4}(1+\mathrm{e}^{-\nu_{P}(s)})^{1/4}},
\tag{122}
$$

---

[20]We write $A_{n}^{i}$ instead of $A_{n}^{i_{l|r}^{\pm}}$ and $D_{n}$ instead of $D_{n}^{l|r,\pm}$ when analytic continuations depends neither on the signs of the real parts of the variables nor on the sign of the real part of their difference.

[21]The function $\kappa_{a}$ was called $\omega_{a}$ in [39].

where $\nu_P(s)$ is the solution of the equation $\chi_P'(\nu_P(s)) = s$, conjectured to be unique and to verify $\text{Re}\,\nu_P(s) > 0$ when $s > 0$. The probability density $p_{\text{flat}}(u) = -\partial_u \mathbb{P}_{\text{flat}}(h(x,t) > u)$ is related by Fourier transform to the generating function as $p_{\text{flat}}(u) = \int_{-\infty}^{\infty} \frac{ds}{2\pi} e^{-isu} \langle e^{ish(x,t)} \rangle_{\text{flat}}$. Making the change of variables $s = -i\chi_P'(\nu)$ and integrating with respect to $u$ then gives for $c > 0$

$$\mathbb{P}_{\text{flat}}(h(x,t) > u) = \sum_{\substack{P \sqsubset \mathbb{Z}+1/2 \\ |P|_+ = |P|_-}} \frac{V_P^2}{4^{|P|}} \int_{c-i\infty}^{c+i\infty} \frac{d\nu}{2i\pi} \frac{e^{t\chi_P(\nu) - u\chi_P'(\nu) + \frac{1}{2}\int_{-\infty}^{\nu} d\nu\, \chi_P''(\nu)^2}}{e^{\nu/4}(1 + e^{-\nu})^{1/4}} . \tag{123}$$

At this stage, there are two differences between (123) and (4): the constraint $|P|_+ = |P|_-$ that $P$ has as many positive and negative elements, and the integration range $(c - i\infty, c + i\infty)$ instead of $(c - i\pi, c + i\pi)$. These differences correspond simply to distinct ways to label the sheets of $\check{\mathcal{R}}$, or equivalently to the choice of a fundamental domain in $\mathcal{R}$ for the group action $\check{\mathfrak{g}}$ when writing $\check{\mathcal{R}} = \mathcal{R}/\check{\mathfrak{g}}$, compare the two representations of $\mathcal{R}$ in figure 13 on the right side $\text{Re}\,\nu > 0$ of the branch cuts. Using (69), the expressions (123) and (4) are equivalent explicit representations of (1) when $c > 0$ since $|P|_+ = |P|_-$ implies that $|P|$ is even.

### 5.1.2 Relation with Fredholm determinants

The probability distribution $\mathbb{P}_{\text{flat}}(h(x,t) > u)$ was also expressed in [39] as the integral over $\nu$, $\text{Re}\,\nu > 0$ of a Fredholm determinant, using the Cauchy determinant identity (21), see section 2.6.3. Another Fredholm determinant expression for $\mathbb{P}_{\text{flat}}(h(x,t) > u)$ was proved by Baik and Liu in [40] using the propagator approach [91, 92], but with a slightly different kernel and an integral over $\nu$, $\text{Re}\,\nu < 0$ instead. Although it had been checked numerically that both expressions do agree, a proper derivation was missing. We show below that the expression in [40] agrees with (4) when $c < 0$. The analyticity on the cylinder, consequence of the trace in (1), justifies that (4) gives the same result for $c < 0$ and $c > 0$, and shows that the expressions in [39] and [40] for flat initial condition are indeed equivalent.

We start with equation (4.2) of [40]. In our notations, Baik and Liu prove that the height function $h(x,t)$ for the totally asymmetric simple exclusion process with flat initial condition, appropriately rescaled according to KPZ universality, has the cumulative distribution function $\mathbb{P}_{\text{flat}}(h(x,t) > u) = F_1(-u;t)$, with

$$F_1(-u;t) = \oint_{|z|<1} \frac{dz}{2i\pi z} e^{-uA_1(z) + tA_2(z) + A_3(z) + B(z)} \det(1 - \mathcal{K}_z^{(1)}) . \tag{124}$$

The contour of integration encircles 0 once in the anti-clockwise direction. Writing $z = -e^{\nu}$, $\text{Re}\,\nu < 0$, one has in terms of the functions of section 4 the identifications $A_1(-e^{\nu}) = \chi_{\emptyset}'(\nu)$, $A_2(-e^{\nu}) = \chi_{\emptyset}(\nu)$, $A_3(-e^{\nu}) = I_0(\nu)$ and $B(-e^{\nu}) = J_{\emptyset}(\nu)$. After some harmless changes of notations using the fact that any $\xi \in S_{z,\text{left}}$ in [40] is of the form $-\kappa_a(\nu)$ for some $a \in \mathbb{Z}+1/2$, the discrete operator $\mathcal{K}_z^{(1)}$ has for kernel

$$\mathcal{K}_z^{(1)}(a,b) = \frac{\exp(\frac{2t}{3}\kappa_a(\nu)^3 + 2u\kappa_a(\nu) + 2\int_{-\infty}^{\nu} d\nu\, \frac{\chi_{\emptyset}''(\nu)}{\kappa_a(\nu)})}{\kappa_a(\nu)(\kappa_a(\nu) + \kappa_b(\nu))} , \tag{125}$$

with $a, b \in \mathbb{Z}+1/2$ and a path of integration contained in $\mathbb{D}$. The rest of the section is essentially a more detailed version of the derivation of equation (22), run backwards. Expanding the Fredholm determinant in (124) as

$$\det(1 - \mathcal{K}_z^{(1)}) = \sum_{P \sqsubset \mathbb{Z}+1/2} (-1)^{|P|} \det(\mathcal{K}_z^{(1)}(a,b))_{a,b \in P} , \tag{126}$$

using the Cauchy determinant identity (21) and making the change of variable $z = -e^\nu$, one finds for any real number $c < 0$

$$F_1(-u;t) = \int_{c-i\pi}^{c+i\pi} \frac{d\nu}{2i\pi} e^{t\chi_\emptyset(\nu) - u\chi_\emptyset'(\nu) + I_0(\nu) + J_\emptyset(\nu)} \sum_{P \sqsubset \mathbb{Z}+1/2} (-1)^{|P|} \tag{127}$$

$$\times \left( \prod_{a \in P} \frac{\exp(\frac{2t}{3}\kappa_a(\nu)^3 + 2u\kappa_a(\nu) + 2\int_{-\infty}^\nu d\nu \frac{\chi_\emptyset''(\nu)}{\kappa_a(\nu)})}{\kappa_a(\nu)} \right) \frac{\prod_{\substack{a,b \in P \\ a>b}}(\kappa_a(\nu) - \kappa_b(\nu))^2}{\prod_{a,b \in P}(\kappa_a(\nu) + \kappa_b(\nu))} .$$

In terms of the functions $\chi_P$, we obtain from (64), (51)

$$F_1(-u;t) = \int_{c-i\pi}^{c+i\pi} \frac{d\nu}{2i\pi} \sum_{P \sqsubset \mathbb{Z}+1/2} (-1)^{|P|} e^{t\chi_P(\nu) - u\chi_P'(\nu) + I_0(\nu) + J_\emptyset(\nu) + \int_{-\infty}^\nu d\nu\, \chi_\emptyset''(\nu)(\chi_P''(\nu) - \chi_\emptyset''(\nu))}$$

$$\times \left( \prod_{a \in P} \frac{1}{\kappa_a(\nu)} \right) \frac{\prod_{\substack{a,b \in P \\ a>b}}(\kappa_a(\nu) - \kappa_b(\nu))^2}{\prod_{a,b \in P}(\kappa_a(\nu) + \kappa_b(\nu))} . \tag{128}$$

In terms of the regularized integral $\fint_{-\infty}^\nu = \lim_{\Lambda \to \infty}(\dots)\log\Lambda + \int_{-\infty}^\nu$ subtracting appropriately logarithmic divergences used in the definition (74) for the functions $J_P$, one has

$$J_\emptyset(\nu) + \int_{-\infty}^\nu d\nu\, \chi_\emptyset''(\nu)(\chi_P''(\nu) - \chi_\emptyset''(\nu)) \tag{129}$$

$$= J_P(\nu) - \frac{1}{2}\fint_{-\infty}^\nu d\nu\, (\chi_P''(\nu) - \chi_\emptyset''(\nu))^2$$

$$= J_P(\nu) - 2 \sum_{a,b \in P} \fint_{-\infty}^\nu \frac{d\nu}{\kappa_a(\nu)\kappa_b(\nu)}$$

$$= J_P(\nu) + \sum_{a,b \in P} \log\left(\frac{(\kappa_a(\nu) + \kappa_b(\nu))^2}{8}\right),$$

where the first equality comes from (74), the second from (161) and the third from (184) using the fact that $2\log(\kappa_a(\nu) + \kappa_b(\nu)) = \log((\kappa_a(\nu) + \kappa_b(\nu))^2)$ when Re $\nu < 0$. We obtain

$$F_1(-u;t) = \int_{c-i\pi}^{c+i\pi} \frac{d\nu}{2i\pi} \sum_{P \sqsubset \mathbb{Z}+1/2} \frac{(-1)^{|P|}}{4^{|P|}} e^{t\chi_P(\nu) - u\chi_P'(\nu) + I_0(\nu) + J_P(\nu)}$$

$$\times \prod_{\substack{a,b \in P \\ a>b}} \left(\frac{\kappa_a(\nu)^2 - \kappa_b(\nu)^2}{8}\right)^2 . \tag{130}$$

From the definition (50) of $\kappa_a(\nu)$, one has $(\kappa_a(\nu)^2 - \kappa_b(\nu)^2)/8 = 2i\pi a/4 - 2i\pi b/4$, which finally gives (4) with $c < 0$.

## 5.2 Sharp wedge initial condition

In this section, we show that the expression (10) for the probability of the KPZ height with sharp wedge initial condition is equivalent to exact results from [39, 40].

### 5.2.1 Relation with the generating function of the height from [39]

The expression (6) for sharp wedge initial condition is a consequence of the generating func-

tion obtained in [39], equation (9), and given for $s > 0$ by

$$\langle e^{sh(x,t)}\rangle_{\mathrm{sw}} = s \sum_{\substack{P,H \sqsubset \mathbb{Z}+1/2 \\ |P|_{\pm} = |H|_{\mp}}} \frac{i^{|P|+|H|} V_P^2 V_H^2}{4^{|P|+|H|}} e^{2i\pi x(\sum_{a \in P} a - \sum_{a \in H} a)} \tag{131}$$

$$\times \frac{e^{t\chi_{P,H}(\nu_{P,H}(s))} \lim_{\Lambda \to \infty} \Lambda^{-|P|^2-|H|^2} e^{\int_{-\Lambda}^{\nu_{P,H}(s)} d\nu \, \chi_{P,H}''(\nu)^2}}{\chi_{P,H}''(\nu_{P,H}(s))} \, ,$$

with Vandermonde determinants $V_P$, $V_H$ defined in (5),

$$\chi_{P,H}(\nu) = \frac{\chi_P(\nu) + \chi_H(\nu)}{2} \tag{132}$$

and $\nu_{P,H}(s)$ the solution of $\chi_{P,H}'(\nu_{P,H}(s)) = s$, conjectured to be unique when $\mathrm{Re}\, s > 0$. The restrictions $|P|_+ = |H|_-$, $|P|_- = |H|_+$ on the number of positive and negative elements of $P$ and $H$ imply in particular that we are summing only over sets $P$ and $H$ with $|P| = |H|$.

As in section 5.1.1, the cumulative distribution function of the height can be derived from the generating function by Fourier transform, and we obtain in terms of $\chi_P^{\Delta}$, $J_P^{\Delta}$ defined in (87), (74)

$$\mathbb{P}_{\mathrm{sw}}(h(x,t) > u) = \sum_{\substack{P,H \sqsubset \mathbb{Z}+1/2 \\ |P|_{\pm} = |H|_{\mp}}} \frac{i^{|P|+|H|} V_P^2 V_H^2}{4^{|P|+|H|}} e^{2i\pi x(\sum_{a \in P} a - \sum_{a \in H} a)} \tag{133}$$

$$\times \int_{c-i\infty}^{c+i\infty} \frac{d\nu}{2i\pi} e^{t\chi_P^{\Delta}(\nu) - u\chi_P'^{\Delta}(\nu) + 2J_P^{\Delta}(\nu)} \, ,$$

with $c > 0$ and $\Delta = P \ominus H$.

One has $|P| + |H| = 2|P \setminus \Delta| + |\Delta|$, $\sum_{a \in P} a - \sum_{a \in H} a = \sum_{a \in A} a - \sum_{a \in \Delta \setminus A} a$ and

$$V_P^2 V_H^2 = V_A^2 V_{\Delta \setminus A}^2 \prod_{a \in P \setminus \Delta} \prod_{\substack{b \in P \cup \Delta \\ b \neq a}} \Big(\frac{2i\pi a}{4} - \frac{2i\pi b}{4}\Big)^2 \, , \tag{134}$$

where $\Delta = P \ominus H$ and $A = P \cap \Delta$. These identities allow to rewrite (133) as

$$\mathbb{P}_{\mathrm{sw}}(h(x,t) > u) = \sum_{\Delta \sqsubset \mathbb{Z}+1/2} \sum_{A \subset \Delta} \sum_{\substack{Q \sqsubset \mathbb{Z}+1/2 \\ Q \cap \Delta = \emptyset}} 1_{\{|P|_{\pm} = |P \ominus \Delta|_{\mp}\}} (i/4)^{2|P \setminus \Delta| + |\Delta|} V_A^2 V_{\Delta \setminus A}^2$$

$$\times \Big(\prod_{a \in P \setminus \Delta} \prod_{\substack{b \in P \cup \Delta \\ b \neq a}} \Big(\frac{2i\pi a}{4} - \frac{2i\pi b}{4}\Big)^2\Big) e^{2i\pi x(\sum_{a \in A} a - \sum_{a \in \Delta \setminus A} a)}$$

$$\times \int_{c-i\infty}^{c+i\infty} \frac{d\nu}{2i\pi} e^{t\chi_P^{\Delta}(\nu) - u\chi_P'^{\Delta}(\nu) + 2J_P^{\Delta}(\nu)} \, , \tag{135}$$

where the sum over $P = Q \cup A$ and $H = P \cup (\Delta \setminus A)$ has been replaced by a sum over $\Delta, A, Q$. Since $A \subset \Delta$ and the function $\chi_P^{\Delta}$ verifies $\chi_P^{\Delta} = \chi_{P \setminus \Delta}^{\Delta}$, all $\chi_P^{\Delta}$ in the integral can be replaced by $\chi_Q^{\Delta}$. Additionally, $P \setminus \Delta = Q$ and $P \cup \Delta = Q \cup \Delta$ shows that further factors are independent of $A$. Finally $|P|_{\pm} = |Q|_{\pm} + |A|_{\pm}$, $|P \ominus \Delta|_{\mp} = |Q|_{\mp} + |\Delta \setminus A|_{\mp} = |Q|_{\mp} + |\Delta|_{\mp} - |A|_{\mp}$, and the constraints $|P|_{\pm} = |P \ominus \Delta|_{\mp}$ can be replaced by $|A| = |\Delta \setminus A|$, implying that $|\Delta|$ is necessarily even, and $|Q|_+ - |Q \ominus \Delta|_- = -|\Delta|/2$, equivalent to $\lambda_+(Q, \Delta) = -|\Delta|/2$ with $\lambda_+$ defined above (43).

In terms of $\Xi_x^\Delta$ defined in (7) and of the functions $\chi^\Delta$, $\chi'^\Delta$, $e^{2J^\Delta}$ on the Riemann surface $\mathcal{R}^\Delta$ defined in (86), (91), (96), we obtain

$$\mathbb{P}_{\mathrm{sw}}(h(x,t) > u) = \sum_{\Delta \sqsubset \mathbb{Z}+1/2} \Xi_x^\Delta \sum_{\substack{Q \sqsubset \mathbb{Z}+1/2 \\ Q \cap \Delta = \emptyset}} 1_{\{\lambda_+(Q,\Delta)=-|\Delta|/2\}} \tag{136}$$

$$\times \int_{c-i\infty}^{c+i\infty} \frac{d\nu}{2i\pi} (e^{t\chi^\Delta - u\chi'^\Delta + 2J^\Delta})([\nu,Q]).$$

We split $\int_{c-i\infty}^{c+i\infty}$ into $\sum_{m=-\infty}^{\infty} \int_{c-2i\pi(m-1/2)}^{c+2i\pi(m+1/2)}$ and shift $\nu$ by $-2i\pi m$. We then use the symmetry by $\overline{\mathcal{T}}$ defined in (44) of the extension to $\overline{\mathcal{R}}$ of $\chi^\Delta$ and $e^{2J^\Delta}$, i.e. we replace $[\nu - 2i\pi m, (Q, \Delta)]$ by $[\nu, ((Q+m) \ominus (B_m \setminus (\Delta+m)), \Delta+m)]$ everywhere since $c > 0$. Making the change of variable $Q \to P \ominus (B_m \setminus (\Delta + m)) - m = (P - m) \ominus (B_{-m} \setminus \Delta)$ followed by $\Delta \to \Delta - m$, the constraint $Q \cap \Delta = \emptyset$ becomes $P \cap \Delta = \emptyset$. Using $\Xi_x^{\Delta-m} = \Xi_x^\Delta$ then leads to

$$\mathbb{P}_{\mathrm{sw}}(h(x,t) > u) = \sum_{\Delta \sqsubset \mathbb{Z}+1/2} \Xi_x^\Delta \sum_{\substack{P \sqsubset \mathbb{Z}+1/2 \\ P \cap \Delta = \emptyset}} \sum_{m=-\infty}^{\infty} 1_{\{\lambda_+((P-m)\ominus(B_{-m}\setminus(\Delta-m)),\Delta-m)=-|\Delta|/2\}} \tag{137}$$

$$\times \int_{c-i\pi}^{c+i\pi} \frac{d\nu}{2i\pi} (e^{t\chi^\Delta - u\chi'^\Delta + 2J^\Delta})([\nu,P]).$$

Using (43), the condition $\lambda_+((P-m) \ominus (B_{-m} \setminus (\Delta-m)), \Delta-m) = -|\Delta|/2$ is equivalent to $\lambda_+(P,\Delta) = m - |\Delta|/2$. We observe that there exists a unique $m \in \mathbb{Z}$ such that the constraint is verified when $|\Delta|$ is even, and that no $m \in \mathbb{Z}$ satisfies the constraint when $|\Delta|$ is odd. Since $\Xi_x^\Delta = 0$ when $|\Delta|$ is odd, this leads to (10), using the definitions (86), (96) of $\chi^\Delta$ and $e^{2J^\Delta}$.

The slightly tedious derivation of (10) from (135) in this section can be understood more directly, but at the price of heavier formalism, by considering a collection $\overline{\overline{\mathcal{R}}}$ of copies of $\mathcal{R}$ and a covering map from $\overline{\overline{\mathcal{R}}}$ to $\overline{\mathcal{R}}$ projecting each copy of $\mathcal{R}$ in $\overline{\overline{\mathcal{R}}}$ to a distinct $\mathcal{R}^\Delta$ in $\overline{\mathcal{R}}$. The functions appearing in (135) can then be interpreted as functions on the components $\mathcal{R}$ of $\overline{\overline{\mathcal{R}}}$, equal at $[\nu,P] \in \mathcal{R}$ to the product of a constant depending only on $A = P \cap \Delta$, which is eventually gathered into $\Xi_x^\Delta$, and a function of $[\nu, P \setminus \Delta] \in \mathcal{R}^\Delta$.

### 5.2.2  Relation with the expression from Baik and Liu [40]

In this section, we show that our result (6) for the cumulative distribution function of KPZ fluctuations with sharp wedge initial condition agrees with the alternative formula by Baik and Liu [40], with an integration on the left side of the branch cuts.

We start with equation (4.10) of [40]. In our notations, Baik and Liu prove that the height function $h(x,t)$ for the totally asymmetric simple exclusion process with domain wall initial condition, appropriately rescaled according to KPZ universality, has the cumulative distribution function $\mathbb{P}_{\mathrm{sw}}(h(x,t) > u) = F_2(-u; t, x)$, with

$$F_2(-u; t, x) = \oint_{|z|<1} \frac{dz}{2i\pi z} e^{-uA_1(z) + tA_2(z) + 2B(z)} \det(1 - \mathcal{K}_z^{(2)}). \tag{138}$$

The contour of integration encircles 0 once in the anti-clockwise direction. Writing $z = -e^\nu$, $\mathrm{Re}\,\nu < 0$, one has in terms of the functions of section 4 the identifications $A_1(-e^\nu) = \chi'_\emptyset(\nu)$, $A_2(-e^\nu) = \chi_\emptyset(\nu)$ and $B(-e^\nu) = J_\emptyset(\nu)$. After some harmless changes of notations using the fact

that any $\xi \in S_{z,\text{left}}$ in [40] is of the form $-\kappa_a(\nu)$ for some $a \in \mathbb{Z} + 1/2$, the discrete operator $\mathcal{K}_z^{(2)}$ has for kernel

$$\mathcal{K}_z^{(2)}(a,b) = \frac{\exp\left(\frac{t}{3}\kappa_a(\nu)^3 + u\kappa_a(\nu) + 2i\pi a x + 2\int_{-\infty}^{\nu} d\nu\, \frac{\chi_\emptyset''(\nu)}{\kappa_a(\nu)}\right)}{\kappa_a(\nu)} \tag{139}$$

$$\times \sum_{c \in \mathbb{Z}+1/2} \frac{\exp\left(\frac{t}{3}\kappa_c(\nu)^3 + u\kappa_c(\nu) - 2i\pi c x + 2\int_{-\infty}^{\nu} d\nu\, \frac{\chi_\emptyset''(\nu)}{\kappa_c(\nu)}\right)}{\kappa_c(\nu)(\kappa_a(\nu) + \kappa_c(\nu))(\kappa_b(\nu) + \kappa_c(\nu))}\,,$$

with $a, b \in \mathbb{Z} + 1/2$. The rest of the section is essentially a more detailed version of the derivation of (25), run backwards. Expanding the Fredholm determinant in (124) as

$$\det(1 - \mathcal{K}_z^{(2)}) = \sum_{P \sqsubset \mathbb{Z}+1/2} (-1)^{|P|} \det(\mathcal{K}_z^{(2)}(a,b))_{a,b \in P}\,, \tag{140}$$

using

$$\det\left(\sum_{c \in \mathbb{Z}+1/2} \mathcal{K}_{a,b,c}\right)_{a,b \in P} = \left(\prod_{a \in P} \sum_{c_a \in \mathbb{Z}+1/2}\right) \det\left(\mathcal{K}_{a,b,c_a}\right)_{a,b \in P}\,, \tag{141}$$

the Cauchy determinant identity

$$\det\left(\frac{1}{\kappa_{c_a} + \kappa_b}\right)_{a,b \in P} = \frac{\left(\prod_{\substack{a,b \in P \\ a>b}}(\kappa_a - \kappa_b)\right)\left(\prod_{\substack{a,b \in P \\ a>b}}(\kappa_{c_a} - \kappa_{c_b})\right)}{\prod_{a,b \in P}(\kappa_{c_a} + \kappa_b)}\,, \tag{142}$$

and making the change of variable $z = -e^\nu$, one finds for any real number $c < 0$

$$F_2(-u; t, x) = \int_{c-i\pi}^{c+i\pi} \frac{d\nu}{2i\pi}\, e^{t\chi_\emptyset(\nu) - u\chi_\emptyset'(\nu) + 2J_\emptyset(\nu)} \sum_{P \sqsubset \mathbb{Z}+1/2} \left(\prod_{a \in P} \sum_{c_a \in \mathbb{Z}+1/2}\right)(-1)^{|P|}$$

$$\times \left(\prod_{a \in P} \frac{\exp\left(\frac{t}{3}\kappa_a(\nu)^3 + u\kappa_a(\nu) + 2i\pi a x + 2\int_{-\infty}^{\nu} d\nu\, \frac{\chi_\emptyset''(\nu)}{\kappa_a(\nu)}\right)}{\kappa_a(\nu)}\right) \tag{143}$$

$$\times \left(\prod_{a \in P} \frac{\exp\left(\frac{t}{3}\kappa_{c_a}(\nu)^3 + u\kappa_{c_a}(\nu) - 2i\pi c_a x + 2\int_{-\infty}^{\nu} d\nu\, \frac{\chi_\emptyset''(\nu)}{\kappa_{c_a}(\nu)}\right)}{\kappa_{c_a}(\nu)}\right)$$

$$\times \frac{\left(\prod_{\substack{a,b \in P \\ a>b}}(\kappa_a(\nu) - \kappa_b(\nu))\right)\left(\prod_{\substack{a,b \in P \\ a>b}}(\kappa_{c_a}(\nu) - \kappa_{c_b}(\nu))\right)}{\left(\prod_{a \in P}(\kappa_a(\nu) + \kappa_{c_a}(\nu))\right)\left(\prod_{a,b \in P}(\kappa_{c_a}(\nu) + \kappa_b(\nu))\right)}\,.$$

Because of the factor $\prod_{\substack{a,b \in P \\ a>b}}(\kappa_{c_a}(\nu) - \kappa_{c_b}(\nu))$, only the tuples $c_a$, $a \in P$ with distinct elements contribute, and one can replace these tuples by finite sets $H \sqsubset \mathbb{Z} + 1/2$ with $|H| = |P|$ up to permutations. Using the identity

$$\left(\prod_{a \in P} \sum_{c_a \in \mathbb{Z}+1/2}\right) 1_{\{\{c_a, a \in P\} = H\}} \frac{\left(\prod_{\substack{a,b \in P \\ a>b}}(\kappa_{c_a} - \kappa_{c_b})\right)}{\left(\prod_{a \in P}(\kappa_a + \kappa_{c_a})\right)\left(\prod_{a,b \in P}(\kappa_{c_a} + \kappa_b)\right)}$$

$$= \frac{\left(\prod_{\substack{a,b \in P \\ a>b}}(\kappa_a - \kappa_b)\right)\left(\prod_{\substack{a,b \in H \\ a>b}}(\kappa_a - \kappa_b)^2\right)}{\prod_{a \in P}\prod_{b \in H}(\kappa_a + \kappa_b)^2} \tag{144}$$

for $P, H \sqsubset \mathbb{Z} + 1/2$, $|P| = |H|$, (132), (64) and (161), this leads to

$$F_2(-u; t, x) = \int_{c-i\pi}^{c+i\pi} \frac{dv}{2i\pi} \sum_{\substack{P,H \sqsubset \mathbb{Z}+1/2 \\ |P|=|H|}} (-1)^{|P|} e^{t\chi_{P,H}(v) - u\chi'_{P,H}(v)} e^{2i\pi x \left( \sum_{a\in P} a - \sum_{a\in H} a \right)}$$

$$\times e^{2J_\emptyset(v) + 2\int_{-\infty}^v dv \, \chi''_\emptyset(v)(\chi''_{P,H}(v) - \chi''_\emptyset(v))} \left( \prod_{a\in P} \frac{1}{\kappa_a(v)} \right) \left( \prod_{a\in H} \frac{1}{\kappa_a(v)} \right)$$

$$\times \frac{\left( \prod_{\substack{a,b\in P \\ a>b}} (\kappa_a(v) - \kappa_b(v)) \right)^2 \left( \prod_{\substack{a,b\in H \\ a>b}} (\kappa_a(v) - \kappa_b(v)) \right)^2}{\left( \prod_{a\in P} \prod_{b\in H} (\kappa_a(v) + \kappa_b(v)) \right)^2} \,. \quad (145)$$

In terms of the regularized integral $\fint_{-\infty}^v = \lim_{\Lambda\to\infty} (\dots) \log \Lambda + \int_{-\infty}^v$ subtracting appropriately logarithmic divergences used in the definition (74) for the functions $J_P$, one has

$$J_\emptyset(v) + \int_{-\infty}^v dv \, \chi''_\emptyset(v)(\chi''_{P,H}(v) - \chi''_\emptyset(v)) \quad (146)$$

$$= \frac{1}{2} \int_{-\infty}^v dv \, \chi''_{P,H}(v)^2 - \frac{1}{2} \int_{-\infty}^v dv \, (\chi''_{P,H}(v) - \chi''_\emptyset(v))^2$$

$$= \frac{1}{2} \int_{-\infty}^v dv \, \chi''_{P,H}(v)^2 - \frac{1}{2} \sum_{a,b\in P} \fint_{-\infty}^v \frac{dv}{\kappa_a(v)\kappa_b(v)} - \frac{1}{2} \sum_{a,b\in H} \fint_{-\infty}^v \frac{dv}{\kappa_a(v)\kappa_b(v)}$$

$$- \sum_{a\in P} \sum_{b\in H} \fint_{-\infty}^v \frac{dv}{\kappa_a(v)\kappa_b(v)}$$

$$= \frac{1}{2} \int_{-\infty}^v dv \, \chi''_{P,H}(v)^2 + \frac{1}{2} \sum_{a,b\in P} \log \left( \frac{\kappa_a(v) + \kappa_b(v)}{\sqrt{8}} \right)$$

$$+ \frac{1}{2} \sum_{a,b\in H} \log \left( \frac{\kappa_a(v) + \kappa_b(v)}{\sqrt{8}} \right) + \sum_{a\in P} \sum_{b\in H} \log \left( \frac{\kappa_a(v) + \kappa_b(v)}{\sqrt{8}} \right),$$

where the first equality comes from (74), the second from (132), (161) and the third from (184). We obtain

$$F_2(-u; t, x) = \int_{c-i\pi}^{c+i\pi} \frac{dv}{2i\pi} \sum_{\substack{P,H \sqsubset \mathbb{Z}+1/2 \\ |P|=|H|}} \frac{i^{|P|+|H|}}{4^{|P|+|H|}} e^{t\chi_{P,H}(v) - u\chi'_{P,H}(v) + \int_{-\infty}^v dv \, \chi''_{P,H}(v)^2} \quad (147)$$

$$\times e^{2i\pi x \left( \sum_{a\in P} a - \sum_{a\in H} a \right)} \left( \prod_{\substack{a,b\in P \\ a>b}} \frac{\kappa_a(v)^2 - \kappa_b(v)^2}{8} \right)^2 \left( \prod_{\substack{a,b\in H \\ a>b}} \frac{\kappa_a(v)^2 - \kappa_b(v)^2}{8} \right)^2.$$

From the definition (50) of $\kappa_a(v)$, one has $(\kappa_a(v)^2 - \kappa_b(v)^2)/8 = 2i\pi a/4 - 2i\pi b/4$, which gives

$$F_2(-u; t, x) = \int_{c-i\pi}^{c+i\pi} \frac{dv}{2i\pi} \sum_{\substack{P,H \sqsubset \mathbb{Z}+1/2 \\ |P|=|H|}} \frac{i^{|P|+|H|} V_P^2 V_H^2}{4^{|P|+|H|}} e^{2i\pi x \left( \sum_{a\in P} a - \sum_{a\in H} a \right)}$$

$$\times e^{t\chi_{P,H}(v) - u\chi'_{P,H}(v) + \int_{-\infty}^v dv \, \chi''_{P,H}(v)^2}, \quad (148)$$

with $V_P$ the Vandermonde determinant defined in (5). Introducing $\Delta = P \ominus H$ and the functions

$\chi_P^\Delta$ and $J_P^\Delta$ from (87), (92), we finally obtain

$$\mathbb{P}_{\mathrm{sw}}(h(x,t) > u) = \sum_{\substack{P,H \sqsubset \mathbb{Z}+1/2 \\ |P|=|H|}} \frac{\mathrm{i}^{|P|+|H|} V_P^2 V_H^2}{4^{|P|+|H|}} \mathrm{e}^{2\mathrm{i}\pi x \left( \sum_{a \in P} a - \sum_{a \in H} a \right)} \tag{149}$$

$$\times \int_{c-\mathrm{i}\pi}^{c+\mathrm{i}\pi} \frac{\mathrm{d}\nu}{2\mathrm{i}\pi} \mathrm{e}^{t\chi_P^\Delta(\nu) - u\chi_P'^\Delta(\nu) + 2J_P^\Delta(\nu)},$$

with $\Delta = P \ominus H$ and $V_P$, $\chi_P^\Delta$, $J_P^\Delta$ defined in (5), (87), (92). The sign of $c$, the integration range and the constraint on the sets $P$ and $H$ differ from (133). This corresponds simply to another choice of fundamental domain for the Riemann surfaces $\mathcal{R}^\Delta$. Indeed, writing $P = Q \cup A$ with $A \subset \Delta$, $Q \cap \Delta = \emptyset$ and using (134) as in the previous section leads to

$$\mathbb{P}_{\mathrm{sw}}(h(x,t) > u) = \sum_{\Delta \sqsubset \mathbb{Z}+1/2} \sum_{A \subset \Delta} \sum_{\substack{Q \sqsubset \mathbb{Z}+1/2 \\ Q \cap \Delta = \emptyset}} \mathbb{1}_{\{|P|=|P \ominus \Delta|\}} (\mathrm{i}/4)^{2|P \setminus \Delta| + |\Delta|} V_A^2 V_{\Delta \setminus A}^2$$

$$\times \left( \prod_{a \in P \setminus \Delta} \prod_{\substack{b \in P \cup \Delta \\ b \neq a}} \left( \frac{2\mathrm{i}\pi a}{4} - \frac{2\mathrm{i}\pi b}{4} \right)^2 \right) \mathrm{e}^{2\mathrm{i}\pi x \left( \sum_{a \in A} a - \sum_{a \in \Delta \setminus A} a \right)}$$

$$\times \int_{c-\mathrm{i}\pi}^{c+\mathrm{i}\pi} \frac{\mathrm{d}\nu}{2\mathrm{i}\pi} \mathrm{e}^{t\chi_P^\Delta(\nu) - u\chi_P'^\Delta(\nu) + 2J_P^\Delta(\nu)}, \tag{150}$$

which parallels (135). Since $|P| = |P \ominus \Delta|$ is equivalent to $|A| = |\Delta \setminus A|$, the same reasoning as from (135) to (136) finally gives

$$\mathbb{P}_{\mathrm{sw}}(h(x,t) > u) = \sum_{\Delta \sqsubset \mathbb{Z}+1/2} \Xi_x^\Delta \sum_{\substack{Q \sqsubset \mathbb{Z}+1/2 \\ Q \cap \Delta = \emptyset}} \int_{c-\mathrm{i}\pi}^{c+\mathrm{i}\pi} \frac{\mathrm{d}\nu}{2\mathrm{i}\pi} (\mathrm{e}^{t\chi^\Delta - u\chi'^\Delta + 2J^\Delta})([\nu, Q]), \tag{151}$$

which is precisely (10).

## 5.3 Multiple-time statistics with sharp wedge initial condition

In this section, we derive (13) starting with a result by Baik and Liu [42]. We also discuss the pole structure on the Riemann surfaces $\mathcal{R}^{\Delta_\ell}$ of the final expression.

### 5.3.1 Derivation of (13) from Baik-Liu [42]

The joint distribution of the height at multiple times $0 < t_1 < \ldots < t_m$ and positions $x_j$ was obtained by Baik and Liu in [42], equation (2.15), with the expression (2.21) for $C(\mathbf{z})$, and (2.51), (2.55) for $D(\mathbf{z})$. Under the replacements $\tau_j \to t_j$, $\gamma_j \to x_j$, $x_j \to -u_j$, one has $\mathbb{P}_{\mathrm{sw}}(h(x_1, t_1) > u_1, \ldots, h(x_m, t_m) > u_m) = F(\vec{t}, \vec{x}, \vec{u})$ where

$$F(\vec{t}, \vec{x}, \vec{u})$$
$$= \oint \frac{\mathrm{d}z_1}{2\mathrm{i}\pi z_1} \cdots \frac{\mathrm{d}z_m}{2\mathrm{i}\pi z_m} \left( \prod_{\ell=1}^{m} \left( \frac{z_\ell}{z_\ell - z_{\ell+1}} \frac{\mathrm{e}^{-u_\ell A_1(z_\ell) + t_\ell A_2(z_\ell)}}{\mathrm{e}^{-u_\ell A_1(z_{\ell+1}) + t_\ell A_2(z_{\ell+1})}} \mathrm{e}^{2B(z_\ell, z_\ell) - 2B(z_{\ell+1}, z_\ell)} \right) \right)$$

$$\times \sum_{n_1, \ldots, n_m = 0}^{\infty} \left( \prod_{\ell=1}^{m} \frac{1}{(n_\ell!)^2} \right) \left( \prod_{\ell=2}^{m} \left( \left(1 - \frac{z_{\ell-1}}{z_\ell}\right)^{n_\ell} \left(1 - \frac{z_\ell}{z_{\ell-1}}\right)^{n_{\ell-1}} \right) \right) \tag{152}$$

$$\times \left( \prod_{\ell=1}^{m} \sum_{\mathcal{P}_\ell, \mathcal{H}_\ell \in (\mathbb{Z}+1/2)^{n_\ell}} \right) \left( \prod_{\ell=1}^{m} \left( \frac{\Delta(U^{(\ell)})^2 \Delta(V^{(\ell)})^2}{\Delta(U^{(\ell)}; V^{(\ell)})^2} \hat{f}_\ell(U^{(\ell)}) \hat{f}_\ell(V^{(\ell)}) \right) \right)$$

$$\times \left( \prod_{\ell=2}^{m} \left( \frac{\Delta(U^{(l)}; V^{(\ell-1)}) \Delta(V^{(l)}; U^{(\ell-1)})}{\Delta(U^{(l)}; U^{(\ell-1)}) \Delta(V^{(l)}; V^{(\ell-1)})} \frac{\mathrm{e}^{-h(V^{(\ell)}, z_{\ell-1}) - h(V^{(l-1)}, z_\ell)}}{\mathrm{e}^{h(U^{(\ell)}, z_{\ell-1}) + h(U^{(\ell-1)}, z_\ell)}} \right) \right),$$

with $|z_m| < \ldots < |z_1| < 1$, $\nu_{m+1} = -\infty$, $x_0 = x_{m+1} = t_0 = t_{m+1} = u_0 = u_{m+1} = 0$. In terms of $z_\ell = -e^{\nu_\ell}$, $\mathrm{Re}\,\nu_\ell < 0$, $-\pi < \mathrm{Im}\,\nu_\ell < \pi$, one has $A_1(z_\ell) = \chi'_\emptyset(\nu_\ell)$ and $A_2(z_\ell) = \chi_\emptyset(\nu_\ell)$ with $\chi_\emptyset$ given by (56) and $B(z_{\ell_1}, z_{\ell_2}) = K_{\emptyset,\emptyset}(\nu_{\ell_1}, \nu_{\ell_2})$ with $K_{\emptyset,\emptyset}$ given by (101). The $n_\ell$-uples $U^{(\ell)}$, $V^{(\ell)}$ are defined in terms of the $n_\ell$-uples $\mathcal{P}_\ell$, $\mathcal{H}_\ell$ as $U^{(\ell)} = (\kappa_a(\nu_\ell), a \in \mathcal{P}_\ell)$, $V^{(\ell)} = (-\kappa_a(\nu_\ell), a \in \mathcal{H}_\ell)$. The quantities $\Delta(W)$, $\Delta(W; W')$ for tuples $W = (w_1, \ldots, w_n)$, $W' = (w'_1, \ldots, w'_{n'})$ are defined as $\Delta(W) = \prod_{1 \le i < j \le n}(w_j - w_i)$, $\Delta(W; W') = \prod_{i=1}^{n} \prod_{i'=1}^{n'}(w_i - w'_{i'})$. The remaining factors are given by $h(U^{(\ell_1)}, z_{\ell_2}) = \sum_{a \in \mathcal{P}_{\ell_1}} \int_{-\infty}^{0} du\, \chi''_\emptyset(u + \nu_{\ell_2})/\kappa_a(u + \nu_{\ell_1})$, $h(V^{(\ell_1)}, z_{\ell_2}) = \sum_{a \in \mathcal{H}_{\ell_1}} \int_{-\infty}^{0} du\, \chi''_\emptyset(u + \nu_{\ell_2})/\kappa_a(u + \nu_{\ell_1})$ and

$$\hat{f}_\ell(U^{(\ell)}) = (-1)^{n_\ell}\, e^{2\sum_{a \in P_\ell} \int_{-\infty}^{\nu_\ell} dv\, \frac{\chi''_\emptyset(v)}{\kappa_a(v)}} \prod_{a \in P_\ell} \frac{e^{(t_\ell - t_{\ell-1})\frac{\kappa_a(\nu_\ell)^3}{3} + (x_\ell - x_{\ell-1})\frac{\kappa_a(\nu_\ell)^2}{2} + (u_\ell - u_{\ell-1})\kappa_a(\nu_\ell)}}{\kappa_a(\nu_\ell)}$$

$$\hat{f}_\ell(V^{(\ell)}) = e^{2\sum_{a \in H_\ell} \int_{-\infty}^{\nu_\ell} dv\, \frac{\chi''_\emptyset(v)}{\kappa_a(v)}} \prod_{a \in H_\ell} \frac{e^{(t_\ell - t_{\ell-1})\frac{\kappa_a(\nu_\ell)^3}{3} - (x_\ell - x_{\ell-1})\frac{\kappa_a(\nu_\ell)^2}{2} + (u_\ell - u_{\ell-1})\kappa_a(\nu_\ell)}}{\kappa_a(\nu_\ell)}. \tag{153}$$

Because of the Vandermonde determinants $\Delta(U^{(\ell)})$, $\Delta(V^{(\ell)})$, only tuples $\mathcal{P}_\ell$, $\mathcal{H}_\ell$ with distinct elements contribute to (152). Since the summand is invariant under permutations of the elements of $\mathcal{P}_\ell$, $\mathcal{H}_\ell$, one can sum over subsets $P_\ell$, $H_\ell$ of $\mathbb{Z} + 1/2$ instead, up to a factor $\prod_{\ell=1}^{m}(n_\ell!)^2$ counting the number of permutations. Making the changes of variables $z_\ell = -e^{\nu_\ell}$, one finds after some simplifications

$$F(\vec{t}, \vec{x}, \vec{u}) = \int_{c_1 - i\pi}^{c_1 + i\pi} \frac{d\nu_1}{2i\pi} \cdots \int_{c_m - i\pi}^{c_m + i\pi} \frac{d\nu_m}{2i\pi} \left( \prod_{\ell=1}^{m} \sum_{\substack{P_\ell, H_\ell \sqsubset \mathbb{Z} + 1/2 \\ |P_\ell| = |H_\ell|}} \right) \tag{154}$$

$$\times \left( \prod_{\ell=1}^{m} \left( (i/4)^{2n_\ell} V_{P_\ell}^2 V_{H_\ell}^2\, e^{2i\pi(x_\ell - x_{\ell-1})(\sum_{a \in P_\ell} a - \sum_{a \in H_\ell} a)} \right. \right.$$

$$\left. \left. \times e^{(t_\ell - t_{\ell-1})\chi_{P_\ell}^{\Delta_\ell}(\nu_\ell) - (u_\ell - u_{\ell-1})\chi'^{\Delta_\ell}_{P_\ell}(\nu_\ell) + 2J_{P_\ell}^{\Delta_\ell}(\nu_\ell)} \right) \right)$$

$$\times \prod_{\ell=1}^{m-1} \frac{(1 - e^{\nu_{\ell+1} - \nu_\ell})^{-1+n_\ell}(1 - e^{\nu_\ell - \nu_{\ell+1}})^{n_{\ell+1}}\, e^{-2K_{P_\ell, P_{\ell+1}}^{\Delta_\ell, \Delta_{\ell+1}}(\nu_\ell, \nu_{\ell+1})}}{\left( \prod_{a \in P_\ell} \prod_{b \in P_{\ell+1}} \left( \frac{2i\pi b - \nu_{\ell+1}}{4} - \frac{2i\pi a - \nu_\ell}{4} \right) \right) \left( \prod_{a \in H_\ell} \prod_{b \in H_{\ell+1}} \left( \frac{2i\pi b - \nu_{\ell+1}}{4} - \frac{2i\pi a - \nu_\ell}{4} \right) \right)},$$

with $c_m < \ldots < c_1 < 0$, $n_\ell = |P_\ell| = |H_\ell|$ and $\Delta_\ell = P_\ell \ominus H_\ell$.

Using $2n_\ell = 2|P_\ell \setminus \Delta_\ell| + |\Delta_\ell|$ and (134), the general factor of the first product in (154) rewrites in terms of the functions $\chi^\Delta$, $\chi'^\Delta$, $e^{2J^\Delta}$ on the Riemann surface $\mathcal{R}^\Delta$ defined in (86), (91), (96) as

$$(i/4)^{2n_\ell} V_{P_\ell}^2 V_{H_\ell}^2\, e^{(t_\ell - t_{\ell-1})\chi_{P_\ell}^{\Delta_\ell}(\nu_\ell) - (u_\ell - u_{\ell-1})\chi'^{\Delta_\ell}_{P_\ell}(\nu_\ell) + 2J_{P_\ell}^{\Delta_\ell}(\nu_\ell)}$$

$$= (i/4)^{|\Delta_\ell|} V_{A_\ell}^2 V_{\Delta_\ell \setminus A_\ell}^2 \left( e^{(t_\ell - t_{\ell-1})\chi^{\Delta_\ell} - (u_\ell - u_{\ell-1})\chi'^{\Delta_\ell} + 2J^{\Delta_\ell}} \right)([\,\nu_\ell, P_\ell\,]), \tag{155}$$

with $A_\ell = P_\ell \cap \Delta_\ell$. Furthermore, one has $1 - e^{\nu_{\ell+1} - \nu_\ell} = e^{-4I_0(\nu_{\ell+1} - \nu_\ell + i\pi)}$,

$1 - e^{\nu_\ell - \nu_{\ell+1}} = e^{-4I_0(\nu_\ell - \nu_{\ell+1} - i\pi)}$ with $I_0$ defined in (68), and the identities $n_\ell = |P_\ell \setminus \Delta_\ell| + |\Delta_\ell|/2$,

$$\frac{\left(\prod_{a \in P\setminus\Delta}\prod_{b\in Q\setminus\Gamma}\left(\frac{2i\pi b-\mu}{4}-\frac{2i\pi a-\nu}{4}\right)\right)\left(\prod_{a\in P\cup\Delta}\prod_{b\in Q\cup\Gamma}\left(\frac{2i\pi b-\mu}{4}-\frac{2i\pi a-\nu}{4}\right)\right)}{\left(\prod_{a\in P}\prod_{b\in Q}\left(\frac{2i\pi b-\mu}{4}-\frac{2i\pi a-\nu}{4}\right)\right)\left(\prod_{a\in P\ominus\Delta}\prod_{b\in Q\ominus\Gamma}\left(\frac{2i\pi b-\mu}{4}-\frac{2i\pi a-\nu}{4}\right)\right)}$$
$$=\prod_{a\in\Delta}\prod_{b\in\Gamma}\left(\frac{2i\pi b-\mu}{4}-\frac{2i\pi a-\nu}{4}\right)^{\frac{1-\sigma_a(P)\sigma_b(Q)}{2}} \tag{156}$$

lead for the general factor of the second product in (154) to

$$\frac{(1-e^{\nu_{\ell+1}-\nu_\ell})^{n_\ell}(1-e^{\nu_\ell-\nu_{\ell+1}})^{n_{\ell+1}}e^{-2K_{P_\ell,P_{\ell+1}}^{\Delta_\ell,\Delta_{\ell+1}}(\nu_\ell,\nu_{\ell+1})}}{\left(\prod_{a\in P_\ell}\prod_{b\in P_{\ell+1}}\left(\frac{2i\pi b-\nu_{\ell+1}}{4}-\frac{2i\pi a-\nu_\ell}{4}\right)\right)\left(\prod_{a\in H_\ell}\prod_{b\in H_{\ell+1}}\left(\frac{2i\pi b-\nu_{\ell+1}}{4}-\frac{2i\pi a-\nu_\ell}{4}\right)\right)} \tag{157}$$
$$=\frac{(1-e^{\nu_{\ell+1}-\nu_\ell})^{|\Delta_\ell|/2}(1-e^{\nu_\ell-\nu_{\ell+1}})^{|\Delta_{\ell+1}|/2}}{\left(\prod_{a\in\Delta_\ell}\prod_{b\in\Delta_{\ell+1}}\left(\frac{2i\pi b-\nu_{\ell+1}}{4}-\frac{2i\pi a-\nu_\ell}{4}\right)^{\frac{1+\sigma_a(A_\ell)\sigma_b(A_{\ell+1})}{2}}\right)}\times\left(e^{-2K^{\Delta_\ell,\Delta_{\ell+1}}}(p_\ell,p_{\ell+1})\right),$$

where $A_\ell = P_\ell \cap \Delta_\ell$, $p_\ell = [\nu_\ell, P_\ell]$ is a point on the Riemann surface $\mathcal{R}^{\Delta_\ell}$ and $e^{2K^{\Delta_\ell,\Delta_{\ell+1}}}$ a function meromorphic on $\mathcal{R}^{\Delta_\ell} \times \mathcal{R}^{\Delta_{\ell+1}}$ defined in (118). Writing $P_\ell = Q_\ell \cup A_\ell$ with $Q_\ell \cap \Delta_\ell = \emptyset$ and $A_\ell \subset \Delta_\ell$ and using that $\chi^\Delta([\nu,P]) = \chi^\Delta([\nu, P\setminus\Delta])$, $e^{2J^\Delta}([\nu,P]) = e^{2J^\Delta}([\nu, P\setminus\Delta])$, $e^{2K^{\Delta,\Gamma}}([\nu,P],[\mu,Q]) = e^{2K^{\Delta,\Gamma}}([\nu, P\setminus\Delta],[\mu, Q\setminus\Gamma])$ are independent of $P\cap\Delta$ and $Q\cap\Gamma$, we obtain

$$F(\vec{t},\vec{x},\vec{u})=\int_{c_1-i\pi}^{c_1+i\pi}\frac{d\nu_1}{2i\pi}\cdots\int_{c_m-i\pi}^{c_m+i\pi}\frac{d\nu_m}{2i\pi}\left(\prod_{\ell=1}^m\sum_{\Delta_\ell\sqsubset\mathbb{Z}+1/2}\sum_{\substack{A_\ell\sqsubset\Delta_\ell\\|A_\ell|=|\Delta_\ell\setminus A_\ell|}}\sum_{\substack{Q_\ell\sqsubset\mathbb{Z}+1/2\\Q_\ell\cap\Delta_\ell=\emptyset}}\right) \tag{158}$$
$$\times\left(\prod_{\ell=1}^m\left((i/4)^{|\Delta_\ell|}V_{A_\ell}^2 V_{\Delta_\ell\setminus A_\ell}^2 e^{2i\pi(x_\ell-x_{\ell-1})(\sum_{a\in A_\ell}a-\sum_{a\in\Delta_\ell\setminus A_\ell}a)}\right.\right.$$
$$\left.\left.\times\left(e^{(t_\ell-t_{\ell-1})\chi^{\Delta_\ell}-(u_\ell-u_{\ell-1})\chi'^{\Delta_\ell}+2J^{\Delta_\ell}}\right)(p_\ell)\right)\right)$$
$$\times\left(\prod_{\ell=1}^{m-1}\left(\frac{(1-e^{\nu_{\ell+1}-\nu_\ell})^{|\Delta_\ell|/2}(1-e^{\nu_\ell-\nu_{\ell+1}})^{|\Delta_{\ell+1}|/2}}{1-e^{\nu_{\ell+1}-\nu_\ell}}\right.\right.$$
$$\left.\left.\times\frac{e^{-2K^{\Delta_\ell,\Delta_{\ell+1}}}(p_\ell,p_{\ell+1})}{\prod_{a\in\Delta_\ell}\prod_{b\in\Delta_{\ell+1}}\left(\frac{2i\pi b-\nu_{\ell+1}}{4}-\frac{2i\pi a-\nu_\ell}{4}\right)^{\frac{1+\sigma_a(A_\ell)\sigma_b(A_{\ell+1})}{2}}}\right)\right),$$

where $p_\ell = [\nu_\ell, Q_\ell]$. This is essentially (13).

### 5.3.2 Pole structure of (16)

The integrand in (16) has potential poles and zeroes at $\nu_{\ell+1} = \nu_\ell + 2i\pi m$, $m \in \mathbb{Z}$. More precisely defining from (14), (15) and (118) the integer

$$\alpha_\ell = -1 + \left(\frac{|\Delta_\ell|}{2} + \frac{|\Delta_{\ell+1}|}{2} - |A_\ell \cap (A_{\ell+1}-m)| - |(\Delta_\ell\setminus A_\ell)\cap((\Delta_{\ell+1}\setminus A_{\ell+1})-m)|\right)$$
$$+\left(|P_\ell|+|P_{\ell+1}|+|\Delta_\ell\cap(\Delta_{\ell+1}-m)|\right. \tag{159}$$
$$\left.-|P_\ell\cap(P_{\ell+1}-m)|-|(P_\ell\cup\Delta_\ell)\cap((P_{\ell+1}\cup\Delta_{\ell+1})-m)|\right),$$

with $P_\ell \cap \Delta_\ell = \emptyset$, $A_\ell \subset \Delta_\ell$ and $|A_\ell| = |\Delta_\ell|/2$ as in (13), the integrand has at the point $v_{\ell+1} = v_\ell + 2i\pi m$ a zero of order $\alpha_\ell$ if $\alpha_\ell > 0$ and a pole of order $-\alpha_\ell$ if $\alpha_\ell < 0$. We show below that for any choice of the integer $m$ and of the sets $\Delta_\ell, A_\ell, P_\ell, \Delta_{\ell+1}, A_{\ell+1}, P_{\ell+1}$ as in (13), one has $\alpha_\ell \geq -1$, with $\alpha_\ell = -1$ if and only if $\Delta_{\ell+1} = \Delta_\ell + m$, $A_{\ell+1} = A_\ell + m$ and $P_{\ell+1} = P_\ell + m$.

We consider first the terms of (159) in the first line within the parenthesis. Using $|A_\ell \cap (A_{\ell+1} - m)| \leq \min(|A_\ell|, |A_{\ell+1}|) = \min(|\Delta_\ell|, |\Delta_{\ell+1}|)/2$, and similarly for $|(\Delta_\ell \setminus A_\ell) \cap ((\Delta_{\ell+1} \setminus A_{\ell+1}) - m)|$, the first parenthesis of (159) is non-negative, and equal to zero if and only if $\Delta_{\ell+1} = \Delta_\ell + m$ and $A_{\ell+1} = A_\ell + m$.

We consider then the terms in the second and third line of (159). After some manipulations, we observe that the sum of these terms is equal to $|P_\ell| - |P_\ell \cap ((P_{\ell+1} - m) \cup (\Delta_{\ell+1} - m))| + |P_{\ell+1}| - |(P_\ell \cup \Delta_\ell) \cap (P_{\ell+1} - m)|$, which is manifestly non-negative. The integrand in (13) may thus have a pole only if both parentheses in (159) are equal to zero. Provided that $\Delta_{\ell+1} = \Delta_\ell + m$, the second parenthesis is equal to zero if and only if $P_{\ell+1} = P_\ell + m$, which concludes the proof.

## A  Derivation of the identity (76)

In this appendix, we derive the identity (76) for the integral $\oint_{\beta_n \cdot P} \chi''(v)^2 dv$, $n \in \mathbb{Z}$. The identity is obviously true for $n = 0$ since $\beta_0 \cdot P$ is homotopic to an empty loop. We consider first the case $n = 1$, for which a detailed calculation is needed, and then generalize to arbitrary $n \in \mathbb{Z}$ by using translation properties of the functions $J_P$.

### A.1  Case $n = 1$

We introduce positive numbers $\epsilon, \delta$, $0 < \epsilon \ll \delta \ll 1$. By definition of the path $\beta_1 \cdot P$, one has

$$\oint_{\beta_1 \cdot P} \chi''(v)^2 dv = \int_{-\infty}^{i(\pi+\delta)-\epsilon} dv\, \chi''_P(v)^2 - \int_{-\infty}^{i(\pi+\delta)+\epsilon} dv\, \chi''_{P\ominus\{1/2\}}(v)^2 , \tag{160}$$

with both paths of integration contained in $\mathbb{D}$ on the right hand side. The path for the second integral on the right has to cross the imaginary axis in the interval $-\pi < \mathrm{Im}\, v < \pi$, see figure 20. At this point, $\delta$ need not be infinitesimal (we only require that $0 < \delta < 2\pi$), but it will be convenient in the following in order to compute some integrals by expanding close to the singularity at $v = i\pi$.

From (64), (63), (50) and $\kappa''_b = 3/\kappa_b$, the function $\chi''_P(v)$ verifies

$$\chi''_P(v) = \chi''_\emptyset(v) + \sum_{b \in P} \frac{2}{\kappa_b(v)} \tag{161}$$

and

$$\chi''_P(v) = \lim_{M \to \infty} \left( \sqrt{\frac{2M}{\pi}} - \sum_{b=-M+1/2}^{M-1/2} \frac{\sigma_b(P)}{\kappa_b(v)} \right). \tag{162}$$

Since $\sigma_{1/2}(P) = -\sigma_{1/2}(P \ominus \{1/2\})$, the function

$$\chi''_P(v) + \sigma_{1/2}(P)/\kappa_{1/2}(v) = \chi''_{P\ominus\{1/2\}}(v) - \sigma_{1/2}(P)/\kappa_{1/2}(v)$$

does not have a branch point at $v = i\pi$, so that

$$\int_{-\infty}^{i(\pi+\delta)-\epsilon} dv \left( \chi''_P(v) + \frac{\sigma_{1/2}(P)}{\kappa_{1/2}(v)} \right)^2 - \int_{-\infty}^{i(\pi+\delta)+\epsilon} dv \left( \chi''_{P\ominus\{1/2\}}(v) - \frac{\sigma_{1/2}(P)}{\kappa_{1/2}(v)} \right)^2 = 0 . \tag{163}$$

This leads to

$$\oint_{\beta_1 \cdot P} \chi''(v)^2 dv = \int_{-\infty}^{i(\pi+\delta)-\epsilon} dv \left( -\frac{1}{\kappa_{1/2}(v)^2} - 2\sigma_{1/2}(P) \frac{\chi_P''(v)}{\kappa_{1/2}(v)} \right) \tag{164}$$
$$-\int_{-\infty}^{i(\pi+\delta)+\epsilon} dv \left( -\frac{1}{\kappa_{1/2}(v)^2} + 2\sigma_{1/2}(P) \frac{\chi_{P\ominus\{1/2\}}''(v)}{\kappa_{1/2}(v)} \right),$$

which, using (161), gives

$$\oint_{\beta_1 \cdot P} \chi''(v)^2 dv = \sigma_{1/2}(P) \int_{-\infty}^{i(\pi+\delta)-\epsilon} dv \left( \frac{\sigma_{1/2}(P)-2}{\kappa_{1/2}(v)^2} - \frac{2\chi_\emptyset''(v)}{\kappa_{1/2}(v)} - \sum_{b\in P\setminus\{1/2\}} \frac{4}{\kappa_{1/2}(v)\kappa_b(v)} \right)$$
$$-\sigma_{1/2}(P) \int_{-\infty}^{i(\pi+\delta)+\epsilon} dv \left( \frac{\sigma_{1/2}(P)+2}{\kappa_{1/2}(v)^2} + \frac{2\chi_\emptyset''(v)}{\kappa_{1/2}(v)} + \sum_{b\in P\setminus\{1/2\}} \frac{4}{\kappa_{1/2}(v)\kappa_b(v)} \right). \tag{165}$$

The integrals needed are computed in appendix C. Using (184) and (185), we obtain after some simplifications

$$\frac{1}{2}\oint_{\beta_1 \cdot P} \chi''(v)^2 dv = i\pi + \sigma_{1/2}(P) \left( i\pi(|P|_+ - |P|_- - 1/2) - 2\log 2 \right. \tag{166}$$
$$\left. + \sum_{b\in P\setminus\{1/2\}} \log \frac{\pi^2(b-1/2)^2}{4} \right),$$

which is equivalent to (76) with $n = 1$.

## A.2 Extension to $n > 1$

For $n \geq 2$, we write the telescopic sum

$$J_{P-n}(v-2i\pi n) - J_{P\ominus B_n}(v) = \sum_{m=1}^{n} \left( J_{(P-m)\ominus B_{n-m}}(v-2i\pi m) \right. \tag{167}$$
$$\left. -J_{(P-m+1)\ominus B_{n-m+1}}(v-2i\pi(m-1)) \right).$$

Noting that $(P-m+1)\ominus B_{n-m+1} = ((P-m)\ominus B_{n-m}+1)\ominus\{1/2\}$, see the group identity (36), one has from (75)

$$\frac{1}{2}\oint_{\beta_n \cdot P} \chi''(v)^2 dv = \frac{1}{2} \sum_{m=1}^{n} \oint_{\beta_1 \cdot ((P-m)\ominus B_{n-m}+1)} \chi''(v)^2 dv . \tag{168}$$

Using the identity (76) with $n = 1$ derived previously, we arrive at

$$\frac{1}{2}\oint_{\beta_n \cdot P} \chi''(v)^2 dv = \frac{1}{2} \sum_{m=1}^{n} \left( W_{(P-m+1)\ominus B_{n-m+1}} - W_{(P-m)\ominus B_{n-m}} \right), \tag{169}$$

whose right hand side telescopically reduces to $(W_{P\ominus B_n} - W_{P-n})/2$. This proves (76) for $n \geq 2$.

## A.3 Extension to $n < 0$

Let $n$ be a positive integer. The replacements $v \to v + 2i\pi n$ and $P \to (P \ominus B_n) - n$ in (75) give

$$J_{P\ominus B_n-n}(v) = J_P(v+2i\pi n) + \frac{1}{2}\oint_{\beta_n \cdot (P\ominus B_n)} \chi''(v)^2 dv . \tag{170}$$

Changing the order of the terms and using $P \ominus B_n - n = (P - n) \ominus B_{-n}$, one has

$$J_P(v + 2i\pi n) = J_{(P-n)\ominus B_{-n}}(v) - \frac{1}{2}\int_{\beta_n \cdot (P \ominus B_n)} \chi''(v)^2 dv \,, \tag{171}$$

which, from (75) with $n$ replaced by $-n$, gives

$$\frac{1}{2}\int_{\beta_{-n}\cdot(P-n)} \chi''(v)^2 dv = -\frac{1}{2}\int_{\beta_n\cdot(P\ominus B_n)} \chi''(v)^2 dv \,. \tag{172}$$

Using the identity (76) for $n > 0$ derived previously, this leads to

$$\frac{1}{2}\int_{\beta_{-n}\cdot(P-n)} \chi''(v)^2 dv = -W_P + W_{P\ominus B_n -n} \,. \tag{173}$$

Replacing $P$ by $P + n$ finally leads to (76) with $n$ replaced by $-n < 0$.

## B  Derivation of the identities (106) and (107)

In this appendix, we derive the identities (106) and (107) for some integrals over the paths $\gamma_{n,0}$ and $\gamma_{0,m}$. The identities are obviously true for $n = 0$ or $m = 0$ since the paths are then homotopic to empty loops. We start with the identity (106), which is proved first for $n = 1$, where a detailed calculation is needed, and then generalized to arbitrary $n \in \mathbb{Z}$ by using translation properties of the functions $K_{P,Q}$. The identity (107) is then obtained by exchanging $\mu$ with $v$ and $P$ with $Q$.

### B.1  Case $n = 1$ for (106)

The integral $\oint_{\gamma_{1,0}} du \, \mathcal{A}_u(\chi_P''(\cdot + v)\chi_Q''(\cdot + \mu))$ is equal to zero if $\text{Re } v < 0$ since the path $\gamma_{1,0}$ is empty then. Therefore, we restrict to $\text{Re } v > 0$ in the rest of this section. We introduce positive numbers $\epsilon, \delta, 0 < \epsilon \ll \delta \ll 1$. We want to compute the integral

$$\oint_{\gamma_{1,0}} du \, \mathcal{A}_u(\chi_P''(\cdot + v)\chi_Q''(\cdot + \mu)) \tag{174}$$

$$= \int_{-\infty}^{i(\pi+\delta)-v-\epsilon} du \, \chi_P''(u+v)\chi_Q''(u+\mu) - \int_{-\infty}^{i(\pi+\delta)-v+\epsilon} du \, \chi_{P\ominus\{1/2\}}''(u+v)\chi_Q''(u+\mu) \,,$$

where the paths of integration in the second line are contained in $\mathbb{D}$. The path for the last integral in the second line has to cross the imaginary axis in the interval $-\pi < \text{Im } v < \pi$, see figure 20. Additionally, if $\text{Re } \mu > \text{Re } v$, the paths of both integrals in the second line must cross the line $\text{Re}(u + \mu) = 0$ in the interval $-\pi < \text{Im}(u + \mu) < \pi$. At this point, $\delta$ need not be infinitesimal (we only require that $0 < \delta < 2\pi$), but it will be convenient in the following in order to compute some integrals by expanding close to the singularity at $u = i\pi - v$.

Because of (162), the function $\chi_P''(v) + \sigma_{1/2}(P)/\kappa_{1/2}(v) = \chi_{P\ominus\{1/2\}}''(v) - \sigma_{1/2}(P)/\kappa_{1/2}(v)$ does not have a branch point at $v = i\pi$, so that

$$\int_{-\infty}^{i(\pi+\delta)-v-\epsilon} du \left(\chi_P''(u+v) + \frac{\sigma_{1/2}(P)}{\kappa_{1/2}(u+v)}\right)\chi_Q''(u+\mu) \tag{175}$$

$$= \int_{-\infty}^{i(\pi+\delta)-v+\epsilon} du \left(\chi_P''(u+v) - \frac{\sigma_{1/2}(P)}{\kappa_{1/2}(u+v)}\right)\chi_Q''(u+\mu) = 0 \,.$$

This leads to

$$\fint_{\gamma_{1,0}} \mathrm{d}u\, \mathcal{A}_u(\chi_P''(\cdot + \nu)\chi_Q''(\cdot + \mu)) \tag{176}$$

$$= -\sigma_{1/2}(P)\left(\fint_{-\infty}^{\mathrm{i}(\pi+\delta)-\nu-\epsilon} \mathrm{d}u\, \frac{\chi_Q''(u+\mu)}{\kappa_{1/2}(u+\nu)} + \int_{-\infty}^{\mathrm{i}(\pi+\delta)-\nu+\epsilon} \mathrm{d}u\, \frac{\chi_Q''(u+\mu)}{\kappa_{1/2}(u+\nu)}\right).$$

Using (161), one has

$$\fint_{\gamma_{1,0}} \mathrm{d}u\, \mathcal{A}_u(\chi_P''(\cdot + \nu)\chi_Q''(\cdot + \mu)) \tag{177}$$

$$= -\sigma_{1/2}(P)\left(\int_{-\infty}^{\mathrm{i}(\pi+\delta)-\nu-\epsilon} \mathrm{d}u\, \frac{\chi_\emptyset''(u+\mu)}{\kappa_{1/2}(u+\nu)} + \int_{-\infty}^{\mathrm{i}(\pi+\delta)-\nu+\epsilon} \mathrm{d}u\, \frac{\chi_\emptyset''(u+\mu)}{\kappa_{1/2}(u+\nu)}\right.$$

$$\left. + 2\sum_{b\in Q}\left(\fint_{-\infty}^{\mathrm{i}(\pi+\delta)-\nu-\epsilon} \frac{\mathrm{d}u}{\kappa_{1/2}(u+\nu)\kappa_b(u+\mu)} + \fint_{-\infty}^{\mathrm{i}(\pi+\delta)-\nu+\epsilon} \frac{\mathrm{d}u}{\kappa_{1/2}(u+\nu)\kappa_b(u+\mu)}\right)\right).$$

The remaining integrals are computed in appendix D. Using (194) and (201), we find

$$\frac{1}{2}\fint_{\gamma_{1,0}} \mathrm{d}u\, \mathcal{A}_u(\chi_P''(\cdot + \nu)\chi_Q''(\cdot + \mu)) = 2\sigma_{1/2}(P)\left(I_0(\mu - \nu + \mathrm{i}\pi) + \sum_{b\in Q}\log\frac{\kappa_b(\mu - \nu + \mathrm{i}\pi)}{\sqrt{8}}\right). \tag{178}$$

Noting from (103) that $W_{P\ominus\{1/2\},Q}(z) - W_{P,Q}(z) = 2\sigma_{1/2}(P)(I_0(z) + \sum_{b\in Q}\log\frac{\kappa_b(z)}{\sqrt{8}})$ and from (104) that $Z_{P\ominus\{1/2\},Q} - Z_{P,Q} = 0$, we finally obtain (106) for $n = 1$ after replacing $P$ by $P+1$ in (178).

### B.2 Extension to $n > 1$ and $n < 0$

The extension of (106) from $n = 1$ to all $n \in \mathbb{Z}$ works essentially the same as in appendix A.

### B.3 Proof of (107)

Exchanging $\nu$ with $\mu$ and replacing $n$ by $m$ transforms the path $\gamma_{n,0}$ to $\gamma_{0,m}$. Replacing also $P$ with $Q$ in (106) gives

$$\frac{1}{2}\fint_{\gamma_{0,m}} \mathrm{d}u\, \mathcal{A}_u(\chi_{Q+m}''(\cdot + \mu)\chi_P''(\cdot + \nu)) \tag{179}$$

$$= 1_{\{\mathrm{Re}\,\mu > 0\}}\left(W_{(Q+m)\ominus B_m,P}(\nu - \mu + \mathrm{i}\pi) - W_{Q+m,P}(\nu - \mu + \mathrm{i}\pi)\right.$$

$$\left. + 1_{\{\mathrm{Re}\,\nu > \mathrm{Re}\,\mu\}}(Z_{(Q+m)\ominus B_m,P} - Z_{Q+m,P})\right).$$

Using (105) then leads to (107).

## C Calculations of some integrals between $-\infty$ and $\nu \in \mathbb{D}$

In this appendix, we compute some integrals between $-\infty$ and $\nu \in \mathbb{D}$, with a path of integration contained in $\mathbb{D}$. For some integrals, indicated by the symbol $\fint$, a regularization at $-\infty$ is needed due to the presence of logarithmic divergences.

## C.1 The functions $\log \kappa_a$ and $\log(\kappa_a + \kappa_b)$ are analytic in $\mathbb{D}$

For any $a \in \mathbb{Z} + 1/2$, the function $\kappa_a$ defined in (50) is analytic in $\mathbb{D}$. When $\operatorname{Re} \nu = 0$, $-\pi < \operatorname{Im} \nu < \pi$ and $\arg(\kappa_a(\nu)) = \operatorname{sgn}(a)\pi/4$. Otherwise, $\operatorname{Re} \nu$ is non-zero and one has

$$\arg(\kappa_a(\nu)) \in \begin{cases} (-3\pi/4, -\pi/4) & \operatorname{Re} \nu > 0 \text{ and } a < 0 \\ (\pi/4, 3\pi/4) & \operatorname{Re} \nu > 0 \text{ and } a > 0 \\ (-\pi/4, \pi/4) & \operatorname{Re} \nu < 0 \end{cases}, \qquad (180)$$

which implies in particular that $\log \kappa_a$ is analytic in $\mathbb{D}$ if the branch cut of the logarithm is chosen as $\mathbb{R}^-$.

For $a, b \in \mathbb{Z} + 1/2$ and $\nu \in \mathbb{D}$, the discussion above imply constraints on the argument of $\kappa_a(\nu) + \kappa_b(\nu)$, $a, b \in \mathbb{Z} + 1/2$. When $\operatorname{Re} \nu = 0$, $-\pi < \operatorname{Im} \nu < \pi$, one has $\arg(\kappa_a(\nu) + \kappa_b(\nu)) = (\operatorname{sgn}(a) + \operatorname{sgn}(b))i\pi/4$. Otherwise, $\operatorname{Re} \nu$ is non-zero and (180) implies

$$\arg(\kappa_a(\nu) + \kappa_b(\nu)) \in \begin{cases} (-3\pi/4, -\pi/4) & \operatorname{Re} \nu > 0, a < 0 \text{ and } b < 0 \\ (\pi/4, 3\pi/4) & \operatorname{Re} \nu > 0, a > 0 \text{ and } b > 0 \\ (-\pi/4, \pi/4) & \operatorname{Re} \nu > 0, ab < 0 \\ (-\pi/4, \pi/4) & \operatorname{Re} \nu < 0 \end{cases}. \qquad (181)$$

In the case $\operatorname{Re} \nu > 0$, $ab < 0$, we have used $\kappa_a(\nu) + \kappa_b(\nu) = 4i\pi(a - b)/(\kappa_a(\nu) - \kappa_b(\nu))$. We observe in particular from (181) that $\arg(\kappa_a(\nu) + \kappa_b(\nu))$ always stays between $-3\pi/4$ and $3\pi/4$, and thus $\kappa_a(\nu) + \kappa_b(\nu)$ never reaches the negative real axis $\mathbb{R}^-$. The function $\log(\kappa_a(\nu) + \kappa_b(\nu))$ is thus analytic in $\mathbb{D}$ for any $a, b \in \mathbb{Z} + 1/2$ with the branch cut of the logarithm chosen as $\mathbb{R}^-$.

## C.2 Integral of $\kappa_a(\nu)^{-1}\kappa_b(\nu)^{-1}$

Let $a, b \in \mathbb{Z} + 1/2$ and $\nu \in \mathbb{D}$. We consider the integral $\int_{-\Lambda}^{\nu} d\nu \, \kappa_a(\nu)^{-1}\kappa_b(\nu)^{-1}$ with a path of integration staying in $\mathbb{D}$ and $|\Lambda| \to \infty$, $\operatorname{Re} \Lambda > 0$. The identity

$$\frac{1}{\kappa_a(\nu)\kappa_b(\nu)} = -\partial_\nu \log(\kappa_a(\nu) + \kappa_b(\nu)) \qquad (182)$$

allows to compute the integral explicitly since $\log(\kappa_a(\nu) + \kappa_b(\nu))$ is analytic in $\mathbb{D}$ when the branch cut of the logarithm is chosen as $\mathbb{R}^-$, as showed in appendix C.1. Taking $|\Lambda| \to \infty$, $\operatorname{Re} \Lambda > 0$, one finds

$$\int_{-\Lambda}^{\nu} \frac{d\nu}{\kappa_a(\nu)\kappa_b(\nu)} \simeq \log \sqrt{8\Lambda} - \log(\kappa_a(\nu) + \kappa_b(\nu)). \qquad (183)$$

For any $\nu \in \mathbb{D}$, the regularized integral is then equal to

$$\fint_{-\infty}^{\nu} \frac{d\nu}{\kappa_a(\nu)\kappa_b(\nu)} = -\log\left(\frac{\kappa_a(\nu) + \kappa_b(\nu)}{\sqrt{8}}\right). \qquad (184)$$

## C.3 Integral of $\chi_\emptyset''(\nu)/\kappa_a(\nu)$ between $-\infty$ and a branch point

Let $\delta$ be a positive real number and $\theta \neq 0$, $-\pi < \theta < \pi$. In the limit $\delta \to 0$ one has

$$\int_{-\infty}^{i(2\pi a + e^{i\theta}\delta)} d\nu \, \frac{\chi_\emptyset''(\nu)}{\kappa_a(\nu)} \simeq \log \sqrt{4\delta} + \frac{i\theta}{2} - \operatorname{sgn}(\theta)\frac{i\pi}{4} + 1_{\{\theta<0\}} i\pi a, \qquad (185)$$

which can be derived by taking $\mu = \nu = 0$ and $\upsilon = i(2\pi a + e^{i\theta}\delta)$ in (200), after rather tedious simplifications using $\kappa_a(i(2\pi a + e^{i\theta}\delta)) = e^{\frac{i\theta}{2} - (1 - 4 1_{\{\theta<0\}} 1_{\{a>0\}})\frac{i\pi}{4}}\sqrt{2\delta}$, $\kappa_b(i(2\pi a + e^{i\theta}\delta)) \simeq \operatorname{sgn}(\theta) e^{-\operatorname{sgn}(a-b)\frac{i\pi}{4}}\sqrt{|4\pi(a-b)|}$ if $\operatorname{sgn}(a) = \operatorname{sgn}(b)$ and $|b| < |a|$, and $\kappa_b(i(2\pi a + e^{i\theta}\delta)) \simeq e^{-\operatorname{sgn}(a-b)\frac{i\pi}{4}}\sqrt{|4\pi(a-b)|}$ otherwise.

## C.4  Integral of $\chi_\emptyset''(\nu)/\kappa_a(\nu)$ as an infinite sum

The indefinite integral of $\chi_\emptyset''(\nu)/\kappa_a(\nu)$ can be rewritten as an infinite sum by expanding $\chi_\emptyset''(\nu)$ as in (162) and computing the integrals using (51) and (184). After careful treatment of the exchange between the integral and the infinite sum, one has

$$\int_{-\infty}^{\nu} d\nu \, \frac{\chi_\emptyset''(\nu)}{\kappa_a(\nu)} = \lim_{M\to\infty} \left( -2I_0(\nu) - \sqrt{\frac{2M}{\pi}} \kappa_a(\nu) + \frac{\kappa_a^2(\nu)}{8} \right. \tag{186}$$
$$\left. + \sum_{b=-M+1/2}^{M-1/2} \log\left(1 + \frac{\kappa_a(\nu)}{\kappa_b(\nu)}\right) \right),$$

with $I_0$ defined in (68). When $\kappa_a(\nu)$ is small enough, i.e. when $\nu$ is in the vicinity of $2i\pi a$, the logarithm can be expanded. Using $2I_0^{(m)}(\nu) = -\delta_{m,1}/4 + (2m-2)!! \sum_{b\in\mathbb{Z}+1/2} \kappa_b(\nu)^{-2m}$ and $\chi_\emptyset^{(m+2)}(\nu) = -(2m-1)!! \sum_{b\in\mathbb{Z}+1/2} \kappa_b(\nu)^{-2m-1}$ for $m \geq 1$, we obtain the identity

$$\int_{-\infty}^{\nu} d\nu \, \frac{\chi_\emptyset''(\nu)}{\kappa_a(\nu)} = -\sum_{m=0}^{\infty} \left( \frac{\kappa_a(\nu)^{2m}}{(2m)!!} 2I_0^{(m)}(\nu) + \frac{\kappa_a(\nu)^{2m+1}}{(2m+1)!!} \chi_\emptyset^{(m+2)}(\nu) \right). \tag{187}$$

# D  Calculations of some integrals depending on $(\nu,\mu) \in \mathbb{D}_2$

In this appendix, we compute some integrals depending on two variables $\nu$ and $\mu$.

## D.1  Domain of analyticity of functions $\log(e^{i\theta}(\kappa_a(\nu) + \kappa_b(\mu)))$

Let $a, b \in \mathbb{Z} + 1/2$ and $(\nu,\mu) \in \mathbb{D}_2$. We are interested in the position in the complex plane of $\kappa_a(\nu) + \kappa_b(\mu)$. Using (180), we obtain

$$\arg\left(i^{\frac{\text{sgn}(a)-\text{sgn}(b)}{2}}(\kappa_a(\nu)+\kappa_b(\mu))\right) \in \begin{cases} (-3\pi/4, \pi/4) & a<0 \quad b<0 \\ (-\pi/4, 3\pi/4) & a>0 \quad b>0 \\ (-3\pi/4, 3\pi/4) & ab<0 \end{cases} \tag{188}$$

for $\text{Re}\,\nu < \text{Re}\,\mu$,

$$\arg\left(i^{\frac{\text{sgn}(b)-\text{sgn}(a)}{2}}(\kappa_a(\nu)+\kappa_b(\mu))\right) \in \begin{cases} (-3\pi/4, \pi/4) & a<0 \quad b<0 \\ (-\pi/4, 3\pi/4) & a>0 \quad b>0 \\ (-3\pi/4, 3\pi/4) & ab<0 \end{cases} \tag{189}$$

for $\text{Re}\,\nu > \text{Re}\,\mu$, and

$$\arg(\kappa_a(\nu)+\kappa_b(\mu)) \in \begin{cases} (-3\pi/4, \pi/4) & a<0 \quad b<0 \\ (-\pi/4, 3\pi/4) & a>0 \quad b>0 \\ (-\pi/4, +\pi/4) & ab<0 \end{cases} \tag{190}$$

for $\text{Re}\,\nu = \text{Re}\,\mu$, $0 < \text{Im}(\nu-\mu) < 2\pi$.

This implies in particular that the function $(\nu,\mu) \mapsto \log\left(i^{\frac{\text{sgn}(a)-\text{sgn}(b)}{2}}(\kappa_a(\nu) + \kappa_b(\mu))\right)$ is analytic in the domain $\{(\nu,\mu) \in \mathbb{D} \times \mathbb{D}, \text{Re}\,\nu < \text{Re}\,\mu\}$ while the function $(\nu,\mu) \mapsto \log\left(i^{\frac{\text{sgn}(b)-\text{sgn}(b)}{2}}(\kappa_a(\nu)+\kappa_b(\mu))\right)$ is analytic in the domain $\{(\nu,\mu) \in \mathbb{D} \times \mathbb{D}, \text{Re}\,\nu > \text{Re}\,\mu\}$.

## D.2 Integral of $\kappa_a(u+v)^{-1}\kappa_b(u+\mu)^{-1}$

Let $a, b \in \mathbb{Z} + 1/2$. We consider the integral $\int_{-\Lambda}^{v} du\, \kappa_a(u+v)^{-1}\kappa_b(u+\mu)^{-1}$ with a path of integration such that $(u+v, u+\mu)$ stays in $\mathbb{D}_2$ and $|\Lambda| \to \infty$, $\mathrm{Re}\,\Lambda > 0$. The identity

$$\frac{1}{\kappa_a(u+v)\kappa_b(u+\mu)} = -\partial_u \log(e^{i\theta}(\kappa_a(u+v) + \kappa_b(u+\mu))) \tag{191}$$

allows to compute the integral explicitly by choosing $\theta$ appropriately so that $\log(e^{i\theta}(\kappa_a(u+v) + \kappa_b(u+\mu)))$ is analytic everywhere on the path of integration, with $\mathbb{R}^-$ the branch cut of the logarithm. According to appendix D.1, one can take $\theta = \theta_{a,b}(\mu - v)$ with

$$\theta_{a,b}(z) = \pi \,\mathrm{sgn}(\mathrm{Re}\,z)\,\frac{\mathrm{sgn}(a) - \mathrm{sgn}(b)}{4} \,. \tag{192}$$

Subtracting the leading term $\log\sqrt{\Lambda}$ in the limit $\Lambda \to \infty$, $\mathrm{Re}\,\Lambda > 0$, the regularized integral is finally equal to

$$\fint_{-\infty}^{v} \frac{du}{\kappa_a(u+v)\kappa_b(u+\mu)} = i\theta_{a,b}(\mu - v) - \log\left(e^{i\theta_{a,b}(\mu-v)}\,\frac{\kappa_a(v+v) + \kappa_b(v+\mu)}{\sqrt{8}}\right). \tag{193}$$

This expression simplifies further when $v \to 2i\pi a - v$ since then $\kappa_a(v+v) \to 0$, and we obtain

$$\fint_{-\infty}^{2i\pi a - v} \frac{du}{\kappa_a(u+v)\kappa_b(u+\mu)} = -\log\left(\frac{\kappa_b(\mu - v + 2i\pi a)}{\sqrt{8}}\right). \tag{194}$$

## D.3 Integral of $\kappa_b(u+\mu)/\kappa_a(u+v)$

Let $a, b \in \mathbb{Z} + 1/2$. We consider the integral $\int_{-\Lambda}^{v} du\, \kappa_b(u+\mu)/\kappa_a(u+v)$ with a path of integration such that $(u+v, u+\mu)$ stays in $\mathbb{D}_2$ and $|\Lambda| \to \infty$, $\mathrm{Re}\,\Lambda > 0$. The identity

$$\frac{\kappa_b(u+\mu)}{\kappa_a(u+v)} = -\partial_u\Bigg(\frac{\kappa_a(u+v)\kappa_b(u+\mu)}{2} \tag{195}$$
$$+ \big((v - 2i\pi a) - (\mu - 2i\pi b)\big)\log\big(e^{i\theta}(\kappa_a(u+v) + \kappa_b(u+\mu))\big)\Bigg)$$

allows to compute the integral explicitly by choosing $\theta = \theta_{a,b}(\mu - v)$ defined in (192) so that $\log(e^{i\theta}(\kappa_a(u+v) + \kappa_b(u+\mu)))$ is analytic everywhere on the path of integration. The contribution of the lower limit of the integral $-\Lambda$ is equal to

$$\Lambda + \big((v - 2i\pi a) - (\mu - 2i\pi b)\big)\log(\sqrt{8\Lambda}\,e^{i\theta_{a,b}(\mu-v)}) - \frac{v + \mu - 2i\pi(a+b)}{2}\,, \tag{196}$$

when $\Lambda \to \infty$. Defining the regularized integral by subtracting the divergent term $\Lambda + \frac{(v-2i\pi a)-(\mu-2i\pi b)}{2}\log\Lambda$ finally leads to

$$\fint_{-\infty}^{v} du\,\frac{\kappa_b(u+\mu)}{\kappa_a(u+v)} = -\frac{v + \mu - 2i\pi(a+b)}{2} - \frac{\kappa_a(v+v)\kappa_b(v+\mu)}{2} \tag{197}$$
$$+ \big((v - 2i\pi a) - (\mu - 2i\pi b)\big)\left(i\theta_{a,b}(\mu - v) - \log\left(e^{i\theta_{a,b}(\mu-v)}\,\frac{\kappa_a(v+v) + \kappa_b(v+\mu)}{\sqrt{8}}\right)\right).$$

### D.4 Integral of $\chi_\emptyset''(u+\mu)/\kappa_a(u+v)$

Let $a \in \mathbb{Z}+1/2$. We consider the integral $\int_{-\infty}^{v} du\, \chi_\emptyset''(u+\mu)/\kappa_a(u+v)$ with a path of integration such that $(u+v, u+\mu)$ stays in $\mathbb{D}_2$. Let $M$ be a positive integer, that will be taken to infinity in the end. Using (61), $\partial_u \zeta(s,u) = -s\zeta(s+1,u)$ and (51), one has

$$\int_{-\infty}^{v} du\, \frac{\chi_\emptyset''(u+\mu)}{\kappa_a(u+v)} = -\sum_{b=-M+1/2}^{M-1/2} \int_{-\infty}^{v} \frac{du}{\kappa_a(u+v)\kappa_b(u+\mu)} \tag{198}$$
$$-\frac{1}{2\sqrt{\pi}} \int_{-\infty}^{v} du\, \frac{e^{i\pi/4}\zeta(\frac{1}{2}, M+\frac{1}{2}+\frac{u+\mu}{2i\pi}) + e^{-i\pi/4}\zeta(\frac{1}{2}, M+\frac{1}{2}-\frac{u+\mu}{2i\pi})}{\kappa_a(u+v)}.$$

The large $M$ asymptotics of the integral in the second line is dominated by the contributions $u \sim M$, for which (59) gives

$$-\frac{1}{2\sqrt{\pi}}\left(e^{i\pi/4}\zeta\left(\frac{1}{2}, M+\frac{1}{2}+\frac{u+\mu}{2i\pi}\right) + e^{-i\pi/4}\zeta\left(\frac{1}{2}, M+\frac{1}{2}-\frac{u+\mu}{2i\pi}\right)\right)$$
$$= \frac{\kappa_M(u+\mu) - \kappa_{-M}(u+\mu)}{2i\pi} + \mathcal{O}(M^{-3/2}). \tag{199}$$

The integrals can then be computed using (193) and (197). After some simplifications, one finds

$$\int_{-\infty}^{v} du\, \frac{\chi_\emptyset''(u+\mu)}{\kappa_a(u+v)} = \lim_{M\to\infty}\left(-M\log M + M\left(1 - \log\frac{\pi}{2}\right) - \frac{\sqrt{2M}}{\sqrt{\pi}}\kappa_a(v+v) \right. \tag{200}$$
$$\left. -\frac{v-\mu-2i\pi a}{4} - \sum_{b=-M+1/2}^{M-1/2}\left(i\theta_{a,b}(\mu-v) - \log\left(e^{i\theta_{a,b}(\mu-v)}\frac{\kappa_a(v+v) + \kappa_b(v+\mu)}{\sqrt{8}}\right)\right)\right),$$

with $\theta_{a,b}$ defined in (192). This expression can be simplified further when $v \to 2i\pi a - v$ since $\kappa_a(v+v) \to 0$. Using (71), we obtain

$$\int_{-\infty}^{2i\pi a-v} du\, \frac{\chi_\emptyset''(u+\mu)}{\kappa_a(u+v)} = -2I_0(\mu-v+2i\pi a). \tag{201}$$

# E   Identities for the coefficients $W_P$, $W_P^\Delta$

In this appendix, we give some identities for the coefficients $W_P$ and $W_P^\Delta$ defined in (77) and (94).

## E.1   Differences of $W_P$

We observe that for any $P \sqsubset \mathbb{Z}+1/2$, the last two terms in the definition (77) are unchanged if the set $P$ is replaced by $P - n$, $n \in \mathbb{Z}$. After some manipulations using $|P|_+ + |P|_- = |P|$, $|P|_+ - |P|_- = \sum_{a\in P}\text{sgn}(a)$ and $\text{sgn}(a-n) = \sigma_a(B_n)\text{sgn}(a)$, one has

$$W_P - W_{P-n} = -i\pi n|P| + \text{sgn}(n)2i\pi|P|\,|P \cap B_n|, \tag{202}$$

with $\text{sgn}(0) = 0$. Using this identity together with $|P \ominus B_n| = |P| + |n| - 2|P \cap B_n|$ and $|(P \ominus B_n) \cap B_n| = |n| - |P \cap B_n|$, we obtain

$$(W_{P\ominus B_n} - W_{P-n}) - (W_{P\ominus B_n - n} - W_P) = \text{sgn}(n)i\pi(|n| - 2|P \cap B_n|)^2, \tag{203}$$

which can be rewritten as

$$(W_{P \ominus B_n} - W_{P-n}) - (W_{(P \ominus B_n)-n} - W_P) \tag{204}$$
$$= i\pi n + 2i\pi \, \text{sgn}(n)\Big(\frac{|n|(|n|-1)}{2} - 2|n| \, |P \cap B_n| + 2|P \cap B_n|^2\Big).$$

In particular, one has

$$(W_{P \ominus B_n} - W_{P-n}) - (W_{P \ominus B_n-n} - W_P) \in i\pi n + 2i\pi \mathbb{Z}. \tag{205}$$

## E.2  Ratios of $e^{W_P}$

We consider now the quantity $e^{W_P}$. Using again $|P|_+^2 - |P|_-^2 = |P| \sum_{a \in P} \text{sgn}(a)$, one has in terms of the Vandermonde determinant $V_P$ defined in (5) the identity

$$e^{W_P} = (-1)^{\frac{|P|(|P|-1)}{2}} e^{-i\pi \sum_{a \in P} a} \frac{(-1)^{|P|} V_P^2}{2^{2|P|}}. \tag{206}$$

Considering ratios, one has in particular

$$e^{W_{P-n} - W_P} = (-1)^{n|P|}, \tag{207}$$

which follows also directly from (202).

We are also interested in the quantity $e^{W_{P \ominus B_n} - W_P}$, which can be computed from (206) using the summation identity

$$\sum_{a \in P \ominus Q} f(a) - \sum_{a \in P} f(a) = \sum_{a \in Q} \sigma_a(P) f(a), \tag{208}$$

where $\sigma_a$ is defined in (29). The summation identity (208) is proved easily from the Venn diagram for the sets $P$ and $Q$. Applied to $Q = B_n$, $f(a) = a$, it gives after some simplifications for $P \sqsubset \mathbb{Z} + 1/2$

$$e^{-i\pi \sum_{a \in P \ominus B_n} a} = e^{-i\pi \sum_{a \in P} a} \times (-1)^{n(n+1)/2} \, i^{\sum_{a \in B_n} \sigma_a(P)}. \tag{209}$$

Similarly, the summation identity (208) implies $|P \ominus B_n| = |P| + \sum_{a \in B_n} \sigma_a(P)$, which leads to

$$(-1)^{\frac{|P \ominus B_n|(|P \ominus B_n|-1)}{2}} = (-1)^{\frac{|P|(|P|-1)}{2}} \times i^{-n} (-1)^{n|P|} (-1)^{n(n+1)/2} (-i)^{\sum_{a \in B_n} \sigma_a(P)}. \tag{210}$$

Putting together the two identities above with (206), we finally obtain

$$e^{W_{P \ominus B_n} - W_P} = i^{-n} (-1)^{n|P|} \times \frac{(-1)^{|P \ominus B_n|} V_{P \ominus B_n}^2}{2^{2|P \ominus B_n|}} \Big/ \frac{(-1)^{|P|} V_P^2}{2^{2|P|}}. \tag{211}$$

## E.3  Ratios of $e^{2W_P^\Delta}$

We consider the quantity $e^{2W_P^\Delta}$ with $W_P^\Delta$ defined in (94). Similar simplifications as in the previous section give

$$e^{2W_{(P+n) \ominus (B_n \setminus (\Delta+n))}^{\Delta+n} - 2W_P^\Delta} = \frac{X_{(P+n) \ominus (B_n \setminus (\Delta+n))}^{\Delta+n}}{X_P^\Delta} \tag{212}$$

and

$$e^{2W^{\Delta}_{P\ominus(B_n\backslash\Delta)}-2W^{\Delta-n}_{P-n}} = e^{2W^{\Delta-n}_{P\ominus(B_n\backslash\Delta)-n}-2W^{\Delta}_P} = \frac{X^{\Delta}_{P\ominus(B_n\backslash\Delta)}}{X^{\Delta}_P}, \tag{213}$$

with

$$X^{\Delta}_P = (i/4)^{2|P\backslash\Delta|+|\Delta|} \prod_{a\in P\backslash\Delta} \prod_{\substack{b\in P\cup\Delta \\ b\neq a}} \Big(\frac{2i\pi a}{4} - \frac{2i\pi b}{4}\Big)^2. \tag{214}$$

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
