# Peer review of "Riemann surfaces for KPZ with periodic boundaries"

_SciPost Physics, doi:SciPost Phys. 8, 008 (2020)_

## Round 1 · Referee Report · Anonymous (Referee 1) · 2019-10-4

Report

This paper presents a way to formulate the distributions of KPZ universality in finite volume (in the crossover regime which interpolates Gaussian and KPZ fluctuations) in terms of meromorphic functions on some Riemann surfaces. Such distributions were obtained previously by the author and Baik-Liu using different approaches, and were expressed in different formulas, both of which are quite complicated so that it was not clear how they match mathematically (although numerically they do match well). In this paper the author provides a proof to the equivalence of previous formulas by two groups. Considering the complicated nature of these distributions,the result of this paper is nontrivial and deserves appreciation of the community.

Another contribution of this paper is to related the distribution function arising finite volume KPZ universality with KdV equations (or more generally, the KP equation). This is very interesting. Similar relation for infinite KPZ universality was observed by Quastel-Remenik at the same time.

Although I did not check the explicit computations of the formulas, the result is convincing since it matches the known results. The relation to KdV equation may generate interest from math and physics community. In conclusion, I strongly recommend publication of this paper.
  • validity: top
  • significance: top
  • originality: top
  • clarity: top
  • formatting: excellent
  • grammar: excellent

Author:  Sylvain Prolhac  on 2019-11-20  [id 651]

(in reply to Report 1 on 2019-10-04)

I thank the referee for the positive feedback on the paper.

---

## Round 1 · Referee Report · Anonymous (Referee 2) · 2019-10-25

Strengths

1- An original contribution, using tools from Riemann surfaces and analytic continuation to provide a more natural setting to express exact formulas for KPZ in finite volume. To the best of my knowledge, this is the first time these tools have been used in this context. 2- Provides a possible framework to study these type of problems (which have usually involved very complicated formulas) in a more streamlined fashion in the future. 3- Establishes a remarkable fact, showing that the distribution of KPZ in finite volume with flat initial data can be understoo in terms of a gas of infinitely many interacting KdV solitons. This is a spectacular addition to the known connections between KPZ models and classical integrable systems.

Weaknesses

1- The main formulas are in a sense not new, but instead come from reformulating the earlier formulas by the author and by Baik-Liu. 2- The role played by the new expressions in terms of analytic continuation of polylogarithms and associated Riemann surfaces on the points discussed in Section 2.6 (such as the connection with KdV solitons) is not completely clear. 3- The paper is very heavy, and computations will be hard to understand for much of its audience (however, the author has included in the paper a relatively gentle introduction to Riemann surfaces and analytic continuation, in the form utilized in the article).

Report

Two of the main goals in research in the KPZ universality class over the last 15-20 years have been to compute the universal limiting distributions for models in the class in increasing levels of generality, and to uncover the mathematical and physical mechanisms underlying the rich structure of the class. This paper makes an important contribution in both these directions.

Most of the work related to the computation of exact limiting distributions has been done for models on the infinite (or sometimes half-infinite) line. During the last couple of years, new work has become available (mainly by the author on the one hand, and by Baik and Liu on the other) showing how Bethe ansatz techniques can be used to get formulas for models in finite volume with periodic boundary conditions. Moreover, these authors have been able to use these techniques to obtain formulas for multi-time distributions, which even in the full line had remained stubbornly out of reach (although, around the same time, Johansson was able to derive this type of formulas on the full line as well). The extension of these techniques to finite volume is non-trivial, and it leads to formulas which are very complicated, even when compared to similar formulas in the full line (which for fixed time are usually given as Fredholm determinants).

The bulk of the paper is devoted to uncovering what is arguably the right mathematical framework in which to express these formulas for KPZ in finite volume, leading in particular to much more unified expressions (for sharp wedge, flat, and stationary initial conditions, which are the ones for which earlier formulas were available), using tools from algebraic geometry. More precisely, the formulas make use of the analytic continuation of polylogarithms with half-integer index; in the flat case the formulas are written as integrals over a loop around an infinite cylinder of a holomorphic differential on the Riemann surface on which these polylogarithms are defined, summed over all the sheets of a ramified covering from this Riemann surface to the infinite cylinder (in the sharp wedge and stationary cases an additional sum is needed and a family of related Riemann surfaces have to be used). Multiple-time formulas are also written using this framework. In a restricted sense these formulas amount to a very clever rewriting of the already available ones (in particular this is how they are obtained). But they actually reveal the natural structure behind them: the original formulas involve a sum over integer subsets P of expressions involving functions related to polylogarithms and indexed by P; here these functions are understood as all coming from the analytic continuation of these polylogarithms, with P indexing the sheets in the associated Riemann surface. In particular, even after translating back from the algebraic geometry language to a formula in terms of sums over integer subsets, this point of view leads to formulas which are slightly simpler than the earlier ones. Another nice consequence of the analysis in the paper is the verification, which was previously lacking, that the author’s formulas coincide with those of Baik and Liu. A natural question opened by the paper, discussed briefly by the author, is whether these algebraic geometry tools can be used already at the level of a microscopic model such as TASEP, in order to provide a more natural framework for the computation of the complicated limits involved in the Bethe ansatz derivations for KPZ in finite volume. It will be interesting to see whether this approach is fruitful in coming years.

The author also makes a series of interesting connections which follow from his work (although it’s not fully clear to me to what extent they depend on the rest of the paper, see comment 3 below). The first one is with stationary large deviations; in particular, the algebraic geometry objects appearing in the formulas in the paper are connected to them. The second one is an interpretation of the formulas in terms of particle-hole excitations. The third one, which in my opinion is the most striking one, is a connection with KdV solitons: the author notices that the exact distribution for KPZ growth with periodic boundary conditions and flat initial data can be written as the average over a certain complex parameter ν of a certain tau function τ(t,x;ν) for the KdV equation (i.e. the second logarithmic derivative of τ solves KdV for each ν). This distribution is thus a certain average of the tau functions associated to a gas of infinitely many solitons with (complex) velocities indexed by ν (additionally a connection to the KP solitons is suggested in the sharp wedge case). This is a consequence of the form of the exact formulas, involving a Cauchy determinant and specific functional forms. Although the author quotes earlier suggestions that KPZ could be connected to KdV solitons, I think this link, particularly in this form, comes to a large extent as a surprise to much of the community. It should be mentioned also that in a paper posted with a couple of days difference, Quastel and Remenik showed that the exact distribution for KPZ growth in the full line is a KP tau function for general initial data (without any additional averaging) for any initial data. In any case, the integrable aspect of KPZ models and random matrices has been the subject of much interest for a long time, and these new connections with classical integrable systems show that this integrability is probably even deeper than previously thought.

The paper is very well written. Since much of its (mathematical and physical) audience is likely not be very deeply acquainted with the theory of analytic continuation and Riemann surfaces, it contains a relatively long introduction to the subject (at least the part that’s relevant to the paper) in Section. I am far from an expert in the algebraic geometry part of the paper, and I found this section to be useful in understanding the rest of it. For this reason, too, I could not check all of the derivation in full detail, but as far as I could check everything seemed correct, and in any case the derivation is very careful and detailed.

As I think should be clear by now, I have a very positive opinion of the paper. It makes an original contribution which should be of interest to both physicists and mathematicians working on exact formulas in the KPZ class, so I think it is a good candidate for publication at SciPost Physics.

There are four additional general comments that I think the author needs to address: 1- As stated in the abstract, a large part of the paper is devoted to the study of the Riemann surface associated to the analytic continuation of half-integer indices. However, from the writing it is not clear (to me at least) how much of what the author does in this respect is new. From a mathematical viewpoint, I’d suggest to make this more explicit, in particular those aspects that are more general (as opposed to being about very specific functions appearing in KPZ formulas). 2- This is related to my first comment. A naive reader may think that going from the earlier formulas to the new ones (Section 5) in this paper is relatively direct, but in fact much of the work is hidden in the rest of the paper, where the algebraic geometry objects needed for this is developed. I’d suggest to explain this more explicitly in Section 2 (I could have missed it). 3- An additional point needs to be clarified: how much of the discussion presented in Section 2.6 depends on, or is informed by, the new formulas? For example, the connection with particle-hole excitations seems to be there already in an earlier paper by the author. And in the case of KdV solitons, the derivation is based on Baik and Liu’s formulas involving a Fredholm determinant, and in view of the complicated formulas and the consequent, necessarily heavy, notation, it’s a bit hard to see exactly how much of a role the Riemann surface formulation is really playing. 4- There are parts in Section 4 where it is hard to tell what is supposed to be an easy computation, what is being left to the reader, and what is going to be proved later in the appendices. As a particular example, equations (106) and (107) are proved in Appendix B, but the text just says “one finds”, and no reference is made to the Appendix. This should be fixed, by making the right references, and pointing out clearly when something is being left out.

Requested changes

0- The four comments already made in the report section. 1- l.100. What do you mean by Z_λ being an Ito discretization? 2- l.175-176. At this stage in the text it is not clear what the “corresponding group actions” refer to, maybe make a reference. 3- l.401. I think it should say something like “The relation to KdV is most visible in the Fredholm det. expression”, which then should be referred to. 4- l.423. Footnote 4 here is a bit unclear. First, the exponential prefactor depends on t and x, so it’s not obvious that the equation is still satisfied. And second, the meaning of “the value of u(x,t)” is not completely clear (presumably it means that (20) is still satisfied). 5- l.463-490. The connection with KP in the sharp wedge case is a bit vague, as stated in the paper. It would be good, however, to explain more clearly what is being stated when saying that this case is “reminiscent” of KP at the beginning of that paragraph. Do you expect that a more concrete connection could be true, especially in view of the full line result? 6- l.542. “leads” → “lead” 7- l.565. “branch point” → “branch points” 8- l.566. Say what you mean by ≃. 9- Figure 4. f+ should be h+. 10- l.597. “neighboring” → “neighborhood” 11- l.598. Again f seems to be h. 12- l.689 and l.707. “Let … be a branched covering” 13- l.727. “allows one to recover” 14- l.756. “ending” 15- l.771. What is the connection to figure 3? 16- l.1110. I guess (63) should be (62).

  • validity: -
  • significance: high
  • originality: high
  • clarity: high
  • formatting: excellent
  • grammar: excellent

Author:  Sylvain Prolhac  on 2019-11-20  [id 650]

(in reply to Report 2 on 2019-10-25)

I thank the referee for the detailed review of the paper.

General comments: 1 - Analytic continuation of polylogarithms is indeed a well known subject. The parts of section 4 which are (presumably) new are the explicit analytic continuation of the functions J and K in terms of which KPZ fluctuations are expressed. This is now made more precise in the first paragraph of section 4.

2 - The fact that the equivalence between the old formulas and the new formulas depends heavily on analytic continuation results from section 4 was indeed mostly emphasized for flat initial condition in the previous version of the paper. I added some text around equation (10) and before equation (13) to make this explicit also for sharp wedge initial condition.

3 - While the relation of the Riemann surfaces with large deviations (section 2.6.1) and particle-hole excitations (section 2.6.2) is rather clear (the full time dynamics is in some sense given by large deviations "made analytic" by summing over all the sheets of the Riemann surfaces, which are in one to one correspondence with particle-hole excitations), I agree that the relation with KdV solitons is less clear (but also probably much deeper). In the classical theory of the KdV equation (which excludes the kind of initial conditions needed for KPZ, e.g. a Heaviside step function for KPZ with flat initial condition), any solution of KdV periodic in space is given in terms of a theta function built from a hyperelliptic compact Riemann surface, and soliton solutions correspond to specific degenerations of these Riemann surfaces. The soliton solution (23) can then be interpreted in terms of the theta function for a specific class of degenerate hyperelliptic Riemann surfaces with $\nu$ parametrizing the position of some branch points. This suggests that the correct interpretation of the parameter $\nu$ should not be a coordinate on a Riemann surface but rather a parameter in the space of Riemann surfaces (the moduli space). This might be crucial in order to understand in a natural way the factors $\Xi$ in (6) and (16). Additionally, integrating tau functions for KdV / KP with respect to moduli parameters seems to occur as well in several other contexts. I added a sentence in the first paragraph of section 2.6.3 to emphasize the moduli interpretation.

4 - For the example provided by the referee, appendix B is actually referred to in the sentence just before "One finds" (now: "There, one finds"). However, references to appendices were slightly broken in the previous version (I had changed from a style file that automatically added ``appendix'' when citing an appendix to the scipost one that does not). This has been corrected in the new version. In principle, formulas proved in the appendices have been indicated consistently throughout the paper.

Other comments: 0 - see above. 1 - I replaced "Ito discretization" by "Ito prescription in the time variable" in the new version. As usual for Langevin equations with multiplicative noise, the Stratonovich prescription (where an infinitesimal increment of Z depends both on Z before and after the increment) and the Ito prescription (where an infinitesimal increment of Z depends only on Z before the increment) lead to distinct solutions of the stochastic partial differential equation. This is the Ito prescription which leads to the proper physical object described by KPZ universality. 2 - References to sections 3.8.2, 3.8.2 have been added to the first paragraph of page 5. 3 - Done, references have been added to the first paragraph of section 2.6.3. 4 - Footnote 4 has been rewritten. Since the term in the exponential is only linear in $x$, its contribution to $u=2\partial_{x}^{2}\log\tau$ does vanish. 5 - KP is the "natural" generalization of KdV from 1 dimension of space to 2 dimensions of space from the point of view of classical integrability: modern books about classical non-linear integral equations such as [55-57] treat both together. Initially, since the dependency in x, y, t of the exponential factors match with those of KP solitons, I was quite convinced that tau functions for KP would appear for KPZ with sharp wedge initial condition. The fact that the prefactors do not match (the Cauchy matrix is squared in M_{sw} below (25)) was a bit disappointing. In view of the recent paper [63] by Quastel and Remenik, I am again inclined to believe that there should also be a proper connection to KP for KPZ with periodic boundaries and sharp wedge initial condition (at least; [63] seems to indicate that it should be much more general). I have not managed to find it so far. Maybe one should instead try to compare (25) to a N-soliton solution to a matrix KP equation, like the one appearing in [63], or maybe the full n-point function (16) has to be considered. I added a sentence about this in the last paragraph of section 2.6.3. 6 - Corrected. 7 - Corrected. 8 - $\simeq$ is the standard notation for the ratio going to 1 in the limit. I added "when $z\to z_{*}$" in order to specify the limit considered. 9 - Corrected. 10 - Corrected. 11 - Corrected. 12 - Corrected. 13 - Corrected. 14 - Corrected. 15 - The path $\theta_{j}^{2}\cdot\mathcal{G}_{P}$ corresponds in figure 3 left to the concatenation of the blue and green dashed paths, and in figure 3 right to the green/blue dashed circle. While on the right, it is clear that this path is homotopic to an empty loop, this is less obvious with the representation on the left. 16 - Yes, corrected.

---

## Round 1 · Referee Report · Anonymous (Referee 3) · 2019-11-10

Strengths

  1. The paper is interesting and important for general understanding of the exact formulas. For their apparent complexity such a general consideration can be a step towards construction of the general theory.

Weaknesses

  1. Author did his best to provide a self-contained introduction to the subject of Riemann surfaces and ramified coverings. However, the article is still not easy to read without good background in the subject.

Report

The manuscript is devoted to reconsideration of exact results for the KPZ fluctuations in periodic systems obtained previously from the Bethe ansatz solution by several authors including the author of the manuscript. The main aim of the paper is to put the existing formulas into the general context of the theory of ramified coverings on Riemann surfaces. The Author considers the formulas of the one-point distribution of KPZ interface height with flat, sharp wedge and stationary initial conditions as well as the mutipoint distribution for sharp wedge initial conditions. It is shown that they can be rewritten in terms of integrals of some homomorphic differentials over the path closed around the cylinder obtained as a quotient of infinitely dimensional Riemann surfaces, which are in turn quotients of the Riemann surface of polylogarithms with half-integer indices. The major part of the manuscript is the introduction to the theory of ramified coverings on the Riemann surfaces. It serves for introducing definitions and objects used to formulate the final results. After fixing necessary concepts Author performs technical but straightforward calculations to show that the announced formulas indeed coincide with the known results. As an important by-product Author proves that different formulas for alike distributions obtained previously by different methods are in fact identical. Also several important connections of the results obtained are discussed.

The paper is interesting and important for general understanding of the exact formulas. For their apparent complexity such a general consideration can be a step towards construction of the general theory. The text is well written. Author did his best to provide a self-contained introduction to the subject of Riemann surfaces and ramified coverings. However, the article is still not easy to read without good background in the subject. Nevertheless, I do not think that the manuscript can be substantially improved in this respect in any way. To conclude, I believe that the manuscript is suitable for publication after correcting the mistypes.

Requested changes

There are two mistypes I've found:
line 249: "a" is missing from the sum in the inline formula
line 598: f should be replaced by h

  • validity: high
  • significance: high
  • originality: top
  • clarity: good
  • formatting: excellent
  • grammar: excellent

Author:  Sylvain Prolhac  on 2019-11-20  [id 649]

(in reply to Report 3 on 2019-11-10)

I thank the referee for the positive feedback on the paper. Both misprints have been corrected.

---

## Round 4 · Referee Report · Anonymous · 2019-12-18

Report

The author complied with all the requirements of referees. The manuscript can be published as it is.

---

## Round 4 · Referee Report · Anonymous · 2019-12-21

Report

All my earlier comments have been addressed. I believe the article can be accepted now for publication.

---

## Round 4 · Referee Report · Anonymous · 2020-1-11

Report

The author improved his manuscript in this revision. I recommend the publication of this paper.

---

## Round 4 · List of Changes

Referee 1:
I thank the referee for the positive feedback on the paper.

Referee 2:
I thank the referee for the detailed review of the paper.

General comments:
1 - Analytic continuation of polylogarithms is indeed a well known subject. The parts of section 4 which are (presumably) new are the explicit analytic continuation of the functions J and K in terms of which KPZ fluctuations are expressed. This is now made more precise in the first paragraph of section 4.

2 - The fact that the equivalence between the old formulas and the new formulas depends heavily on analytic continuation results from section 4 was indeed mostly emphasized for flat initial condition in the previous version of the paper. I added some text around equation (10) and before equation (13) to make this explicit also for sharp wedge initial condition.

3 - While the relation of the Riemann surfaces with large deviations (section 2.6.1) and particle-hole excitations (section 2.6.2) is rather clear (the full time dynamics is in some sense given by large deviations ``made analytic'' by summing over all the sheets of the Riemann surfaces, which are in one to one correspondence with particle-hole excitations), I agree that the relation with KdV solitons is less clear (but also probably much deeper). In the classical theory of the KdV equation (which excludes the kind of initial conditions needed for KPZ, e.g. a Heaviside step function for KPZ with flat initial condition), any solution of KdV periodic in space is given in terms of a theta function built from a hyperelliptic compact Riemann surface, and soliton solutions correspond to specific degenerations of these Riemann surfaces. The soliton solution (23) can then be interpreted in terms of the theta function for a specific class of degenerate hyperelliptic Riemann surfaces with $\nu$ parametrizing the position of some branch points. This suggests that the correct interpretation of the parameter $\nu$ should not be a coordinate on a Riemann surface but rather a parameter in the space of Riemann surfaces (the moduli space). This might be crucial in order to understand in a natural way the factors $\Xi$ in (6) and (16). Additionally, integrating tau functions for KdV / KP with respect to moduli parameters seems to occur as well in several other contexts. I added a sentence in the first paragraph of section 2.6.3 to emphasize the moduli interpretation.

4 - For the example provided by the referee, appendix B is actually referred to in the sentence just before ``One finds'' (now: ``There, one finds''). However, references to appendices were slightly broken in the previous version (I had changed from a style file that automatically added ``appendix'' when citing an appendix to the scipost one that does not). This has been corrected in the new version. In principle, formulas proved in the appendices have been indicated consistently throughout the paper.

Other comments:
0 - see above.
1 - I replaced ``Ito discretization'' by ``Ito prescription in the time variable'' in the new version. As usual for Langevin equations with multiplicative noise, the Stratonovich prescription (where an infinitesimal increment of Z depends both on Z before and after the increment) and the Ito prescription (where an infinitesimal increment of Z depends only on Z before the increment) lead to distinct solutions of the stochastic partial differential equation. This is the Ito prescription which leads to the proper physical object described by KPZ universality.
2 - References to sections 3.8.2, 3.8.2 have been added to the first paragraph of page 5.
3 - Done, references have been added to the first paragraph of section 2.6.3.
4 - Footnote 4 has been rewritten. Since the term in the exponential is only linear in $x$, its contribution to $u=2\partial_{x}^{2}\log\tau$ does vanish.
5 - KP is the ``natural'' generalization of KdV from 1 dimension of space to 2 dimensions of space from the point of view of classical integrability: modern books about classical non-linear integral equations such as [55-57] treat both together. Initially, since the dependency in x, y, t of the exponential factors match with those of KP solitons, I was quite convinced that tau functions for KP would appear for KPZ with sharp wedge initial condition. The fact that the prefactors do not match (the Cauchy matrix is squared in M_{sw} below (25)) was a bit disappointing. In view of the recent paper [63] by Quastel and Remenik, I am again enclined to believe that there should also be a proper connection to KP for KPZ with periodic boundaries and sharp wedge initial condition (at least; [63] seems to indicate that it should be much more general). I have not managed to find it so far. Maybe one should instead try to compare (25) to a N-soliton solution to a matrix KP equation, like the one appearing in [63], or maybe the full n-point function (16) has to be considered. I added a sentence about this in the last paragraph of section 2.6.3.
6 - Corrected.
7 - Corrected.
8 - $\simeq$ is the standard notation for the ratio going to 1 in the limit. I added ``when $z\to z_{*}$'' in order to specify the limit considered.
9 - Corrected.
10 - Corrected.
11 - Corrected.
12 - Corrected.
13 - Corrected.
14 - Corrected.
15 - The path $\theta_{j}^{2}\!\cdot\!\mathcal{G}_{P}$ corresponds in figure 3 left to the concatenation of the blue and green dashed paths, and in figure 3 right to the green/blue dashed circle. While on the right, it is clear that this path is homotopic to an empty loop, this is less obvious with the representation on the left.
16 - Yes, corrected.

Referee 3:
I thank the referee for the positive feedback on the paper. Both misprints have been corrected.

Additional changes:
Added $t_0=u_0=0$ after equation (13) and $x_0=0$ after equation (14).

---

## Editorial Decision

published